# Fast Non-Log-Concave Sampling under Nonconvex Equality and Inequality Constraints with Landing

**Kijung Jeon**
Georgia Institute of Technology
kjeon@gatech.edu

**Michael Muehlebach**
MPI-IS
michaelm@tue.mpg.de

**Molei Tao**
Georgia Institute of Technology
mtao@gatech.edu

## Abstract

Sampling from constrained statistical distributions is a fundamental task in various fields including Bayesian statistics, computational chemistry, and statistical physics. This article considers sampling from a constrained distribution that is described by an unconstrained density, as well as additional equality and/or inequality constraints, which often make the constraint set nonconvex. Existing methods struggle in the presence of such nonconvex constraints, as they rely on projections, which are computationally expensive or intractable, are specialized to either inequality or equality constraints, and often lack rigorous quantitative convergence guarantees. In this paper, we introduce *Overdamped Langevin with LAnding* (OLLA), a new framework that can design overdamped Langevin dynamics accommodating both nonlinear equality and inequality constraints. The proposed dynamics also deterministically corrects trajectories along the normal direction of the constraint surface, thus obviating the need for explicit projections. We show that, under suitable regularity conditions on the target density and the feasible set $\Sigma \subset \mathbb{R}^d$, OLLA converges exponentially fast in 2-Wasserstein distance to the constrained target density $\rho_\Sigma(x) \propto \exp(-f(x))d\sigma_\Sigma$. Lastly, through experiments, we demonstrate the efficiency of OLLA compared to known constrained Langevin algorithms and their slack variable variants, highlighting its favorable computational cost and fast empirical mixing.[1]

## 1 Introduction

Sampling from complex, constrained statistical distributions is a fundamental problem in machine learning, with applications ranging from Bayesian inference under structured priors to training generative models with safety or fairness constraints. When there is no constraint, a prominent class of sampling techniques is centered around (overdamped) Langevin dynamics, where the drift is set to the gradient of the log-target density. These have gained significant traction due to their strong theoretical guarantees: for example under log-concave target densities [1–8] or more general densities that satisfy relaxed conditions such as isoperimetric inequalities [9, 10] one can obtain fast, non-asymptotic convergence rates. Langevin dynamics can even be generalized, via a Riemannian Langevin approach, to sample under convex constraints [e.g., 11, 12]. However, extending Langevin-based approaches to sample from distributions supported on nonconvex sets $\Sigma \subset \mathbb{R}^d$ remains a major challenge. Existing techniques typically rely on projection steps, which are computationally expensive and require convexity to ensure convergence. Moreover, most methods offer limited or no quantitative convergence guarantees in this case; in fact, even if the density is log-concave, a nonconvex constraint can easily make the target distribution much harder to sample from, also rendering the analysis more difficult. This is a critical bottleneck for many emerging machine learning

---

[1]All code is provided in the following repository: https://github.com/KraitGit/OLLA

39th Conference on Neural Information Processing Systems (NeurIPS 2025).

applications – such as imitation learning [13] or constrained generative modeling – where the feasible set is implicitly defined by complex equality and inequality constraints.

In this article, we introduce OLLA (Overdamped Langevin with LAnding), a suite of projection-free stochastic dynamics that could serve as the foundation for sampling from constrained distributions. OLLA avoids projections by combining two key ideas: (1) relying on local approximations of the feasible set to guide the sampling and (2) introducing a restitution mechanism called "landing" that guarantees convergence to the feasible set. These techniques build on ideas previously developed in the context of nonconvex optimization, where they have been shown to provide scalable and effective algorithms [14–16], and share close connection to several powerful constrained sampling approaches in the literature [17–21]. We adapt and extend the ideas to the sampling setting, which makes OLLA both computationally efficient and theoretically grounded. Our main contributions are summarized as follows:

- **(Unified treatment of constraints)** OLLA is described by a stochastic differential equation (SDE) that, in contrast to related prior works, enforces both equality and inequality constraints. This is achieved by constructing the tangent space to the constraint manifold and projecting the overdamped Langevin drift and diffusion terms onto the tangent space resulting in a simple least-squares problem. OLLA recovers the classical equality-only constrained Langevin dynamics as a special case, yet seamlessly accommodates arbitrary smooth inequality constraints without resorting to slack variables or projections.
- **(Exponential convergence)** We prove that the continuous version of OLLA converges to the constrained target distribution $\rho_\Sigma$ at an exponential rate under appropriate regularity assumptions. Our convergence results are non-asymptotic and characterized by the 2-Wasserstein distance in all scenarios (equality constraints only, inequality constraints only, and mixed).
- **(Efficient SDE discretization with trace-estimation)** We introduce OLLA-H, a computationally efficient Euler-Maruyama (EM) discretization of the aforementioned SDE that features a Hutchinson trace estimator [22] for approximating the Itô-Stratonovich correction term arising from the diffusion. As a result, OLLA-H has low computational cost per iteration, even in high dimensions, and it achieves relatively accurate sampling and empirical mixing, an aspect that we demonstrate in various numerical experiments.

## 2 Related Works

We first focus our literature review on recent works that consider Langevin sampling under nonlinear constraints. One of the closest touching points is [17], which describes a gradient descent approach on the KL divergence. The algorithm shares some similarities to ours in that projections are avoided, but the work focused on equality constraints only, whereas OLLA covers both equality and inequality constraints. The work [23] proposes the use of slack variables to incorporate inequality constraints to change a mixed problem into an equality-only problem, which comes at the cost of additional spurious dimensions. In a similar vein, [18] designs a particle-based variational inference method to incorporate inequality constraints. The method is effective at sampling under inequality constraints only, suffers, however, from high computational cost in high dimensions due to the estimation of associated boundary integrals.

OLLA is inspired by recent methods in nonlinear optimization [15, 14, 16] that use a similar landing mechanism and avoid projections onto the feasible set. There has also been important work by [19–21] who introduced a constrained Langevin dynamics based on numerical schemes such as SHAKE, RATTLE [24–26], and including Metropolis-Hastings corrections [21]. These works mainly focus on equality-only constraints, although inequality constraints can be incorporated via including slack variables or applying reflection at the boundary. We use these algorithms as baselines and refer them to Constrained Langevin (CLangevin) [20], Constrained Hamiltonian Monte Carlo (CHMC) [20], and Constrained generalized Hybrid Monte Carlo (CGHMC) [21].

In the present work, constraints are handled through a careful decomposition of the stochastic dynamics on the boundary into normal and tangential parts, thereby avoiding projections and even enabling infeasible initialization. There have also been alternative algorithm designs that, however, do not share these features, for example, [27, 28] (based on projection), [29, 30] (barrier functions), [31] (reflections), or [11, 32, 12] (mirror maps). Other works by [33, 34] introduce penalties for constraint violations or relax the notion of constraint satisfaction [35]. In addition, we note that, although closely related, constrained sampling is not the same as sampling on manifolds [36–39].

# 3 Preliminaries & Notations

We consider sampling from a target density supported on the compact and connected Riemannian submanifold of $\mathbb{R}^d$ defined by $\Sigma := \{x \in \mathbb{R}^d \mid h(x) = 0, g(x) \le 0\}$ where $h : \mathbb{R}^d \to \mathbb{R}^m$ and $g : \mathbb{R}^d \to \mathbb{R}^l$ are smooth functions. To guarantee that all constraint-related constructions are well-posed, we impose the Linear Independence Constraint Qualification (LICQ) condition [40].

> **Definition 1.** The functions $h, g$ satisfy LICQ if $\{\nabla h_1(x), ... \nabla h_m(x)\} \cup \{\nabla g_i(x)\}_{i \in I_x}$ are linearly independent for every $x \in \Sigma$, where $I_x$ denotes the set of active inequality constraints, i.e., $I_x := \{i \in [l] \mid g_i(x) \ge 0\}$.

As a result of LICQ, the tangent space of $\Sigma$ at $x \in \Sigma$ can be defined as

$$T_x\Sigma := \{v \in \mathbb{R}^d \mid \nabla h(x)v = 0, \nabla g_i(x)v = 0, \ \forall i \in I_x\}$$

and its orthogonal projector onto $T_x\Sigma$ is $\Pi(x) = I - \nabla J(x)^T G(x)^{-1} \nabla J(x)$, where $J(x) := \left[h_1(x), ..., h_m(x), g_{i_1}(x) + \epsilon, ..., g_{i_{|I_x|}}(x) + \epsilon\right]^T$, $\{i_1, ..., i_{|I_x|}\} = I_x$ denotes the stacking of constraint functions for some $\epsilon > 0$ and $G(x) := \nabla J(x)\nabla J(x)^T$ is its associated Gram matrix.

For any smooth $f : \Sigma \to \mathbb{R}$ and a smooth vector field $X : \Sigma \to \mathbb{R}^d$, the intrinsic gradient and divergence on $\Sigma$ are given by $\nabla_\Sigma f(x) = \Pi(x)\nabla f(x)$, $\mathrm{div}_\Sigma X(x) = \mathrm{Tr}\left(\Pi(x)\nabla X(x)\right)$, where $\nabla$ denotes the usual Euclidean gradient or Jacobian in $\mathbb{R}^d$. Our target density is set to be $\rho_\Sigma(x) \propto \exp(-f(x))d\sigma_\Sigma$ on $\Sigma$ with $d\sigma_\Sigma$ being the induced Hausdorff measure on $\Sigma$. We write $\rho_t, \tilde{\rho}_t$ be the law of OLLA and the projected process of OLLA onto $\Sigma$ at time $t$. On $\Sigma$, the natural extension of KL divergence and Fisher information take the form:

$$\mathsf{KL}^\Sigma(\rho\|\rho_\Sigma) := \int_\Sigma \rho \ln\left(\frac{\rho}{\rho_\Sigma}\right) d\sigma_\Sigma \quad \text{and} \quad I^\Sigma(\rho\|\rho_\Sigma) := \int_\Sigma \rho\|\nabla_\Sigma \ln\left(\frac{\rho}{\rho_\Sigma}\right)\|_2^2 d\sigma_\Sigma$$

We now streamline notations appearing in Section 4. A complete list of symbols and precise technical definitions are included in Appendix A. In particular, let $\pi(x)$ denote the nearest-point (Euclidean) projection onto $\Sigma$, and let $\lambda_{\mathrm{LSI}}$ be the log-Sobolev constant of $(\Sigma, \rho_\Sigma)$. We then assume the existence of the following constants.

$$M_h := \sup_{x_0 \in \mathrm{supp}(\rho_0)} \|h(x_0)\|_2, \quad M_g := \sup_{x_0 \in \mathrm{supp}(\rho_0)} \|g(x_0)\|_2,$$

over the support of the initial law $\rho_0$. The constant $\kappa$ (Lemma C.1, Lemma C.3) and $\delta$ captures the regularity of $\Sigma$ and $\hat{U}_\delta$ denotes the tubular neighborhood of width $\delta$ with a special "recovery" property (see Theorem C.1, Theorem C.2 for the precise definitions of $\delta$ and $\hat{U}_\delta$).

# 4 Main results

## 4.1 Construction of OLLA via Least Squares

We now derive the continuous-time dynamics of OLLA by choosing the drift vector $q$ and the symmetric diffusion matrix $Q$ to be the closest—in a least-squares sense—to the unconstrained usual Langevin coefficients, subject to enforcing both the equality constraints $\{h_i(x)\}_{i=1}^m$ and the active inequality constraints $\{g_j(x)\}_{j \in I_x}$. This is achieved by applying Itô's lemma to each constraint function $h_i$ and $g_j$ and splitting the change in, for example, $h_i$, into a martingale term $\nabla h_i(X_t)^T Q(X_t)dW_t$ and a drift term $\left[\nabla h_i(X_t)^T q(X_t) + \frac{1}{2}\mathrm{Tr}\left(\nabla^2 h_i(X_t)Q(X_t)Q(X_t)^T\right)\right] dt$. By choosing $Q$ so that $Q(x)\nabla h_i(x) = Q(x)\nabla g_j(x) = 0$, the martingale piece vanishes exactly in the normal directions. Simultaneously, we pick the drift vector $q$ to satisfy the linear equation

$$\nabla h_i^T q + \frac{1}{2}\mathrm{Tr}\left(\nabla^2 h_i QQ^T\right) + \alpha h_i = 0, \quad \nabla g_j^T q + \frac{1}{2}\mathrm{Tr}\left(\nabla^2 g_j QQ^T\right) + \alpha(g_j + \epsilon) = 0 \quad (1)$$

so that $h(X_t) = h_i(X_0)e^{-\alpha t}$ and $g_j(X_t) + \epsilon = (g_j(X_0) + \epsilon)e^{-\alpha t}$, where hyperparameters $\alpha, \epsilon > 0$ denote the landing or boundary repulsion rate, respectively. This enforces $g_j(X_t)$ to hit 0 in finite time $t = \frac{1}{\alpha}\ln((g_j(X_0) + \epsilon)/\epsilon)$, after which $g_j(X_t) \le 0$ remains forever. As a result, this approach removes any noise and drift direction in the normal of constraints and implants a pure drift normal to constraints, guaranteeing exponential decay of both equality and active-inequality constraints at a rate $\alpha$. This yields the closed-form SDE in Proposition 1 and Lemma 1.

**Proposition 1** (Construction of OLLA and its closed form SDE). *Consider the following SDE:*

$$dX_t = q(X_t)dt + Q(X_t)dW_t \tag{2}$$

*where*

$$Q := \underset{\bar{Q} \in \mathbb{R}^{d \times d}}{\text{argmin}} \| \sqrt{2}I - \bar{Q} \|_F^2 \quad s.t \quad \begin{cases} \bar{Q}\nabla h_i = 0, \ \forall i \in [m], \\ \bar{Q}\nabla g_j = 0, \ \forall j \in I_x, \end{cases}$$

$$q := \underset{\bar{q} \in \mathbb{R}^d}{\text{argmin}} \| \bar{q} + \nabla f \|_2^2 \quad s.t \quad \begin{cases} \nabla h_i^T \bar{q} + \frac{1}{2} \mathit{Tr}\left(\nabla^2 h_i Q Q^T\right) + \alpha h_i = 0, & \forall i \in [m], \\ \nabla g_j^T \bar{q} + \frac{1}{2} \mathit{Tr}\left(\nabla^2 g_j Q Q^T\right) + \alpha(g_j + \epsilon) = 0, & \forall j \in I_x. \end{cases}$$

*Then, there exists a closed form SDE (OLLA) of* (2) *given by:*

$$dX_t = -[\Pi(X_t)\nabla f(X_t) + \alpha \nabla J(X_t)^T G^{-1}(X_t)J(X_t)]dt + \mathcal{H}(X_t)dt + \sqrt{2}\Pi(X_t)dW_t \tag{3}$$

*where*

$$\mathcal{H} := -\nabla J^T G^{-1} \left[ \mathit{Tr}\left(\nabla^2 h_1 \Pi\right), ..., \mathit{Tr}\left(\nabla^2 h_m \Pi\right), \mathit{Tr}\left(\nabla^2 g_{i_1} \Pi\right), ..., \mathit{Tr}\left(\nabla^2 g_{i_{|I_x|}} \Pi\right) \right]^T \tag{4}$$

*is the associated mean curvature correction term of* $\Sigma_{I_x} := \left\{ x \in \mathbb{R}^d \mid h(x) = 0, g_{I_x}(x) = 0 \right\}$.

**Remark 1** (Mean curvature = Itô-Stratonovich correction). By technical stochastic-calculus identities on manifolds (see, e.g., Rousset et al. [19], Lemma 3.19), the Itô-Stratonovich correction arising from the Stratonovich SDE

$$dX_t = -[\Pi(X_t)\nabla f(X_t) + \alpha \nabla J(X_t)^T G^{-1}(X_t)J(X_t)]dt + \sqrt{2}\Pi(X_t) \circ dW_t \tag{5}$$

coincides exactly with the mean-curvature term $\mathcal{H}(x)$ of $\Sigma_{I_x} := \left\{ x \in \mathbb{R}^d \mid h(x) = 0, g_{I_x}(x) = 0 \right\}$.

**Remark 2** (Relation to orthogonal direction samplers from variational KL). Zhang et al. [17] introduced an overdamped-Langevin sampler for equality constraints by minimizing the constrained KL divergence via an orthogonal-space variational formulation, and Zhang et al. [18] also used a similar approach to handle single inequality constraint. In the absence of inequality constraints, our OLLA dynamics coincide with the equality constrained sampler of [17] up to a modified potential $\hat{f}(x) = f(x) + \frac{1}{2}\ln\det(G)$, since the mean curvature correction satisfies

$$\mathcal{H}(x) = \mathsf{div}\Pi(x) + \Pi(x)\nabla\ln\left((\det G(x))^{\frac{1}{2}}\right) \tag{6}$$

(see Rousset et al. [19], Lemma 3.21). Correspondingly, whereas their framework yields a stationary measure proportional to $e^{-f(x)}\delta_\Sigma(dx)$ under the coarea formula, OLLA converges to the Riemannian volume-weighted density $e^{-f(x)}d\sigma_\Sigma(x)$ on $\Sigma$ (see Rousset et al. [19], Lemma 3.2). In addition to this, the work [18] enforces a single inequality constraint via a purely deterministic normal drift $-\alpha\nabla g/\|\nabla g\|_2$, without any stochastic component in the tangential direction. OLLA instead projects noise and drift vectors tangentially even during the landing phase, thereby preserving exploration on the evolving manifold $\Sigma^t := \left\{ x \in \mathbb{R}^d \mid h(x) = h(X_0)e^{-\alpha t} \right\}$ and improving mixing.

**Lemma 1** (Exponential decay of constraint functions). *The dynamics induced by* (3) *satisfy the following properties almost surely for* $\forall i \in [m], \forall j \in I_{X_0}$:

$$h_i(X_t) = h_i(X_0)e^{-\alpha t}, \quad t \geq 0 \tag{7}$$

*and*

$$\begin{cases} g_j(X_t) = -\epsilon + (g_j(X_0) + \epsilon)e^{-\alpha t}, & t \leq \frac{1}{\alpha}\ln\left(\frac{g_j(X_0) + \epsilon}{\epsilon}\right) \\ g_j(X_t) \leq 0, & t \geq \frac{1}{\alpha}\ln\left(\frac{g_j(X_0) + \epsilon}{\epsilon}\right) \end{cases}$$

*with* $g(X_t) \leq 0, \forall t \geq 0$ *for* $j \notin I_{X_0}$, *where* $I_x := \{k \in [l] \mid g_k(x) \geq 0\}$ *is the index set of active inequality constraints.*

## 4.2 Non-asymptotic Convergence Analysis of OLLA

In this subsection, we establish non-asymptotic convergence guarantees for OLLA in three scenarios – equality only case, inequality only case, mixed case – by recognizing that OLLA has a rapid landing

property driven by landing rate $\alpha$ and natural mixing on the landed manifold induced by the LSI constant. For the detailed notations and proofs, we refer to the Appendix A and related Appendices.

**Equality-Only Scenario.** When the constraints consist of only smooth equalities $h(x) = 0$, the continuous-time OLLA dynamics (3) can be written as

$$dX_t = - \left[ \Pi(X_t)\nabla f(X_t) + \alpha\nabla h(X_t)^T G^{-1}(X_t)h(X_t) \right] dt + \sqrt{2}\Pi(X_t) \circ dW_t \qquad (8)$$

The drift term naturally decomposes into a tangential term, which moves along the constraint surface, and a normal landing term, which forces each coordinate $h_i(X_t)$ to decay exponentially fast, as summarized in Lemma 1.

Once $X_t \sim \rho_t$ lies within a tubular neighborhood of $\Sigma$, the nearest projection map $\pi$ onto $\Sigma$ becomes available, and the projected process $Y_t = \pi(X_t) \sim \tilde{\rho}_t$ can be defined (Theorem C.1). Then, the regularity of $\Sigma$ naturally implies that $\|X_t - Y_t\|_2 \lesssim \|h(X_t)\|_2 = \mathcal{O}(e^{-\alpha t})$ holds (Lemma C.1), leading to the contraction of $W_2(\rho_t, \tilde{\rho}_t) = \mathcal{O}(e^{-\alpha t})$ (Lemma E.1), where $\rho_t, \tilde{\rho}_t$ are the laws of $X_t, Y_t$, respectively. Furthermore, the combination of Lipschitzness of $\pi$ and $\|X_t - Y_t\|_2 = \mathcal{O}(e^{-\alpha t})$ enable us to write the projected process $Y_t$ as overdamped Langevin dynamics on $\Sigma$ with noisy drift vector and diffusion matrix whose norm is bounded by $\mathcal{O}(e^{-\alpha t})$ (Corollary E.1). Therefore, the effect of noisy terms can be dominated by the effect of the LSI constant, which leads to the exponentially fast convergence of $\mathsf{KL}^\Sigma(\tilde{\rho}_t \| \rho_\Sigma)$ (Lemma E.3). A rigorous combination of these insights gives the following theorem.

**Theorem 1** (Convergence result for equality-constrained OLLA). *Suppose assumptions (C1) to (C4) hold. Let $X_t$ be the stochastic process following the equality-constrained OLLA and let $\rho_t, \tilde{\rho}_t$ be the law of $X_t$ and its projection $Y_t = \pi(X_t)$ on $\Sigma$ for $t \geq t_{cut}$, $t_{cut} := \max\left\{\frac{1}{\alpha}\ln\delta, \frac{1}{\alpha}\ln C_5\right\}$, respectively. Then, for $\alpha > 2\lambda_{LSI}$ for all $t \geq t_{cut}$, it holds that*

$$W_2(\rho_t, \rho_\Sigma) \leq \frac{M_h}{\kappa}e^{-\alpha t} + \sqrt{\frac{2}{\lambda_{LSI}}\mathsf{KL}^\Sigma(\tilde{\rho}_t \| \rho_\Sigma)}$$

*where*

$$\mathsf{KL}^\Sigma(\tilde{\rho}_t \| \rho_\Sigma) \leq \exp\left(-2\lambda_{LSI}(t - t_{cut}) - \frac{2\lambda_{LSI}C_5}{\alpha}(e^{-\alpha t} - e^{-\alpha t_{cut}})\right)\left[\mathsf{KL}^\Sigma(\tilde{\rho}_{t_{cut}} \| \rho_\Sigma) + C_7\right]$$

*for some constants $C_5 = \mathcal{O}\left(1 + \frac{C_{L_A}M_h}{\kappa} + \left(\frac{C_{L_A}M_h}{\kappa}\right)^2\right)$, $C_7 := \frac{C_6 e^{-\alpha t_{cut}}}{\alpha - 2\lambda_{LSI}} > 0$ with $C_{L_A}$ being the Lipschitz constant of $\nabla\pi(x)\Pi(x)$ on $\hat{U}_\delta(\Sigma)$.*

**Inequality-Only Scenario.** With smooth inequalities $g(x) \leq 0$, we introduced a small boundary repulsion parameter $\epsilon > 0$ so that each initially violated constraint $g_j > 0$ is driven to the boundary within a finite landing time $t_{cut} = \frac{1}{\alpha}\ln((g_j(X_0) + \epsilon)/\epsilon)$.

From the Fokker-Planck perspective, the normal probability flux at the boundary satisfies $\langle n, J_t \rangle = -\alpha\epsilon\rho_t$ where $J_t$ is the probability current density of $\rho_t$ and $n$ is the outward unit normal vector of $\Sigma$. The boundary repulsion enforces the outward probability flow to become zero after the landing time $t_{cut}$, and the normal probability flux vanishes to zero after time $t_{cut}$ (Lemma F.1). This enables us to ignore the boundary integral appearing on the time derivative of $\mathsf{KL}(\rho_t \| \rho_\Sigma)$ guaranteeing exponential decay driven by the effect of the LSI constant. Therefore, the following theorem comes by directly analyzing the time derivative of $\mathsf{KL}(\rho_t \| \rho_\Sigma)$ after time $t_{cut}$.

**Theorem 2** (Convergence result for inequality-constrained OLLA). *Suppose assumptions (C1) to (C4) hold. Let $X_t$ be the stochastic process following the inequality-constrained OLLA and let $\rho_t$ be the law of $X_t$. Then, for $t \geq t_{cut}, t_{cut} := \frac{1}{\alpha}\ln\left(\frac{M_g + \epsilon}{\epsilon}\right)$,*

$$W_2(\rho_t, \rho_\Sigma) \leq \sqrt{\frac{2}{\lambda_{LSI}}\mathsf{KL}^\Sigma(\rho_t \| \rho_\Sigma)}$$

*where*

$$\mathsf{KL}^\Sigma(\rho_t \| \rho_\Sigma) \leq e^{-2\lambda_{LSI}(t - t_{cut})}\mathsf{KL}^\Sigma(\rho_{t_{cut}} \| \rho_\Sigma).$$

**Mixed Scenario.** In the mixed setting, OLLA's dynamics are sensitive to the boundary repulsion from $g$. It is noteworthy that $\Sigma$ remains unchanged for different inequality functions $g$, as long as the boundary of the feasible set, where $g(x) = 0$ intersects $h(x) = 0$, is identical. Nevertheless, the choice of function $g$ affects the landing dynamics, which in turn can alter the convergence rate of OLLA.

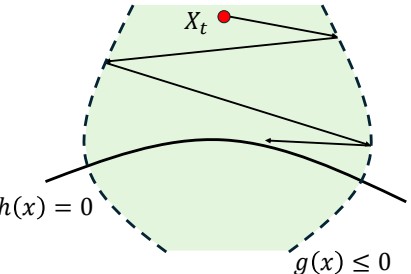

Figure 1: OLLA trajectory in mixed case

To quantify this, we define the projected manifold $\Sigma_p := \pi(\{x \in \mathbb{R}^d \mid h(x) = p, g(x) \le 0\})$ and assume the norm of boundary velocity $v_p^b$ of $\partial\Sigma_p$ is regulated by $V\|p\|_2^\beta$ for some $V, \beta > 0$ (Assumption (M2)). Additionally, we assume $\Sigma_p$ lies inside $\text{int}(\Sigma)$ for $0 < \|p\|_2 < \delta$ to avoid stopping behavior of $Y_t$ on $\partial\Sigma$ (Assumption (M1)). Under these assumptions, the trajectory of $X_t$ can be illustrated as in Figure 1 and the proof ideas of equality-only and inequality-only can be seamlessly combined. The following theorem arises from a rigorous integration of previous high-level ideas. We also provide a theorem with a relaxed version of Assumption (M1) in the appendix; see Remark 3 (when $X_0 \sim \delta(x_0)$), Remark 4, and Corollary A.1 for the details.

**Theorem 3** (Convergence result for mixed-constrained OLLA). *Suppose assumptions (C1) to (C4) and (M1) to (M3) hold. Define $X_t$ to be the stochastic process following OLLA dynamics (3) and $\tilde{\rho}_t$ be the law of $Y_t := \pi(X_t)$ after $t \ge t_{cut}$, $t_{cut} := \max\left\{ \frac{1}{\alpha}\ln\left(\frac{M_g + \epsilon}{\epsilon}\right), \frac{1}{\alpha}\ln\left(\frac{M_h}{\delta}\right), \frac{1}{\alpha}\ln(\tilde{C}_5) \right\}$. Then, for $\alpha > 2\lambda_{LSI}$ and $\beta \ge 1$, the following non-asymptotic convergence rate of $W_2(\rho_t, \rho_\Sigma)$ can be obtained as follows*

$$W_2(\rho_t, \rho_\Sigma) \le \frac{M_h}{\kappa}e^{-\alpha t} + \sqrt{\frac{2}{\lambda_{LSI}}\mathsf{KL}^\Sigma(\tilde{\rho}_t \| \rho_\Sigma)}$$

*where*

$$\mathsf{KL}^\Sigma(\tilde{\rho}_t \| \rho_\Sigma) \le \exp\left( -2\lambda_{LSI}(t - t_{cut}) - \frac{2\lambda_{LSI}\tilde{C}_5}{\alpha}(e^{-\alpha t} - e^{-\alpha t_{cut}}) \right) \left[ \mathsf{KL}^\Sigma(\tilde{\rho}_{t_{cut}} \| \rho_\Sigma) + \tilde{C}_7 + \tilde{C}_8 \right]$$

*for some constants $G_4, G_5, G_6, \tilde{C}_6 > 0$, $\tilde{C}_7 := (\tilde{C}_6 + \alpha G_4 G_5 M_h)\frac{e^{-\alpha t_{cut}}}{\alpha - 2\lambda_{LSI}}$,*

$$\tilde{C}_5 = \mathcal{O}\left(1 + \frac{\tilde{C}_{L_A} M_h}{\kappa} + \left(\frac{\tilde{C}_{L_A} M_h}{\kappa}\right)^2\right), \quad \tilde{C}_8 := (G_6 V M_h^\beta)\frac{e^{-\alpha\beta t_{cut}}}{\alpha\beta - 2\lambda_{LSI}},$$

*and with $\tilde{C}_{L_A}$ being the Lipschitz constant of $\nabla\pi(x)\Pi(x)$ on $\hat{U}_\delta(\Sigma)$.*

### 4.3 Euler-Maruyama Discretization & Hutchinson Trace Estimation

To implement OLLA in practice, we discretize the continuous-time SDE by the Euler-Maruyama (EM) update. At each iteration, we compute three components for the drift vector: the projected gradient drift, the landing drift, and the mean curvature correction $\mathcal{H}$. In particular, $\mathcal{H}$ (4) requires forming the full Hessian of each constraint and computing traces of the form $\mathsf{Tr}\left(\Pi\nabla^2 h_i\right)$ and $\mathsf{Tr}\left(\Pi\nabla^2 g_j\right)$, an $\mathcal{O}(d \cdot \text{grad-cost})$ operation that quickly becomes infeasible in high dimensions. We therefore employ the Hutchinson trace estimator [22], which gives the following approximation:

$$\mathsf{Tr}\left(\Pi\nabla^2 h_i\right) \approx \frac{1}{N}\sum_{k=1}^N (\Pi v_k)^T(\nabla^2 h_i v_k), \quad \mathsf{Tr}\left(\Pi\nabla^2 g_j\right) \approx \frac{1}{N}\sum_{k=1}^N (\Pi v_k)^T(\nabla^2 g_j v_k) \quad (9)$$

for each $i \in [m], j \in I_x$ where $v_k \sim \mathcal{N}(0, I_d)$ are independent standard normal samples. Each Hessian-Vector Product (HVP) $\nabla^2 h_i v_k$ (or $\nabla^2 g_j v_k$) can be computed at a cost similar to one gradient evaluation, so $N$ probes incur only $\mathcal{O}(N \cdot \text{grad-cost})$ computational cost rather than $\mathcal{O}(d \cdot \text{grad-cost})$, saving significant computational cost in high-dimension circumstances. In our experiments, $N = 5$ suffices to achieve low variance estimates that match the performance of the full Hessian computation.

By combining these numerical schemes, we arrive at the full algorithm of OLLA in Algorithm 1.

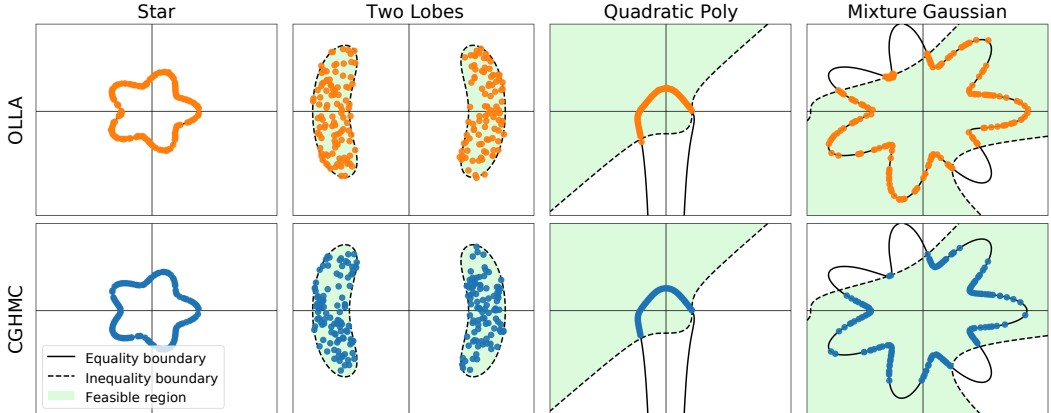

Figure 2: Scatter plots of 200 samples from OLLA (top row) and CGHMC (bottom row) on four 2D synthetic examples. Solid lines show equality constraints, dashed lines show inequality boundaries, and green shaded areas mark feasible region by inequality constraints. OLLA closely matches the CGHMC samples in each scenario.

## 5 Experiments

To demonstrate the sampling accuracy and efficiency, we compare OLLA and its Hutchinson-accelerated variant (OLLA-H) against three standard constrained samplers: CLangevin [20], CHMC [20], and CGHMC [21]. While the three baselines were originally designed for equality constraints, we introduce one slack variable per inequality constraint via $g_j(x) + \frac{1}{2}\delta_j^2 = 0$ for each $j \in [l]$ so that $g_j(x) \leq 0$ is enforced. CGHMC, meanwhile, does not rely on slack variables, but instead uses a Metropolis-Hasting correction step not only to reject samples based on the energy of the proposed samples, but also based on the inequality constraint violation. Therefore, CGHMC allows more accurate and unbiased sampling from the constrained distribution than CHMC and CLangevin in the long run, and we use it as a ground truth for measuring the accuracy of our methods.

### 5.1 Synthetic 2D Examples

We first evaluate sampling on two-dimensional manifolds: (1) star-shaped equality manifold, (2) two-lobe inequality manifold both with uniform density, (3) manifold defined by quadratic-polynomial equality and inequality constraints with a standard Gaussian target, (4) Gaussian mixture with nine components restricted to a seven lobes manifold by a nonlinear inequality (both for mixed scenario).

For each problem, we ran 200 independent chains in parallel with 5,000 steps, and collected the last 200 samples. We then computed the $W_2$ distance and energy distance between the empirical distribution of the samples and the target distribution, as well as the constraint violation defined by $\mathbb{E}[|h(x)|], \mathbb{E}[\max g(x)^+]$, respectively. The results are shown in Figure 2 and Figure 3. Further details of example setups and additional results are included in Appendix H.1.

**Sampling Accuracy & Constraint Violation of OLLA.** As shown in Figure 3, OLLA and OLLA-H match the performance of CHMC and CLangevin in both Wasserstein and energy-distance metrics, demonstrating that our landing-based approach attains sampling accuracy on par with established methods. Also, constraint violations for OLLA and OLLA-H remained at low levels without computationally expensive projection steps.

Table 1: Effect of $\alpha$ on $W_2^2, \mathbb{E}[|h|]$

| $\alpha$ | $W_2^2$ | $\mathbb{E}[|h(x)|]$ |
|---|---|---|
| 1 | $0.363_{\pm 0.064}$ | $0.682_{\pm 0.017}$ |
| 10 | $0.200_{\pm 0.035}$ | $0.130_{\pm 0.001}$ |
| 100 | $0.159_{\pm 0.032}$ | $0.017_{\pm 0.001}$ |
| 200 | $0.121_{\pm 0.019}$ | $0.008_{\pm 0.001}$ |

**Effect of Hyperparameters $\alpha$ and $\epsilon$.** We further examine the influence of the landing rate $\alpha$ and boundary repulsion $\epsilon$ on sampling accuracy and constraint satisfaction. As shown in Tables 1 and 2, increasing $\alpha$ accelerates convergence, yielding smaller $W_2^2$ values and reduced equality constraint violations, consistent with our theoretical prediction that larger $\alpha$ enhances the landing and contraction rates toward $\Sigma$. However, excessively large $\alpha$ leads to numerical instability,

Table 2: Effect of $\epsilon$ on $W_2^2, \mathbb{E}[\max g^+]$

| $\epsilon$ | $W_2^2$ | $\mathbb{E}[\max g^+(x)]$ |
|---|---|---|
| 0.1 | $0.151_{\pm 0.026}$ | $0.082_{\pm 0.017}$ |
| 1 | $0.108_{\pm 0.011}$ | $0.067_{\pm 0.027}$ |
| 5 | $0.123_{\pm 0.018}$ | $0.040_{\pm 0.015}$ |
| 10 | $0.112_{\pm 0.034}$ | $0.019_{\pm 0.006}$ |

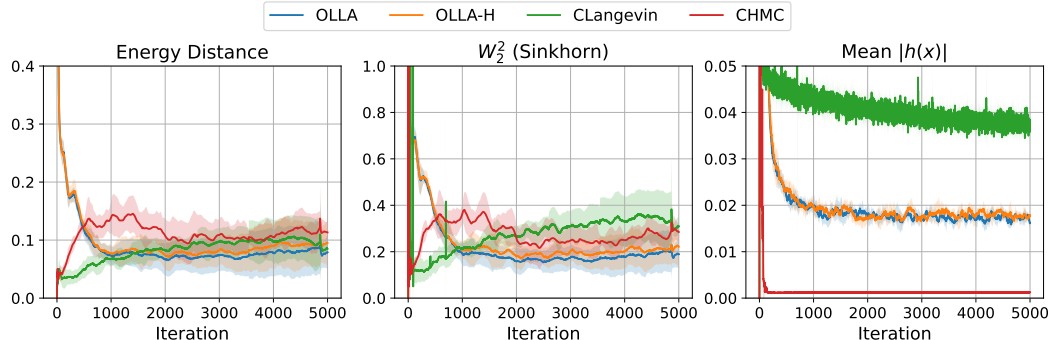

Figure 3: Convergence diagnostics on the Gaussian mixture of 9 components on the 7 lobes manifold with $\alpha = 100, \epsilon = 1$. From left to right: (1) energy distance to CGHMC samples, (2) squared $W_2^2$ distance to CGHMC samples, and (3) mean constraint violation $\mathbb{E}[|h|]$. Solid lines and shaded bands show the mean $\pm 1$ SD over five independent runs. Both OLLA and OLLA-H rapidly decrease $\mathbb{E}[|h|]$ down to small values and maintain it there, which achieving the lowest energy and $W_2^2$ errors.

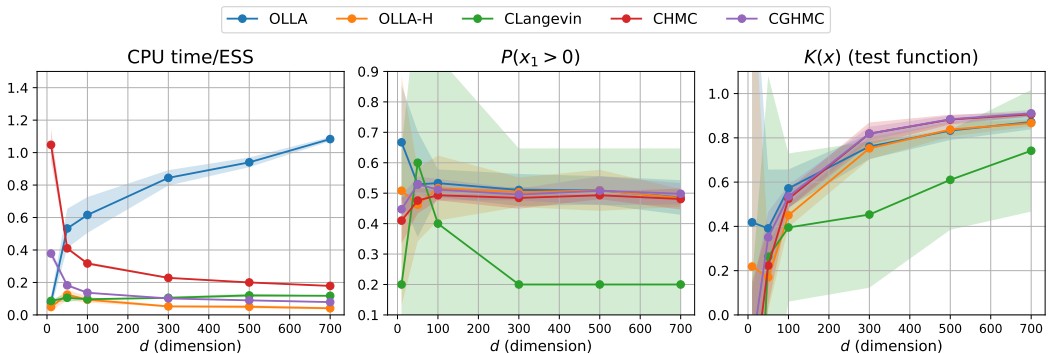

Figure 4: Sampling performance and accuracy as the dimension $d$ increases (with $m = l = 5$). From left to right: (1) CPU time per ESS versus $d$, (2) Estimated probability $P(x_1 > 0)$ versus $d$, (3) Estimated value of $K(x)$ versus $d$. Shaded bands shows $\pm$ 1SD over five runs.

causing $W_2^2$ to rise and the sampler to collapse (Table 7). A similar trend is observed for $\epsilon$. Stronger repulsion lowers inequality violations, but beyond a certain range, $W_2^2$ remains nearly unchanged, aligning with the continuous-time theory that $\epsilon$ primarily affects the finite landing time rather than the asymptotic convergence rate. Overly large $\epsilon$ values can again destabilize the dynamics and lead to numerical breakdown (Table 8).

## 5.2 Scaling of OLLA under High-Dimensionality and Large Number of Constraints

We assess the robustness and scalability of OLLA and OLLA-H using a synthetic *stress-test* problem that enables explicit control over the ambient dimension $d$ and the numbers of equality and inequality constraints $(m, l)$. Samples are drawn from a uniform distribution on a constrained manifold $\Sigma$ defined by linear and quadratic constraints. Each algorithm runs for 1,000 iterations with burn-in and thinning, and we vary one of $d$, $m$, or $l$ while fixing the others to disentangle their individual effects.

We evaluate two key metrics: (1) CPU time per effective sample (CPU/ESS), and (2) the accuracy of representative test function estimates such as $P(x_1 > 0)$ and $K(x) = \sin(x_1)e^{x_2} + \log(|x_3| + 1)\tanh(x_4) + \prod_{i=5}^{9} \cos(x_i)$. Detailed experimental setups are provided in Appendix H.2.

**Scaling under Dimension.** Figure 4 illustrates the sampling performance of algorithms as $d$ increases from 10 to 700 (with $m = l = 5$). On the left, CPU time/ESS of OLLA-H remains essentially flat around 0.05s/sample while OLLA grows linearly (reaching $\approx 1.1$s/sample at $d = 700$), and CHMC and CLangevin stay at $0.2 - 0.3$s/sample and 0.1s/sample respectively. In the center, both OLLA-H, CHMC, and CGHMC maintain $P(x_1 > 0) \approx 0.5$, whereas CLangevin collapses to $\approx 0.2$ in high dimensions, indicating severe bias. On the right, the estimate of nonlinear test function $K(x)$ shows that OLLA, OLLA-H, CHMC, and CGHMC all produce virtually identical estimates

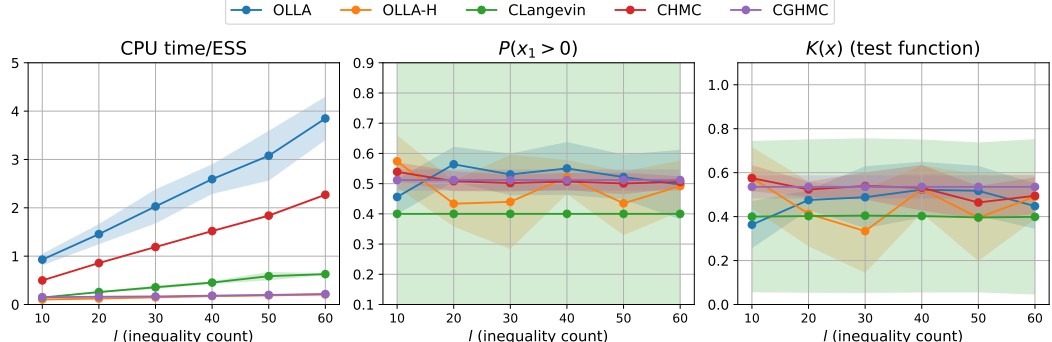

Figure 5: Sampling performance and accuracy as the number of inequality constraints $l$ increases (with $d = 100, m = 5$). From left to right: (1) CPU time per ESS versus $l$, (2) Estimated probability $P(x_1 > 0)$ versus $l$, (3) Estimated value of $K(x)$ versus $l$. Shaded bands shows $\pm$ 1SD over five runs. Note that OLLA and CGHMC results on CPU time/ESS overlap almost perfectly, suggesting their comparable performance on this metric.

even as $d$ grows, while CLangevin lags behind. Overall, the results indicate that OLLA-H scales, maintaining reliable performance as the dimension $d$ increases.

**Scaling under the Number of Constraints.** With $d = 100$, we separately increased the number of equalities $m$ (with $l = 5$) and the number of inequalities $l$ (with $m = 5$). In the presence of a large number of equalities, OLLA and OLLA-H may lose their edge over equality constrained specialized baselines. In these situations, the maximum $\alpha$ that is stable in the discrete algorithm becomes limited to prevent significant discretization error. However, we observe that even, at near limit value of $\alpha$, $\mathbb{E}[|h|]$ may remain relatively high unless the step size $\Delta t$ is further reduced. We add an additional study for this on Appendix H.2. By contrast, when adding more inequalities, OLLA-H stays relatively accurate and shows very low CPU time/ESS (Figure 5).

### 5.3 Molecular System with Realistic Potential

We further evaluate samplers on a molecular system with realistic potentials and geometric constraints. The system incorporates equality constraints $h$ (fixed bond lengths and angles) and inequality constraints $g$ (steric hindrance), alongside torsion and Weeks-Chandler-Anderson potential terms. Experiments were conducted for increasing dimensions $d = 3N_{\text{atom}}$ by varying the number of atoms. Detailed experimental details are provided in Appendix H.3 and Table 3 summarizes the results.

Table 3: Estimates of radius of gyration squared ($R_g^2$) and total CPU time for sampling (in brackets) on the molecular system with realistic potential (complex $\Sigma$, small $|I_x|$, large $N_{\text{newton}}$). The average constraint violation for OLLA-H was below $0.007$ (equality), while projection-based samplers maintained violations below $0.0001$. Inequality violations were observed to be $0$ for all algorithms. The $(d, m, l)$ configurations are: $(15, 7, 6), (30, 17, 36), (45, 27, 91), (60, 37, 171), (90, 57, 406)$.

| method / dim ($d$) | 15 | 30 | 45 | 60 | 90 |
|---|---|---|---|---|---|
| OLLA-H ($N = 0$) | $1.392 \pm 0.026$ (**78**s) | $5.414 \pm 0.047$ (**151**s) | $12.240 \pm 0.102$ (**254**s) | $21.820 \pm 0.098$ (**334**s) | $49.080 \pm 0.223$ (**704**s) |
| OLLA-H ($N = 5$) | $1.370 \pm 0.032$ (182s) | $5.424 \pm 0.045$ (400s) | $12.200 \pm 0.155$ (656s) | $21.780 \pm 0.098$ (916s) | $48.940 \pm 0.273$ (1732s) |
| CLangevin | $1.396 \pm 0.012$ (421s) | $5.526 \pm 0.015$ (3620s) | $12.400 \pm 0.000$ (10940s) | $22.140 \pm 0.049$ (22600s) | $49.960 \pm 0.049$ (52220s) |
| CHMC | $1.410 \pm 0.000$ (147s) | $5.580 \pm 0.000$ (468s) | $12.500 \pm 0.000$ (1012s) | $22.200 \pm 0.000$ (1712s) | $49.960 \pm 0.049$ (3660s) |
| CGHMC | $1.410 \pm 0.000$ (135s) | $5.580 \pm 0.000$ (282s) | $12.500 \pm 0.000$ (467s) | $22.200 \pm 0.000$ (652s) | $49.940 \pm 0.049$ (1116s) |

As dimension $d$ grows, the feasible set $\Sigma$ becomes increasingly complex, while the number of active inequality constraints $|I_x|$ remains small. However, the numerous equality constraints make each projection step demanding for projection-based methods (CHMC, CLangevin, CGHMC), resulting in substantial Newton iteration $N_{\text{newton}}$ and increased computational costs. Particularly, CLangevin is significantly affected from this, whereas CGHMC is faster but still slower than OLLA-H ($N = 0$).

In contrast, increasing the Hutchinson probe $N$ from $N = 0$ to $N = 5$ improves the mean equality constraint violation across all dimensions $d$, reducing it from a range of $\approx 0.0055$ down to $\approx 0.00035$. Despite this, both OLLA-H configurations ($N = 0, N = 5$) yield comparable test function estimation accuracy. This suggests that the constraint violation at $N = 0$ was already sufficiently low, rendering the improvement's impact on the final sampling accuracy negligible. Consequently, the $N = 0$ variant—where HVP evaluations are completely skipped—achieves significantly lower computational cost. This behavior aligns with findings in Zhang et al. [17], suggesting that omitting the Itô-Stratonovich (mean curvature) correction term has negligible practical impact on sampling accuracy, while markedly improving efficiency.

### 5.4  Bayesian Logistic Regression with Fairness and Monotonicity Constraints

We evaluate the samplers on a high-dimensional Bayesian logistic regression task using a two-layer neural network trained on the German Credit dataset [41]. The setup enforces fairness through equality constraints $h$ ensuring parity in true positive rate and false positive rate between demographic groups, along with monotonicity constraints $g$ on selected input data. Further details of experimental setup are provided in Appendix H.3.

This problem poses significant challenges for projection-based samplers. Step sizes effective in unconstrained scenarios led to projection failures for CLangevin and CHMC, and to acceptance rates below $5\%$ for CGHMC. To maintain stability, their step sizes were substantially reduced. In contrast, the projection-free OLLA-H remained stable across all settings and, for fairness, was also evaluated with the same reduced step size as CLangevin.

As summarized in Table 4, OLLA-H ($N = 0$) consistently attains the lowest test negative log-likelihood (NLL) while being orders of magnitude faster than projection-based baselines. Although the projection-based methods achieve tighter feasibility, OLLA-H maintains small violations without noticeable degradation in accuracy. These results highlight OLLA-H's robustness and computational advantage in high-dimensional constrained Bayesian logistic regression task.

Table 4: Test NLL and total CPU time for sampling (in brackets) on the Bayesian logistic regression with fairness and monotonicity constraints (high-dimensional $\Sigma$). The average constraint violations for OLLA-H were below $0.005$ (equality) and $0.15$ (inequality), while projection-based samplers (CLangevin, CHMC, CGHMC) maintained feasibility below $0.0008$ (equality) and no inequality violation.

| method / dim ($d$) | 1986 | 4994 | 9986 | 49986 | 100002 |
|---|---|---|---|---|---|
| OLLA-H ($N = 0$) | $0.514 \pm 0.013$ | $0.521 \pm 0.008$ | $0.524 \pm 0.014$ | $0.523 \pm 0.011$ | $0.520 \pm 0.015$ |
| | (**63**s) | (**70**s) | (**70**s) | (**81**s) | (**82**s) |
| OLLA-H ($N = 5$) | $0.520 \pm 0.013$ | $0.524 \pm 0.008$ | $0.505 \pm 0.004$ | $0.517 \pm 0.011$ | $0.516 \pm 0.004$ |
| | (159s) | (180s) | (205s) | (189s) | (197s) |
| CLangevin | $0.573 \pm 0.004$ | $0.568 \pm 0.013$ | $0.564 \pm 0.022$ | $0.580 \pm 0.005$ | $0.570 \pm 0.011$ |
| | (1162s) | (1176s) | (1194s) | (1428s) | (1370s) |
| CHMC | $0.599 \pm 0.015$ | $0.595 \pm 0.020$ | $0.599 \pm 0.017$ | $0.606 \pm 0.004$ | $0.605 \pm 0.004$ |
| | (526s) | (532s) | (561s) | (586s) | (611s) |
| CGHMC | $0.600 \pm 0.007$ | $0.600 \pm 0.009$ | $0.606 \pm 0.003$ | $0.598 \pm 0.020$ | $0.601 \pm 0.007$ |
| | (76s) | (77s) | (82s) | (83s) | (88s) |

## 6  Conclusion & Future works

We have presented Overdamped Langevin with Landing (OLLA), a projection-free SDE sampler that enforces nonlinear equality and inequality constraints by deterministically "landing" trajectories onto the feasible set while retaining full tangential noise, and proved that its continuous dynamics converge exponentially fast in 2-Wasserstein distance under appropriate regularity. Building on this, we proposed OLLA-H, an EM discretization that uses a Hutchinson trace estimator for approximating the Itô–Stratonovich correction at only $\mathcal{O}(N \cdot \text{grad-cost})$ per step, and showed in both 2D and high-dimensional tests that it matches the accuracy of established constrained samplers while drastically reducing runtime. Future work will include non-asymptotic convergence guarantees for the discrete algorithm—closing the gap between SDE theory and implementation—and developing OLLA variants that remain stable even with many equality constraints in very high dimensions, further extending its scope to large-scale constrained probabilistic inference.

## Acknowledgements

KJ and MT are grateful for partial supports by NSF Grants DMS-1847802, DMS-2513699, DOE Grants NA0004261, SC0026274, and Richard Duke Fellowship. MM is supported by the German Research Foundation. The authors thank Andre Wibisono for insightful discussion.

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

# A    Table of Key Notations, Assumptions, and Remarks

Table 5: Table of Key Notations

| Symbol | Definition | Descriptions |
|---|---|---|
| $h$ | $h(x) = [h_1(x), \ldots, h_m(x)]^T$ | Equality constraints |
| $g$ | $g(x) = [g_1(x), \ldots, g_n(x)]^T$ | Inequality constraints |
| $\Sigma$ | $\{x \in \mathbb{R}^d \mid h(x) = 0, g(x) \leq 0\}$ | Constraint manifold |
| $I_x$ | $\{i \in [l] \mid g_i(x) \geq 0\} = \{i_1, .., i_{\lvert I_x \rvert}\}$ | Active index set of inequalities |
| $g_{I_x}$ | $g_{I_x}(x) = [g_{i_1}(x), ...g_{i_{\lvert I_x \rvert}}(x)]^T$ | Active inequality constraints |
| $J(x)$ | $\{h(x)^T, g_{i_1}(x) + \epsilon, ..., g_{i_{\lvert I_x \rvert}}(x) + \epsilon\}^T$ | Constraint-correction vector |
| $\Pi(x)$ | $I - \nabla J(x)^T G(x)^{-1} \nabla J(x)$ | Orthogonal projector onto $T_x \Sigma$ |
| $T_x \Sigma$ | $\{v \in \mathbb{R}^d \mid \nabla h(x)v = 0, \nabla g_{I_x}(x)v = 0\}$ | Tangent space of $\Sigma$ at $x$ |
| $\nabla_\Sigma f$ | $\Pi(x) \nabla f(x)$ | Intrinsic gradient on $\Sigma$ (C) |
| $\mathrm{div}_\Sigma X$ | $\mathrm{Tr}\left(\Pi(x) \nabla X(x)\right)$ | Intrinsic divergence on $\Sigma$ (C) |
| $d\sigma_\Sigma$ | Induced surface (Hausdorff) measure of $\Sigma$ | Surface measure on $\Sigma$ |
| $G(x)$ | $\nabla J(x) \nabla J(x)^T$ | Gram matrix, full rank assumed |
| $U_\epsilon(\Sigma)$ | Tubular neighborhood of $\Sigma$ with reach $\epsilon$ | Usual tubular neighborhood |
| $\hat{U}_\delta(\Sigma)$ | Recoverable tubular neighborhood with width $\delta$ | See details in (C.1, C.2) |
| $M_h$ | $\sup_{x_0 \in \mathrm{supp}(\rho_0)} \|h(x_0)\|_2$ | Initial bound of $h(x)$ |
| $M_g$ | $\sup_{x_0 \in \mathrm{supp}(\rho_0)} \|g(x_0)\|_2$ | Initial bound of $g(x)$ |
| $\pi(x)$ | $\arg\min_{y \in \Sigma} \|x - y\|_2$ | Nearest point projection onto $\Sigma$ |
| $\epsilon$ | Boundary repulsion rate | Controls effect of repulsion. |
| $\alpha$ | Landing rate | Controls constraint decay |
| $\lambda_{\mathrm{LSI}}$ | Log–Sobolev constant of $\rho_\Sigma$ on $\Sigma$ | Enable $\mathsf{KL}^\Sigma \leq \frac{1}{2\lambda_{\mathrm{LSI}}} I^\Sigma$ |
| $\rho_t$ | Density of the process $X_t$ at time $t$ | Law of $X_t$ (following OLLA) |
| $\tilde{\rho}_t$ | Density of the projected process $Y_t$ at time $t$ | Law of $Y_t := \pi(X_t)$ |
| $\rho_\Sigma$ | Target (stationary) density on $\Sigma$ | Proportional to $\exp(-f)d\sigma_\Sigma$ |
| $\mathsf{KL}^\Sigma(\rho\|\pi)$ | $\int_\Sigma \rho \ln \frac{\rho}{\pi} d\sigma_\Sigma$ | KL-divergence on $\Sigma$ |
| $I^\Sigma(\rho\|\pi)$ | $\int_\Sigma \rho \|\nabla_\Sigma \ln \frac{\rho}{\pi}\|_2^2 d\sigma_\Sigma$ | Fisher information on $\Sigma$ |
| $\kappa$ | Regularity constant of $\Sigma$ | See details in (C.1, C.3) |
| $\Sigma_p$ | $\Sigma_p := \pi(\{x \in \mathbb{R}^d \mid h(x) = p, g(x) \leq 0\})$ | Projected manifold for $\|p\|_2 \leq \delta$ |
| $v_p^b$ | Boundary velocity of $\partial \Sigma_p$ | See details in Theorem 3 |

**Assumption 1** (Assumptions for exponential convergence of OLLA). The followings are assumptions used for proving the convergence results of OLLA:

(C1) **LICQ**. The linear independence constraint qualification holds on $\Sigma$ if, for every $x \in \Sigma$,

$$\{\nabla h_i(x)\}_{i=1}^m \cup \{\nabla g_j(x)\}_{j \in I_x}$$

is a set of linearly independent vectors.

(C2) **Manifold regularity**. The Riemannian manifold $\Sigma$ is compact and connected. Also, the equality constraint function $h(x)$ is coercive, i.e., $\|h(x)\|_2 \to \infty$ as $\|x\|_2 \to \infty$. Assume $\nabla h(x) \neq 0$, $\forall x \in \Sigma$ and $\dim(\Sigma) = d$ when there are only inequality constraints.

(C3) **Initial constraints bounds**. The initial distribution $\rho_0$ satisfies

$$M_h := \sup_{x_0 \in \text{supp}(\rho_0)} \|h(x_0)\|_2 < \infty, \quad M_g := \sup_{x_0 \in \text{supp}(\rho_0)} \|g(x_0)\|_2 < \infty.$$

(C4) **Log-Sobolev constant**. The stationary distribution $\rho_\Sigma(x) \propto \exp(-f(x))d\sigma_\Sigma$ satisfies a Log-Sobolev Inequality (LSI) with constant $\lambda_{\text{LSI}}$ so that

$$\mathsf{KL}^\Sigma(\rho\|\rho_\Sigma) \leq \frac{1}{2\lambda_{\text{LSI}}} I^\Sigma(\rho\|\rho_\Sigma).$$

*Note: Compactness of $\Sigma$ with non-negativity of Ricci curvature guarantees the LSI condition. See [42–47] or Remark 6 for detailed description.*

(M1) **Regularity of $\Sigma_p$**. The projected manifold $\Sigma_p := \pi(\{x \in \mathbb{R}^d \mid h(x) = p, g(x) \leq 0\})$ lies inside the interior of $\Sigma$ for $0 < \|p\|_2 < \delta$, where $\delta$ is the width of recoverable tubular neighborhood $\hat{U}_\delta(\Sigma)$.

(M2) **Regularity of $\partial\Sigma_p$**. The boundary velocity $v_p^b$ of $\partial\Sigma_p$ appearing on Leibniz integral rule satisfies $\sup_{x \in \partial\Sigma_p} \|v_p^b\|_2 \leq V\|p\|_2^\beta$ for some $V > 0, \beta > 0$. Also, assume $M_\Sigma := \sup_{\|p\|_2 < \delta} \sigma_{\partial\Sigma_p}(\partial\Sigma_p) < \infty$.

(M3) **Bound on $\rho_t, \rho_\Sigma$**. There exists the constants $G_1, G_2, G_3 > 0$ such that $G_1 := \sup_{t \geq 0, x \in \Sigma} \tilde{\rho}_t < \infty$ and $0 < G_2 \leq \rho_\Sigma \leq G_3$ for every $x \in \Sigma$.

**Remark 3** (Comments on the assumptions (M1) and (M3)).

• Although the assumption (M1) is stated for all small perturbation $p \in \mathbb{R}^m, 0 < \|p\|_2 < \delta$, in our anaylsis on Theorem 3, we only require:

$$\Sigma_t := \pi(\{x \in \mathbb{R}^d \mid h(x) = h(X_0)e^{-\alpha t}, g(x) \leq 0\}) \subset \text{int}(\Sigma)$$

for $t \geq t_{\text{cut}}$, $t_{\text{cut}} := \max\left\{\frac{1}{\alpha}\ln\left(\frac{M_g + \epsilon}{\epsilon}\right), \frac{1}{\alpha}\ln\left(\frac{M_h}{\delta}\right), \frac{1}{\alpha}\ln(\tilde{C}_5)\right\}$. Hence, when $X_0 \sim \delta(x_0)$ for some $x_0 \in \mathbb{R}^d$, we can replace the global requirement "for all $p$ with $0 < \|p\|_2 < \delta$" by a weaker, one-dimensional condition:

(M1′) If $X_0 \sim \delta(x_0)$ for some $x_0 \in \mathbb{R}^d$, let $u = \frac{h(x_0)}{\|h(x_0)\|_2}$ and assume

$$\Sigma_{su} := \pi(\{x \in \mathbb{R}^d \mid h(x) = su, g(x) \leq 0\}) \subset \text{int}(\Sigma), \quad \text{for all } s \in (0, \delta).$$

• For the uniform boundedness $\sup_{t \geq 0, x \in \Sigma} \tilde{\rho}_t < \infty$ in the assumption (M3), we recall from the proof of Lemma G.2 that $\tilde{\rho}_t$ satisfies the following Fokker-Planck equation:

$$\partial_t \tilde{\rho}_t = -\text{div}_\Sigma(\tilde{\rho}_t(\nabla_\Sigma \ln \rho_\Sigma + b_N)) + \sum_{k=1}^d \text{div}_\Sigma(\text{div}_\Sigma(\tilde{\rho}_t(f_k + \delta_k))(f_k + \delta_k)).$$

Since $\|\delta_k\|_2 = \mathcal{O}(e^{-\alpha t})$, the second-order differential operator of the Fokker-Planck equation becomes uniformly elliptic for sufficiently large $t$. In the absense of boundary conditions (equality-only case), the result from Saloff-Coste [48] would then guarantee $\sup_{t \geq 0, x \in \Sigma} \tilde{\rho}_t < \infty$. However,

the non-standard boundary conditions imposed on $\Sigma$ in our setting preclude a direct application of those results. We therefore retain the uniform boundedness of $\tilde{\rho}_t$ as an explicit assumption.

**Remark 4** (Relaxed assumption of **(M1)**). We remark that the assumption **(M1)** can be strong in practice. To mitigate this issue, we propose the following assumption **(M1″)** which is a milder assumption than **(M1)**. It assumes

(M1″) Let $t_1 := \max\left\{\frac{1}{\alpha}\ln\left(\frac{M_g+\epsilon}{\epsilon}\right)\right), \frac{1}{\alpha}\ln\left(\frac{M_h}{\delta}\right)\right\}$ and define $p_t := \mathbb{P}(\pi(X_t) \in \partial\Sigma)$ for $t \geq t_1$. Suppose there exist $\gamma_p, C_p > 0, t_p \geq t_1$ such that $p_t, \partial_t p_t \leq C_p e^{-\gamma_p t} \leq 1/2$ for $t \geq t_p$.

We note that this assumption is a necessary condition of the previous assumption (M1), and the landing property guarantees $\lim_{t\to\infty} p_t = 0$. Also, this assumption ensures that both $p_t, \partial_t p_t$ decays exponentially fast, which can be satisfied depending on how fast the sticking behavior of $\pi(X_t)$ at the bounadry is attenuated as $t$ increases.

> **Corollary A.1** (Convergence result for mixed-constrained OLLA with milder assumption). *Suppose assumptions (C1) to (C4) and (M1″) to (M3) hold. Define $X_t$ to be the stochastic process following OLLA dynamics (3) and $\tilde{\rho}_t$ to be the law of $Y_t := \pi(X_t)$ after $t \geq t_{cut}, t_{cut} := \max\left\{\frac{1}{\alpha}\ln\left(\frac{M_g+\epsilon}{\epsilon}\right), \frac{1}{\alpha}\ln\left(\frac{M_h}{\delta}\right), \frac{1}{\alpha}\ln(\tilde{C}_5), t_p\right\}$. Then, the following non-asymptotic convergence rate of $W_2(\rho_t, \rho_\Sigma)$ holds for $\alpha > 2\lambda_{LSI}, \beta \geq 1$, and $t > t_{cut}$:*
>
> $$W_2(\rho_t, \rho_\Sigma) \leq \frac{M_h}{\kappa}e^{-\alpha t} + \sqrt{\frac{2}{\lambda_{LSI}}\mathsf{KL}^\Sigma(\tilde{\rho}_t^{\Sigma^\circ}||\rho_\Sigma) + C_p e^{-\gamma_p t} \cdot diam_\Sigma(\Sigma)^2}$$
>
> *where*
>
> $$\mathsf{KL}^\Sigma(\tilde{\rho}_t^{\Sigma^\circ}||\rho_\Sigma) \leq \exp\left(-2\lambda_{LSI}(t - t_{cut}) - \frac{2\lambda_{LSI}\tilde{C}_5}{\alpha}(e^{-\alpha t} - e^{-\alpha t_{cut}}) - \frac{C_p}{\gamma_p}\left(e^{-\gamma_p t} - e^{-\gamma_p t_{cut}}\right)\right) \times$$
> $$[\mathsf{KL}^\Sigma(\tilde{\rho}_{t_{cut}}^{\Sigma^\circ}||\rho_\Sigma) + \tilde{C}_7^{\Sigma^\circ} + \tilde{C}_8^{\Sigma^\circ}]$$
>
> *for some constants $G_4^{\Sigma^\circ}, G_5, G_6^{\Sigma^\circ}, \tilde{C}_6 > 0$, $\tilde{C}_7^{\Sigma^\circ} := (\tilde{C}_6 + \alpha G_4^{\Sigma^\circ} G_5 M_h)\frac{e^{-\alpha t_{cut}}}{\alpha - 2\lambda_{LSI}}$,*
>
> $$\tilde{C}_5 = \mathcal{O}\left(1 + \frac{\tilde{C}_{L_A}M_h}{\kappa} + \left(\frac{\tilde{C}_{L_A}M_h}{\kappa}\right)^2\right), \quad \tilde{C}_8^{\Sigma^\circ} := (G_6^{\Sigma^\circ}VM_h^\beta)\frac{e^{-\alpha\beta t_{cut}}}{\alpha\beta - 2\lambda_{LSI}},$$
>
> *and with $\tilde{C}_{L_A}$ being the Lipschitz constant of $\nabla\pi(x)\Pi(x)$ on $\hat{U}_\delta(\Sigma)$, $\tilde{\rho}_t^{\Sigma^\circ}$ being the conditional laws of $Y_t$ given $Y_t \in int(\Sigma)$.*

*Proof.* We slightly adjust the argument in the proof of Theorem 3 to show this statement, and the definitions of constants are shared with Theorem 3. We first observe that the probability measure $\tilde{\rho}_t$ can be decomposed as follows:

$$\tilde{\rho}_t = (1 - p_t)\tilde{\rho}_t^{\Sigma^\circ} + p_t\tilde{\rho}_t^{\partial\Sigma}, \quad t \geq t_{cut}$$

where $p_t := \tilde{\rho}_t(\partial\Sigma)$ and

$$\tilde{\rho}_t^{\Sigma^\circ} := \frac{\tilde{\rho}_t|_{\Sigma^\circ}}{1 - p_t}, \quad \tilde{\rho}_t^{\partial\Sigma} := \frac{\tilde{\rho}_t|_{\partial\Sigma}}{p_t},$$

are the conditional laws of $\tilde{\rho}_t$ restricted on $\Sigma$ and $\partial\Sigma$, respectively.

Because $\tilde{\rho}_t$ is the convex combination of $\tilde{\rho}_t^{\Sigma^\circ}$ and $\tilde{\rho}_t^{\partial\Sigma}$, the convexity of the 2-Wasserstein distance implies

$$W_2^\Sigma(\tilde{\rho}_t, \rho_\Sigma)^2 \leq (1 - p_t)W_2^\Sigma(\tilde{\rho}_t^{\Sigma^\circ}, \rho_\Sigma)^2 + p_t W_2^\Sigma(\tilde{\rho}_t^{\partial\Sigma}, \rho_\Sigma)^2.$$

Here, we note that

$$W_2^\Sigma(\tilde{\rho}_t^{\Sigma^\circ}, \rho_\Sigma) \leq \sqrt{\frac{2}{\lambda_{LSI}}\mathsf{KL}^\Sigma(\tilde{\rho}_t^{\Sigma^\circ}||\rho_\Sigma)}$$

by Lemma E.6 and the fact that $\tilde{\rho}_t^{\Sigma^\circ}$ has its support on $\mathrm{int}(\Sigma)$. Also, we note that

$$\partial_t \tilde{\rho}_t^{\Sigma^\circ} = \frac{1}{1-p_t}\left(\partial_t \tilde{\rho}_t + (\partial_t p_t)\tilde{\rho}_t^{\Sigma^\circ}\right)$$

because $\tilde{\rho}_t = (1-p_t)\tilde{\rho}_t^{\Sigma^\circ}$ holds for $x \in \mathrm{int}(\Sigma)$. Then, the same approach used in Lemma G.2 gives

$$\partial_t \mathsf{KL}^\Sigma(\tilde{\rho}_t^{\Sigma^\circ}||\rho_\Sigma) = \partial_t \int_{\Sigma_t} \tilde{\rho}_t^{\Sigma^\circ} \ln\left(\frac{\tilde{\rho}_t^{\Sigma^\circ}}{\rho_\Sigma}\right) d\sigma_\Sigma$$

$$= \underbrace{\frac{1}{1-p_t}\int_{\Sigma_t} \partial_t \tilde{\rho}_t \ln\left(\frac{\tilde{\rho}_t^{\Sigma^\circ}}{\rho_\Sigma}\right) d\sigma_\Sigma}_{\text{Term (1)}} + \underbrace{\int_{\partial\Sigma_t} \tilde{\rho}_t^{\Sigma^\circ}\left[\ln\left(\frac{\tilde{\rho}_t^{\Sigma^\circ}}{\rho_\Sigma}\right) - 1\right]\langle v_t^b, n_t\rangle d\sigma_{\partial\Sigma_t}}_{\text{Term (2)}}$$

$$+ \underbrace{\frac{\partial_t p_t}{1-p_t}\mathsf{KL}^\Sigma\left(\tilde{\rho}_t^{\Sigma^\circ}||\rho_\Sigma\right)}_{\text{Term (3)}} + \underbrace{\partial_t \int_{\Sigma_t} \tilde{\rho}_t^{\Sigma^\circ} d\sigma_\Sigma}_{=0}$$

where the last equality holds from the Leibniz integral rule with $v_t^b$ being the velocity vector of the boundary of $\Sigma_t := \pi(\{x \in \mathbb{R}^d \mid h(x) = h(X_0)e^{-\alpha t}, g(x) \leq 0\})$. Therefore, the expression of $\partial_t \tilde{\rho}_t$ implies that Term (1) becomes

$$\text{Term (1)} = \underbrace{\int_{\Sigma_t}\left[\tilde{\rho}_t^{\Sigma^\circ}(\nabla_\Sigma \ln\rho_\Sigma + b_N) - \sum_{k=1}^d \mathsf{div}_\Sigma(\tilde{\rho}_t^{\Sigma^\circ}(f_k+\delta_k))(f_k+\delta_k)\right]\nabla_\Sigma \ln\left(\frac{\tilde{\rho}_t^{\Sigma^\circ}}{\rho_\Sigma}\right) d\sigma_\Sigma}_{\text{Term (1-1)}}$$

$$- \underbrace{\int_{\partial\Sigma_t}\left\langle\left[\tilde{\rho}_t^{\Sigma^\circ}(\nabla_\Sigma \ln\rho_\Sigma + b_N) - \sum_{k=1}^d \mathsf{div}_\Sigma(\tilde{\rho}_t^{\Sigma^\circ}(f_k+\delta_k))(f_k+\delta_k)\right]\ln\left(\frac{\tilde{\rho}_t^{\Sigma^\circ}}{\rho_\Sigma}\right), n_t\right\rangle d\sigma_{\partial\Sigma_t}}_{\text{Term (1-2)}}$$

using $\tilde{\rho}_t = (1-p_t)\tilde{\rho}_t^{\Sigma^\circ}$. Therefore, Lemma G.2 implies Term (1) can be bounded by

$$|\text{Term (1)}| \leq -(1-\tilde{C}_5 e^{-\alpha t})I^\Sigma(\tilde{\rho}_t^{\Sigma^\circ}||\rho_\Sigma) + \tilde{C}_6 e^{-\alpha t} + \alpha G_4^{\Sigma^\circ} G_5 M_h e^{-\alpha t}$$

with $G_4^{\Sigma^\circ} := G_3 M_\Sigma \max\left\{\frac{1}{e}, \left|\frac{G_1^{\Sigma^\circ}}{G_2}\ln\left(\frac{G_1^{\Sigma^\circ}}{G_2}\right)\right|\right\}$ and $G_1^{\Sigma^\circ} := \sup_{t \geq t_{cut}, x \in \Sigma}\tilde{\rho}_t^{\Sigma^\circ} \leq 2G_1$, and the Term (2) can be bounded by

$$|\text{Term (2)}| \leq G_6^{\Sigma^\circ} V M_h^\beta e^{-\alpha\beta t}.$$

with $G_6^{\Sigma^\circ} := G_4^{\Sigma^\circ} + G_1^{\Sigma^\circ} M_\Sigma$. Also, Term (3) is upper bounded by

$$\text{Term (3)} \leq C_p e^{-\gamma_p t}\mathsf{KL}^\Sigma(\tilde{\rho}_t^{\Sigma^\circ}||\rho_\Sigma)$$

Therefore, combining the bounds with the LSI condition gives the following inequality:

$$\partial_t\mathsf{KL}^\Sigma(\tilde{\rho}_t^{\Sigma^\circ}||\rho_\Sigma) \leq -2\lambda_{\text{LSI}}\left(1 - \tilde{C}_5 e^{-\alpha t} - \frac{C_p}{2\lambda_{\text{LSI}}}e^{-\gamma_p t}\right)\mathsf{KL}^\Sigma(\tilde{\rho}_t^{\Sigma^\circ}||\rho_\Sigma) + \left(\tilde{C}_6 + \alpha G_4^{\Sigma^\circ} G_5 M_h\right)e^{-\alpha t}$$

$$+ G_6^{\Sigma^\circ} V M_h^\beta e^{-\alpha\beta t}.$$

Hence, applying the Grönwall-type inequality recovers the following inequality:

$$\mathsf{KL}^\Sigma(\tilde{\rho}_t^{\Sigma^\circ}||\rho_\Sigma) \leq \exp\left(-2\lambda_{\text{LSI}}(t - t_{\text{cut}}) - \frac{2\lambda_{\text{LSI}}\tilde{C}_5}{\alpha}(e^{-\alpha t} - e^{-\alpha t_{cut}}) - \frac{C_p}{\gamma_p}\left(e^{-\gamma_p t} - e^{-\gamma_p t_{cut}}\right)\right) \times$$

$$[\mathsf{KL}^\Sigma(\tilde{\rho}_{t_{\text{cut}}}^{\Sigma^\circ}||\rho_\Sigma) + \int_{t_{cut}}^t \exp\left(2\lambda_{\text{LSI}}(s - t_{\text{cut}}) + \frac{2\lambda_{\text{LSI}}\tilde{C}_5}{\alpha}(e^{-\alpha s} - e^{-\alpha t_{cut}}) + \frac{C_p}{\gamma_p}\left(e^{-\gamma_p s} - e^{-\gamma_p t_{cut}}\right)\right) \times$$

$$\left[\left(\tilde{C}_6 + \alpha G_4^{\Sigma^\circ} G_5 M_h\right)e^{-\alpha s} + G_6^{\Sigma^\circ} V M_h^\beta e^{-\alpha\beta s}\right] ds]$$

which can again be summarized as follows:

$$\mathsf{KL}^\Sigma(\tilde\rho_t^{\Sigma^\circ}||\rho_\Sigma) \le \exp\left(-2\lambda_{\mathrm{LSI}}(t - t_{\mathrm{cut}}) - \frac{2\lambda_{\mathrm{LSI}}\tilde C_5}{\alpha}(e^{-\alpha t} - e^{-\alpha t_{\mathrm{cut}}}) - \frac{C_p}{\gamma_p}\left(e^{-\gamma_p t} - e^{-\gamma_p t_{\mathrm{cut}}}\right)\right) \times$$

$$[\mathsf{KL}^\Sigma(\tilde\rho_{t_{\mathrm{cut}}}^{\Sigma^\circ}||\rho_\Sigma) + \tilde C_7^{\Sigma^\circ} + \tilde C_8^{\Sigma^\circ}]$$

where $\tilde C_7^{\Sigma^\circ} := (\tilde C_6 + \alpha G_4^{\Sigma^\circ} G_5 M_h)\frac{e^{-\alpha t_{\mathrm{cut}}}}{\alpha - 2\lambda_{\mathrm{LSI}}}$ and $\tilde C_8^{\Sigma^\circ} := (G_6^{\Sigma^\circ} V M_h^\beta)\frac{e^{-\alpha\beta t_{\mathrm{cut}}}}{\alpha\beta - 2\lambda_{\mathrm{LSI}}}$.

Lastly, we observe that

$$W_2^\Sigma(\tilde\rho_t^{\partial\Sigma}, \rho_\Sigma) := \left(\inf_{\pi \in \Pi(\tilde\rho_t^{\partial\Sigma}, \rho_\Sigma)} \int_{\Sigma\times\Sigma} d_\Sigma(p,q)^2 \pi(dp, dq)\right)^{\frac12} \le \mathrm{diam}_\Sigma(\Sigma)$$

where $\mathrm{diam}_\Sigma(\Sigma) := \sup\{d_\Sigma(x,y) \mid x, y \in \Sigma\} < \infty$ due to the compactness of $\Sigma$. Therefore, combining the above upper bounds, we recover the following inequality:

$$W_2^\Sigma(\tilde\rho_t, \rho_\Sigma) \le \sqrt{\frac{2}{\lambda_{\mathrm{LSI}}}\mathsf{KL}^\Sigma(\tilde\rho_t^{\Sigma^\circ}||\rho_\Sigma) + C_p e^{-\gamma_p} \cdot \mathrm{diam}_\Sigma(\Sigma)^2)}$$

Therefore, applying the same reasoning as in Theorem 3 completes the proof. $\qquad\square$

**Remark 5** (Effect of number of Hutchinson probes $N$ on the decaying speed of constraint functions)**.** In this remark, under the single equality case, we demonstrate that exponential decay of $h$ via OLLA-H and analyze the effects of the number of Hutchinson probes $N$. When there is only a single equality constraint, we recall that the OLLA-H update rule is given as

$$X_{k+1} = X_k + (b(X_k) + \epsilon_k(X_k))\,\Delta t + \sqrt{2\Delta t}\Pi(X_k)\xi_k, \quad \xi_k \sim \mathcal{N}(0, I_d)$$

where $b(x) := -\Pi(x)\nabla f(x) - \alpha\nabla h(x)G(x)^{-1}h(x) - \nabla h(x)G^{-1}(x)\mathsf{Tr}\left(\Pi(x)\nabla^2 h(x)\right)$ and

$$\hat S(x) = \frac{1}{N}\sum_{i=1}^N v_{k,i}^T\Pi(x)\nabla^2 h(x)v_{k,i}, \quad \epsilon_k(x) := -\nabla h(x)G(x)^{-1}\left(\hat S(x) - \mathsf{Tr}\left(S(x)\right)\right)$$

with $S(x) := \Pi(x)\nabla^2 h(x)$ and $v_{k,i} \sim \mathcal{N}(0, I_d)$. Also, we note that $\mathbb{E}\left[\epsilon_k(X_k) \mid X_k\right] = 0$ and

$$\mathbb{E}\left[\epsilon_k(X_k)\epsilon_k(X_k)^T \mid X_k\right] = \frac{\nabla h(X_k)\nabla h(X_k)^T}{\|\nabla h(X_k)\|^4} \cdot \mathrm{Var}(\hat S(X_k)|X_k) = \frac{2\nabla h(X_k)\nabla h(X_k)^T}{N\|\nabla h(X_k)\|^4}\|S(X_k)\|_F^2.$$

**Proposition A.1** (Exponential decay of constraint function under OLLA-H)**.** *. Assume the single equality constraint scenario and the following conditions:*

- *(Dissipativity of $b(x)$)* $\quad \langle x, b(x)\rangle \le -m\|x\|^2 + c$ *for some $m > 0, c \ge 0$.*

- *(Linear growth of $b(x)$)* $\quad \|b(x)\|^2 \le A + B\|x\|^2$ *for some $A, B \ge 0$.*

- *(Boundedness of tangential Hessian-gradient ratio)* $\quad C_h = \sup_{x\in\mathbb{R}^d}\frac{\|\Pi(x)\nabla^2 h(x)\|_F^2}{\|\nabla h(x)\|^2} < \infty.$

- *($L_1$-smoothness of $h$)* $\quad$ *For some $L_1 > 0$, $\|\nabla^2 h(x)\| \le L_1$ for any $x \in \mathbb{R}^d$.*

- *($L_2$-Lipschitzness of $\nabla^2 h$)* $\quad$ *For some $L_2 > 0$, $\|\nabla^2 h(x) - \nabla^2 h(y)\| \le L_2\|x - y\|$ for any $x \in \mathbb{R}^d$.*

- *(Step size control)* $\quad 0 < \Delta t < \min\left\{1, \frac{m}{B}, \frac{1}{2m}\right\}.$

*Then, the OLLA-H dynamics with $N$ Hutchinson probes have the following decaying property of $h$:*

$$\mathbb{E}\left[h(X_K)\right] \le (1 - \alpha\Delta t)^K \mathbb{E}\left[h(X_0)\right] + \mathcal{O}\left(\frac{\Delta t}{\alpha}\left(d + \frac{1}{N}\right)\right)$$

*for $K \ge 0$.*

*Proof.* We first prove that $\sup_{k\geq 0} \mathbb{E}\|\Delta X_k\|^2 < \infty$ where $\Delta X_k := X_{k+1} - X_k$. To show this, we define $\mathcal{F}_k := \sigma(X_m, \xi_m, v_{m,j} \mid m \leq k, j \leq N)$ to be the $k$ th canonical filtration and observe that

$$
\begin{aligned}
\mathbb{E}\left[\|X_{k+1}\|^2 \mid \mathcal{F}_k\right] = \|X_k\|^2 &+ 2\Delta t\langle X_k, b(X_k)\rangle + \Delta t^2\|b(X_k)\|^2 + \Delta t^2 \mathbb{E}\left[\|\epsilon_k(X_k)\|^2 \mid X_k\right] \\
&+ 2\Delta t \mathbb{E}\left[\|\Pi(X_k)\xi_k\|^2 \mid X_k\right]
\end{aligned}
$$

because $\mathbb{E}[\xi_k \mid X_k] = \mathbb{E}[\epsilon_k(X_k) \mid X_k] = 0$ and $\Delta X_k = (b(X_k) + \epsilon_k(X_k))\Delta t + \sqrt{2\Delta t}\Pi(X_k)\xi_k$. Applying the disspativity and linear growth assumption, we have

$$
\mathbb{E}\left[\|X_{k+1}\|^2 \mid \mathcal{F}_k\right] \leq (1 - (2m - B\Delta t)\Delta t)\|X_k\|^2 + \left(2c + 2(d-1) + (A + \frac{2C_h}{N})\Delta t\right)\Delta t
$$

because

$$
\mathbb{E}\left[\|\epsilon_k(X_k)\|^2 \mid X_k\right] \leq \frac{2}{N}\frac{\|\Pi(X_k)\nabla^2 h(X_k)\|_F^2}{\|\nabla h(X_k)\|^2} \leq \frac{2C_h}{N}
$$

holds from the tangential Hessian-gradient ratio assumption, and $\mathbb{E}\left[\|\Pi(X_k)\xi_k\|^2 \mid X_k\right] = d - 1$. Because the step size control assumption guarantees $(1 - (2m - B\Delta t)\Delta t) \in (0, 1)$, we have the following inequality:

$$
\sup_{k\geq 0} \mathbb{E}[\|X_k\|^2] \leq \frac{2c + 2(d-1) + (A + \frac{2C_h}{N})\Delta t}{2m - B\Delta t} := M_N < \infty
$$

by taking expectations and iterating the recursion. Therefore, it holds that

$$
\begin{aligned}
\mathbb{E}\left[\|\Delta X_k\|^2 \mid \mathcal{F}_k\right] = \mathbb{E}\left[\|\Delta X_k\|^2 \mid X_k\right] &= \Delta t^2\|b(X_k)\|^2 + \Delta t^2 \mathbb{E}\left[\|\epsilon_k(X_k)\|^2 \mid X_k\right] + 2\Delta t(d-1) \\
&\leq \Delta t^2\left(A + B\|X_k\|^2\right) + \frac{2C_h\Delta t^2}{N} + 2\Delta t(d-1)
\end{aligned}
$$

and

$$
\sup_{k\geq 0} \mathbb{E}\|\Delta X_k\|^2 \leq \Delta t^2\left(A + BM_N + \frac{2C_h}{N}\right) + 2\Delta t(d-1) := M_1\Delta t^2 + M_2\Delta t < \infty
$$

with $M_1 := A + BM_N + \frac{2C_h}{N}$ and $M_2 = 2(d-1)$.

Similarly, let us define $B(X_k) := (b(X_k) + \epsilon_k(X_k))\Delta t$. Then, under the similar proof, it holds that

$$
\begin{aligned}
\mathbb{E}\left[\|B(X_k)\|^2 \mid \mathcal{F}_k\right] = \mathbb{E}\left[\|B(X_k)\|^2 \mid X_k\right] &= \Delta t^2\|b(X_k)\|^2 + \Delta t^2 \mathbb{E}\left[\|\epsilon_k(X_k)\|^2 \mid X_k\right] \\
&\leq \Delta t^2\left(A + B\|X_k\|^2\right) + \frac{2C_h\Delta t^2}{N}
\end{aligned}
$$

and

$$
\sup_{k\geq 0} \mathbb{E}\left[\|B(X_k)\|^2\right] \leq \Delta t^2\left(A + BM_N + \frac{2C_h}{N}\right) = \Delta t^2 M_1. \tag{10}
$$

Next, we prove the decaying property of $h : \mathbb{R}^d \to \mathbb{R}$. From the 2nd order Taylor expansion on $h$, the following holds almost surely:

$$
\begin{aligned}
h(X_{k+1}) &= h(X_k) + \nabla h(X_k)\Delta X_k + \frac{1}{2}\Delta X_k^T \nabla^2 h(\tilde{X}_k)\Delta X_k \\
&= (1 - \alpha\Delta t)h(X_k) - \mathsf{Tr}\left(\Pi(X_k)\nabla^2 h(X_k)\right)\Delta t + \left(\zeta_k^T\Pi(X_k)\nabla^2 h(X_k)\Pi(X_k)\zeta_k\right)\Delta t \\
&\quad + \frac{1}{2}\Delta X_k^T \nabla^2 h(\tilde{X}_k)\Delta X_k - \left(\zeta_k^T\Pi(X_k)\nabla^2 h(X_k)\Pi(X_k)\zeta_k\right)\Delta t
\end{aligned}
$$

where $\tilde{X}_k$ is some point $\in \mathbb{R}^d$ between $X_k$ and $X_{k+1}$ and $\Delta X_k := X_{k+1} - X_k$. Therefore, by applying thet previous observations, we get the following:

$$
\mathbb{E}[h(X_{k+1})|\mathcal{F}_k] \leq (1 - \alpha\Delta t)h(X_k) + \frac{L}{2}\mathbb{E}\left[\|B(X_k)\|^2 \mid X_k\right] + L\Delta t\mathbb{E}\left[\|\Delta X_k\|^2 \mid X_k\right]
$$

which implies

$$
\mathbb{E}[h(X_{k+1})] \leq (1 - \alpha\Delta t)\mathbb{E}\left[h(X_k)\right] + \frac{L\Delta t^2(M_1(1 + 2\Delta t) + 2M_2)}{2}.
$$

By applying the telescoping sum with expectation, we recover the following formula:

$$\mathbb{E}\left[h(X_K)\right] \le (1 - \alpha \Delta t)^K \mathbb{E}\left[h(X_0)\right] + \frac{L\Delta t(M_1(1 + 2\Delta t) + 2M_2)}{2\alpha}.$$

Finally, we note that

$$M_N = \mathcal{O}\left(d + \Delta t\left(1 + \frac{1}{N}\right)\right), \quad M_1 = \mathcal{O}\left(d + \Delta t\left(1 + \frac{1}{N}\right) + \frac{1}{N}\right), \quad M_2 = \mathcal{O}(d)$$

and, therefore,

$$\mathbb{E}\left[h(X_K)\right] \le (1 - \alpha \Delta t)^K \mathbb{E}\left[h(X_0)\right] + \mathcal{O}\left(\frac{\Delta t}{\alpha}\left(d + \frac{1}{N}\right)\right).$$

$\square$

As we can see in the above proposition, the effect of $N$ scales with $\mathcal{O}(1/N)$ and vanishes as $N \to \infty$. Therefore, the usage of the Hutchinson estimator does not affect the convergence speed of constraint functions under the single-equality scenario with sufficiently regular $h$ and $f$.

**Remark 6** (Geometric control of $\lambda_{\text{LSI}}$). We state the following result from geometric analysis that provides a lower bound for $\lambda_{\text{LSI}}$:

**Theorem A.1** (Informal, [49, 47]). *Let $\Sigma$ be a compact Riemannian manifold with diameter $D$ and non-negative Ricci curvature. Then, $\lambda_{LSI} \ge \frac{\lambda_1}{1 + 2D\sqrt{\lambda_1}}$, or $\lambda_{LSI} \ge \frac{\pi^2}{(1+2\pi)D^2}$ holds, where $\lambda_1$ is the first eigenvalue of the Laplace-Beltrami operator on $\Sigma$.*

This theorem implies if the constraints shrink the diameter $D$ (or increase $\lambda_1$) of $\Sigma$, the lower bound of $\lambda_{\text{LSI}}$ increases, so the on-manifold decay $\mathsf{KL}^{\Sigma}(\tilde{\rho}_t || \rho_\Sigma)$ accelerates (dominated by its exponential rate $-2\lambda_{\text{LSI}}$ appearing in Theorem 1 or Theorem 3).

# B    Algorithms of OLLA, OLLA-H, and baselines

---

**Algorithm 1** Euler–Maruyama discretization of OLLA & OLLA-H

---

1: **Input:** initial point $x_0 \in \mathbb{R}^d$, step size $\Delta t$, number of steps $K$, landing rate $\alpha$, boundary repulsion rate $\epsilon$, potential $f$, constraints $\{h_i\}_{i=1}^m$, $\{g_j\}_{j=1}^l$, Hutchinson probe numbers $N$, `mode` $\in \{\text{OLLA}, \text{OLLA-H}\}$
2: **Output:** sample trajectory $\{x_k\}_{k=0}^K$

---

3: **for** $k = 0, \ldots, K-1$ **do**
4:     Evaluate constraints: $\{h_i(x_k)\}_{i=1}^m$, $\{g_j(x_k)\}_{j=1}^l$
5:     Compute: $\nabla f(x_k)$, $J(x_k)$, $\nabla J(x_k)$
6:     Compute: $G(x_k) \leftarrow \nabla J(x_k) \nabla J(x_k)^\mathsf{T}$
7:     **if** $\text{rank}(G) < m + I_x$: **then** $G^{-1} \leftarrow G^\dagger$ **else** $G^{-1} \leftarrow G^{-1}$
8:     **if** `mode` $= \text{OLLA}$:
        Compute: $\texttt{Tr} \leftarrow [\mathsf{Tr}\left(\Pi(x_k)\nabla^2 h_1\right), ..., \mathsf{Tr}\left(\Pi(x_k)\nabla^2 g_{i_{|I_x|}}\right)]^T$
9:     **else**
        Compute: $\texttt{Tr} \leftarrow \sum_{k=1}^N [v_k^T \Pi(x_k)\nabla^2 h_1 v_k, ..., v_k^T \Pi(x_k)\nabla^2 g_{i_{|I_x|}} v_k]^T$, $v_k \sim \mathcal{N}(0, I_d)$
10:    Compute: $\mathcal{H}(x_k) \leftarrow \nabla J(x_k)^T G^{-1}(x_k)\texttt{Tr}$
11:    Compute: $q(x_k) \leftarrow -\Pi(x_k)\nabla f(x_k) - \alpha \nabla J(x_k)^T G^{-1}(x_k) J(x_k) + \mathcal{H}(x_k)$
12:    Update: $x_{k+1} \leftarrow x_k + q(x_k)\Delta t + \sqrt{2\Delta t}\Pi(x_k)\xi_k$ where $\xi_k \sim \mathcal{N}(0, I_d)$
13: **end for**
14: **return** $\{x_k\}_{k=0}^K$

---

**Remark 7** (Pseudo-inverse of Gram matrix when LICQ fails). A second issue of OLLA arises if the Gram matrix $G = \nabla J\nabla J^T$ becomes singular. This may happen when the LICQ condition momentarily fails near the neighborhood of $\Sigma$. In that case, we replace the inverse $G^{-1}$ with the Moore-Penrose pseudo-inverse $G^\dagger$. Because $G^\dagger$ still projects onto the row space of $\nabla J$ and annihilates its null space, it enforces the same exact orthogonality to constraint gradients, preserving the exponential landing behavior and numerical stability of OLLA.

---

**Algorithm 2** Constrained Langevin (CLangevin) with slack variables [19, 20]

---

1: **Input:** initial position $x_0 \in \Sigma$, step size $\Delta t$, number of steps $K$, potential $f$, constraints $\{h_i\}_{i=1}^m$, $\{g_j\}_{j=1}^l$, projection iterations $L$, tolerance $\tau$, regularization $\lambda$
2: **Output:** sample trajectory $\{x_k\}_{k=0}^K$

---

3: Initialize slack variables: $s_{0,j} \leftarrow \sqrt{\max\{-2g_j(x_0), 0\}}$ for $j \in [l]$
4: Set the extended state: $s_0 \leftarrow (s_{0,1}, ..., s_{0,l}) \in \mathbb{R}^l$, $y_0 \leftarrow (x_0, s_{0,1}, ..., s_{0,l}) \in \mathbb{R}^{d+l}$
5: **for** $k = 0, \ldots, K-1$ **do**
6:     Draw $\xi_k \sim \mathcal{N}(0, I_{d+l})$
7:     Compute the augmented constraint vector:

$$J(y_k) = \left[h_1(x_k), \ldots, h_m(x_k), g_1(x_k) + \frac{1}{2}s_{k,1}^2, \ldots, g_l(x_k) + \frac{1}{2}s_{k,l}^2\right]^T$$

8:     Constrained update:

$$y_{k+1} \leftarrow y_k - \nabla f_{\text{ext}}(y_k)\Delta t + \sqrt{2\Delta t}\xi_k + \nabla J(y_k)^T \lambda_k$$

   where $f_{\text{ext}}(x, s) = f(x)$ and $\lambda_k$ is chosen such that $\|J(y_{k+1})\|_\infty \leq \tau$ by Newton's method with regularization $\lambda$ and max iterations $L$
9:     Set : $x_k \leftarrow y_k[1:d]$, $s_k \leftarrow y_k[d+1:l]$
10: **end for**
11: **Output:** $\{x_k\}_{k=0}^K$

---

**Algorithm 3** Constrained HMC (CHMC) with slack variables [19, 20]

---

1: **Input:** initial position $x_0 \in \Sigma$, step size $\Delta t$, number of steps $K$, potential $f$, constraints $\{h_i\}_{i=1}^m, \{g_j\}_{j=1}^l$, projection iterations $L$, tolerance $\tau$, regularization $\lambda$, friction $\gamma$

2: **Output:** sample trajectory $\{x_k\}_{k=0}^K$

---

3: Initialize slack variable: $s_{0,j} \leftarrow \sqrt{\max\{-2g_j(x_0), 0\}}$ for $j \in [l]$

4: Set the extended state: $s_0 \leftarrow (s_{0,1}, ..., s_{0,l}) \in \mathbb{R}^l$, $y_0 \leftarrow (x_0, s_{0,1}, ..., s_{0,l}) \in \mathbb{R}^{d+l}$

5: Sample momentum $p_0 \sim \mathcal{N}(0, I_{d+l})$ such that $\nabla J(y_0)p_0 = 0$

6: **for** $k = 0, \ldots, K - 1$ **do**

7:     Draw $\xi_k, \xi_{k+1/2} \sim \mathcal{N}(0, I_{d+l})$

8:     Compute the augmented constraint vector:

$$J(y_k) = \left[h_1(x_k), \ldots, h_m(x_k), g_1(x_k) + \frac{1}{2}s_{k,1}^2, \ldots, g_l(x_k) + \frac{1}{2}s_{k,l}^2\right]^T$$

9:     **Midpoint Euler step**:

$$p_{k+1/4} = p_k - \frac{\Delta t}{4}\gamma(p_k + p_{k+1/4}) + \sqrt{\Delta t \gamma}\xi_k + \nabla J(y_k)^T \lambda_{k+1/4}$$

    such that $\nabla J(y_k)p_{k+1/4} = 0$

10:     **Verlet step - (1)**:

$$p_{k+1/2} = p_{k+1/4} - \frac{\Delta t}{2}\nabla f_{\text{ext}}(y_k) + \nabla J(y_k)^T \lambda_{k+1/2}$$
$$y_{k+1} = y_k + p_{k+1/2}\Delta t$$

    such that $\|J(y_{k+1})\|_\infty \leq \tau$ by Newton's method with regularization $\lambda$ and max iterations $L$

11:     **Verlet step - (2)**:

$$p_{k+3/4} = p_{k+1/2} - \frac{\Delta t}{2}\nabla f_{\text{ext}}(y_{k+1}) + \nabla J(y_{k+1})\lambda_{k+3/4}$$

    such that $\nabla J(y_{k+1})^T p_{k+3/4} = 0$

12:     **Midpoint Euler step:**

$$p_{k+1} = p_{k+3/4} - \frac{\Delta t}{4}\gamma(p_{k+3/4} + p_{k+1}) + \sqrt{\Delta t \gamma}\xi_{k+1} + \nabla J(y_{k+1})\lambda_{k+1}$$

    such that $\nabla J(y_{k+1})^T p_{k+1} = 0$

13:     Set : $x_k \leftarrow y_k[1 : d]$, $s_k \leftarrow y_k[d + 1 : l]$

14: **end for**

15: **Output:** $\{x_k\}_{k=0}^K$

---

---

**Algorithm 4** Constrained generalized HMC (CGHMC) with MH correction [19, 21]

---

1: **Input:** initial position $x_0 \in \Sigma$, step size $\Delta t$, number of steps $K$, potential $f$, equality constraints $\{h_i\}_{i=1}^m$, $\{g_j\}_{j=1}^l$, projection iterations $L$, tolerance $\tau$, regularization $\lambda$, friction $\gamma$
2: **Output:** sample trajectory $\{x_k\}_{k=0}^K$

---

3: Sample momentum $p_0 \sim \mathcal{N}(0, I_d)$ such that $\nabla J(x_0)p_0 = 0$
4: **for** $k = 0, \ldots, K-1$ **do**
5:     Draw $\xi_k, \xi_{k+1/2} \sim \mathcal{N}(0, I_d)$
6:     Compute the augmented constraint vector: $J(x_k) = [h_1(x_k), \ldots, h_m(x_k)]$
7:     **Midpoint Euler step**:

$$p_{k+1/4} = p_k - \frac{\Delta t}{4}\gamma(p_k + p_{k+1/4}) + \sqrt{\Delta t \gamma}\xi_k + \nabla J(x_k)^T \lambda_{k+1/4}$$

    such that $\nabla J(x_k)p_{k+1/4} = 0$
8:     Compute the Hamiltonian: $H(x_k, p_{k+1/4}) = f(x_k) + \frac{1}{2}\|p_{k+1/4}\|_2^2$
9:     **Verlet step - (1)**:

$$p_{k+1/2} = p_{k+1/4} - \frac{\Delta t}{2}\nabla f(x_k) + \nabla J(x_k)^T \lambda_{k+1/2}$$
$$\tilde{x}_{k+1} = x_k + p_{k+1/2}\Delta t$$

    such that $\|J(\tilde{x}_{k+1})\|_\infty \leq \tau$ by Newton's method with regularization $\lambda$ and max iterations $L$
10:     **Verlet step - (2)**:

$$\tilde{p}_{k+3/4} = p_{k+1/2} - \frac{\Delta t}{2}\nabla f(\tilde{x}_{k+1}) + \nabla J(\tilde{x}_{k+1})\lambda_{k+3/4}$$

    such that $\nabla J(\tilde{x}_{k+1})^T \tilde{p}_{k+3/4} = 0$
11:     Compute the Hamiltonian: $H(\tilde{x}_{k+1}, \tilde{p}_{k+3/4}) = f(\tilde{x}_{k+1}) + \frac{1}{2}\|\tilde{p}_{k+3/4}\|_2^2$
12:     **Metropolis-Hasting Correction:** With probability

$$\min\left\{\exp\left(-(H(\tilde{x}_{k+1}, \tilde{p}_{k+3/4}) - H(x_k, p_{k+1/4}))\right), 1\right\}$$

    set

$$(x_{k+1}, p_{k+\frac{3}{4}}) = (\tilde{x}_{k+1}, \tilde{p}_{k+\frac{3}{4}}), \quad \text{if } g(\tilde{x}_{k+1}) \leq 0$$

    Otherwise, reject and flip momentum $(x_{k+1}, p_{k+\frac{3}{4}}) = (x_k, -p_{k+\frac{1}{4}})$
13:     **Midpoint Euler step:**

$$p_{k+1} = p_{k+3/4} - \frac{\Delta t}{4}\gamma(p_{k+3/4} + p_{k+1}) + \sqrt{\Delta t \gamma}\xi_{k+1} + \nabla J(x_{k+1})\lambda_{k+1}$$

    such that $\nabla J(x_{k+1})^T p_{k+1} = 0$
14: **end for**
15: **Output:** $\{x_k\}_{k=0}^K$

---

# C Proof of Basic Properties on $\Sigma$

## C.1 Intrinsic Gradient on $\Sigma$

Recall that our target manifold $\Sigma$ is defined by $\Sigma := \{x \in \mathbb{R}^d \mid h(x) = 0, g(x) \leq 0\}$. For a smooth function $f : \Sigma \to \mathbb{R}$, the Riemannian gradient $\nabla_\Sigma f$ is defined by the relation

$$\langle \nabla_\Sigma f(x), v \rangle = df_x(v), \quad \text{for every } v \in T_x\Sigma$$

To check $\Pi(x)\nabla f(x) = \nabla_\Sigma f(x)$, observe that $\Pi(x)\nabla f(x) \in T_x\Sigma$ and $\Pi(x)\nabla f(x)$ reproduces $df_x$ on every tangent vector. This is because if $v \in T_x\Sigma$, $\langle \Pi(x)\nabla f(x), v \rangle = \langle \nabla f(x), \Pi(x)v \rangle = \langle \nabla f(x), v \rangle = df_x(v)$ due to the symmetric nature of $\Pi(x)$. Therefore, the intrinsic gradient on $\Sigma$ can be extrinsically defined by $\nabla_\Sigma f(x) = \Pi(x)\nabla f(x)$.

## C.2 Intrinsic Divergence on $\Sigma$

Recall that if $X \in \mathcal{X}(\Sigma)$ where $\mathcal{X}(\Sigma)$ is a set of smooth vector fields on $\Sigma$, then the divergence of $X$ on $\Sigma$ is defined by $\mathsf{div}_\Sigma X(x) = \sum_{i=1}^{d-(m+|I_x|)} \langle \nabla_{E_i}^\Sigma X, E_i \rangle$ where $\nabla^\Sigma$ is Levi-Civita connection on $\Sigma$ and $\{E_1, ... E_{d-(m+|I_x|)}\}$ is an orthonormal frame of $T_x\Sigma$. To check its extrinsic formula, first observe that the induced Levi-Civita connection is given by $\nabla_Y^\Sigma X(x) := (\nabla_Y X)^\top = \Pi(x)\nabla_Y X(x)$ where $X, Y \in \mathcal{X}(\Sigma)$, $\top$ indicates tangential component on $\Sigma$, and $\nabla$ is the Levi-Civita connection (or usual directional gradient) in $\mathbb{R}^d$. Then, if $X$ and $\{E_1, ... E_{d-(m+I_x)}\}$ are extended to the ambient space $\mathbb{R}^d$, it holds that

$$\mathsf{div}_\Sigma X = \sum_{i=1}^{d-(m+|I_x|)} \langle \Pi\nabla_{E_i} X, E_i \rangle = \sum_{i=1}^{d-(m+|I_x|)} \langle \Pi\nabla X E_i, E_i \rangle = \mathsf{Tr}(\Pi\nabla X)$$

where the last equation is obtained by taking a basis $\{E_1, ... E_{d-(m+|I_x|)}, \nu_1, ..., \nu_{m+|I_x|}\}$ of $\mathbb{R}^d$ with $\{\nu_1, ..., \nu_{m+|I_x|}\}$ being the orthonormal basis of $(T_x\Sigma)^\perp$, and applying the definition of trace.

## C.3 Recoverable Tubular Neighborhood of Boundaryless Riemannian Manifold

Let $U_\epsilon(\Sigma) := \{x \in \mathbb{R}^d \mid \mathsf{dist}(x, \Sigma) < \epsilon\}$ be a tubular neighborhood of $\Sigma$ with reach $\epsilon$, whose existence is guaranteed by the compactness of $\Sigma$ [50]. In the proof of Theorem 1, we require a property that enables us to recover the unique $x \in U_\epsilon(\Sigma)$ such that $y = \pi(x)$ and $h(x) = p$, given the information of $y \in \Sigma$ and $p \in \mathbb{R}^m$.

The following theorem indicates the existence of such a nice neighborhood of $\Sigma$ such that any $x$ in this neighborhood can be recovered from the information of $y, p$. In addition to this, we need a regularity lemma to connect the decrease of $\|h(x)\|_2$ with the decrease of $\mathsf{dist}(x, \Sigma)$, which is a crucial property to show the convergence of the $W_2$ distance.

---

**Lemma C.1** (Regularity lemma). *Assume $h \in C^2$ and LICQ condition is satisfied on $\Sigma$. Then, there exist constants $\hat{\epsilon}, \kappa > 0$ such that for all $x \in U_{\hat{\epsilon}}(\Sigma) \subset U_\epsilon(\Sigma)$*

$$\|h(x)\|_2 \geq \kappa\|x - \pi(x)\|_2$$

*where $\kappa = \frac{1}{2}\min_{y \in \Sigma} \sigma_{min}(\nabla h(y)) > 0$.*

---

*Proof.* For each $y \in \Sigma$, $\nabla h(y)$ has full rank by the LICQ condition, so its smallest singular value $\sigma_y := \sigma_{\min}(\nabla h(y)) > 0$. Since, any $x \in U_\epsilon(\Sigma)$ can be decomposed into $x = y + v$ for some $y \in \Sigma, v \in N_y(\Sigma) := (T_y\Sigma)^\perp$, Taylor's theorem gives

$$h(x) = h(y + v) = \nabla h(y)v + R_y(v)$$

where the norm of the remainder term $R_y(v)$ is bounded above by $\|R_y(v)\|_2 \leq \frac{1}{2}M\|v\|_2^2$ for $M := \sup_{y \in U_{\hat{\epsilon}}(\Sigma)}\|\nabla^2 h(y)\|_2 < \infty$. Hence,

$$\|h(x)\|_2 = \|h(y + v)\|_2 \geq \|\nabla h(y)v\|_2 - \|R_y(v)\|_2 \geq \sigma_y\|v\|_2 - \frac{1}{2}M\|v\|_2^2.$$

Furthermore, because $h \in C^2$ and $y \mapsto \sigma_{min}(\nabla h(y))$ is a continuous function on $\Sigma$, the compactness of $\Sigma$ implies there exists $\sigma = \min_{y \in \Sigma} \sigma_{min}(\nabla h(y)) > 0$. Also, choose $0 < \hat{\epsilon} \leq \epsilon$ such that $M\hat{\epsilon} \leq \sigma$. Then whenever $x \in U_{\hat{\epsilon}}(\Sigma)$ so that $\|v\|_2 < \hat{\epsilon}$, we get

$$\|h(x)\|_2 \geq \sigma\|v\|_2 - \frac{1}{2}M\|v\|_2^2 \geq (\sigma - \frac{1}{2}M\hat{\epsilon})\|v\|_2 \geq \frac{\sigma}{2}\|v\|_2 = \frac{\sigma}{2}\|x - \pi(x)\|_2.$$

$\square$

Lemma C.1 shows that an upper bound on $\|h(x)\|_2$ yields an upper bound on $\|x - \pi(x)\|_2$; however, this guarantee holds only once $x$ has entered $U_{\hat{\epsilon}}$. Therefore, to ensure that $x$ indeed enters $U_{\hat{\epsilon}}$ whenever $\|h(x)\|_2$ is sufficiently small, we appeal to Lemma C.2.

**Lemma C.2** (Entrance cutoff of $U_{\hat{\epsilon}}(\Sigma)$ in terms of $\|h(x)\|_2$). *Assume $h$ is continuous and coercive, i.e. $\|h(x)\|_2 \to \infty$ whenever $\|x\|_2 \to \infty$. Then, there exists $\delta^* > 0$ such that $\mathrm{dist}(x, \Sigma) < \hat{\epsilon}$ if $\|h(x)\|_2 < \delta^*$ where $\hat{\epsilon} > 0$ is the constant defined in Lemma C.1.*

*Proof.* First, we show that $h$ is a proper map. Let $C$ be a compact set in $\mathbb{R}^m$ so that it is closed and bounded. Then, $h^{-1}(C)$ is closed because $h$ is continuous. Also, suppose $h^{-1}(C)$ were unbounded so that there is a sequence $\{x_k\}_{k \in \mathbb{N}} \in h^{-1}(C)$ with $\|x_k\|_2 \to \infty$. Then, by coercivity of $h$, $\|h(x_k)\|_2 \to \infty$ while every $h(x_k)$ lies in $C$, which is bounded and leads to a contradiction. Therefore, $h^{-1}(C)$ is bounded and closed and therefore is compact by the Heine-Borel theorem. This proves $h$ is a proper map.

Now, define a set $S_{\hat{\epsilon}} := \{x \in \mathbb{R}^d \mid \mathrm{dist}(x, \Sigma) \geq \hat{\epsilon}\}$ and assume $\inf_{x \in S_{\hat{\epsilon}}}\|h(x)\|_2 = 0$. In this case, there must be a sequence $\{x_k\}_{k \in \mathbb{N}} \subset S_{\hat{\epsilon}}$ with $\|h(x_k)\|_2 \to 0$. This implies $x_k \in h^{-1}(\bar{B}(0, 1))$ for $\forall k \geq K$ for some $K \in \mathbb{N}$. Since $h^{-1}(\bar{B}(0, 1))$ is a compact set, there is a subsequence $\{x_{k_j}\}$ of $\{x_k\}$ converging to some $x_*$ from the Bolzano-Weierstrass theorem. Subsequently, the continuity of $h$ implies $h(x_*) = \lim_{j \to \infty} h(x_{k_j}) = 0$ and $x_* \in \Sigma$. However, because $S_{\hat{\epsilon}}$ is closed and every $x_{k_j} \in S_{\hat{\epsilon}}$, it should satisfy $x_* \in S_{\hat{\epsilon}} \Rightarrow \mathrm{dist}(x_*, \Sigma) \geq \hat{\epsilon}$, which is a contradiction that $x_* \in \Sigma \Leftrightarrow \mathrm{dist}(x, \Sigma) = 0$. Therefore, $\delta^* := \inf_{x \in S_{\hat{\epsilon}}}\|h(x)\|_2 > 0$ holds and $\mathrm{dist}(x, \Sigma) < \hat{\epsilon}$ if $\|h(x)\|_2 < \delta^*$. $\square$

**Theorem C.1** (Recoverable tubular neighborhood). *Let $\Sigma := \{x \in \mathbb{R}^d \mid h(x) = 0\}$ be a compact and boundaryless Riemannian manifold with LICQ condition. Then, there exists a **recoverable tubular neighborhood** of $\Sigma$ with width $\delta$, $\hat{U}_\delta(\Sigma) := \{x \in \mathbb{R}^d \mid \|h(x)\|_2 < \delta\} \subset U_{\hat{\epsilon}}(\Sigma)$ for some $\delta > 0$ such that*

1. *The nearest-point projection map $\pi : \hat{U}_\delta(\Sigma) \to \Sigma, \pi(x) = \mathrm{argmin}_{y \in \Sigma}\|x - y\|_2$ is well-defined.*

2. *The following recovery map $\zeta(y, p)$ is well-defined*

$$\zeta(y, p) : \Sigma \times B(0, \delta) \to \hat{U}_\delta(\Sigma), \quad \zeta(y, p) = y + \nabla h(y)^T L(y, p) \tag{11}$$

   *where $L : \Sigma \times B(0, \delta) \to \mathbb{R}^m$ is the $C^1$ function such that $h(\zeta(y, p)) = p$ and $\pi(\zeta(y, p)) = y$ for $\forall(y, p) \in \Sigma \times B(0, \delta)$.*

*In this case, we refer to $\hat{U}_\delta(\Sigma)$ as the **recoverable tubular neighborhood** of $\Sigma$.*

*Proof.* Define $F((y, p), L) : \mathbb{R}^{(d+m)+m} \to \mathbb{R}^m$ by $F((y, p), L) = h(y + \nabla h(y)^T L) - p$. Then, for each $y \in \Sigma$, $F((y, 0), 0) = h(y) - 0 = 0$. Also, $\nabla_L F((y, p), L) = \nabla h(y + \nabla h(y)^T L)\nabla h(y)^T$ implies $\nabla_L F((y, 0), 0) = \nabla h(y)\nabla h(y)^T$, which is invertible due to full rank assumption of $\nabla h(y)$.

Therefore, by the implicit function theorem, there exists $\delta_y > 0$ and an open set $U_y := \{(y', p') \in \Sigma \times \mathbb{R}^m \mid d_\Sigma(y, y') < \delta_y, \|p'\|_2 < \delta_y\} \subset \mathbb{R}^{d+m}$ such that there exists a unique $C^1$ function $L : U_y \to \mathbb{R}^m$ satisfying $L(y, 0) = 0$ and $F((y, p), L(y, p)) = 0$ for $\forall(y, p) \in U_y$.

Now, observe that $\cup_{y \in \Sigma} U_y$ is the open cover of $\Sigma \times \{0\}$, which is a compact set. Therefore, using its finite subcovers, we can pick $\hat{\delta} > 0$ such that for $\forall y \in \Sigma$ and $\|p\|_2 < \hat{\delta}$, $L(y, p)$ is

well-defined $C^1$ function on $\Sigma \times B(0, \hat{\delta})$ and the recovery map $\zeta(y, p) : \Sigma \times B(0, \hat{\delta}) \to \mathbb{R}^d$, $\zeta(y, p) = y + \nabla h(y)^T L(y, p)$ is also well-defined and $C^1$ on $\Sigma \times B(0, \hat{\delta})$ such that $h(\zeta(y, p)) = p$ holds (from $F((y, p), L(y, p)) = 0$).

Finally, setting $\delta = \min\left\{\hat{\delta}, \delta^*\right\}$ (where the constant $\delta^*$ comes from Lemma C.2) and $\hat{U}_\delta(\Sigma) := \left\{x \in \mathbb{R}^d \mid \|h(x)\|_2 < \delta\right\} \subset U_{\hat{\epsilon}}(\Sigma)$ enables the well-defined nearest-point projection map $\pi$ on $\hat{U}_\delta(\Sigma)$. By applying $\pi$ to $\zeta(y, p)$, we recover the formula $\pi(\zeta(y, p)) = \pi(y + \nabla h(y)^T L(y, p)) = y$, $\forall (y, p) \in \Sigma \times B(0, \delta)$. $\qquad \square$

### C.4 Recoverable Tubular Neighborhood of Riemannian Manifold with Boundary

The following results generalize the proceeding lemmas and theorems by incorporating inequality constraints alongside the equality constraints.

**Lemma C.3** (Regularity lemma with boundary). *Let $\Sigma = \left\{x \in \mathbb{R}^d \mid h(x) = 0, g(x) \le 0\right\}$. Assume $h, g \in C^2$ and LICQ condition is satisfied on $\Sigma$. Then, there exist constants $\hat{\epsilon}, \kappa > 0$ such that for all $x \in U_{\hat{\epsilon}}(\Sigma) \subset U_\epsilon(\Sigma)$*

$$\|h(x)\|_2 + \|g_{I_{\pi(x)}}(x)\|_2 \ge \kappa \|x - \pi(x)\|_2$$

*where $\kappa = \frac{1}{2} \min_{y \in \Sigma} \sigma_{\min}\left(\left[\nabla h(y)^T, \nabla g_{I_y}(y)^T\right]\right) > 0$.*

*Proof.* For every subset $I \subset [l]$, consider $\Sigma_I := \{y \in \Sigma \mid I_y = I\}$. Then for $y \in \Sigma_I$, $\nabla J_I(y) := [\nabla h(y)^T, \nabla g_I(y)^T]^T \in \mathbb{R}^{(m+|I|) \times d}$ has full rank due to the LICQ condition, so its smallest singular value $\sigma_y^I := \sigma_{\min}(\nabla J_I(y)) > 0$. Because $y \mapsto \sigma_y^I$ is continuous on $\Sigma_I$, we also have $\sigma := \min_{I \subset [l]} \inf_{y \in \Sigma_I} \sigma_y^I > 0$ by Lemma C.4. Also, since any $x \in U_\epsilon(\Sigma)$ can be decomposed into $x = y + v$ for some $y \in \Sigma, v \in N_y(\Sigma)$, Taylor's theorem gives

$$h(x) = h(y + v) = \nabla h(y)v + R_h(v), \quad g_{I_y}(x) = g_{I_y}(y + v) = \nabla g_{I_y}(y)v + R_g(v)$$

where the norm of the remainder term $R_h(v), R_g(v)$ is bounded above by $\|R_h(v)\|_2 \le \frac{1}{2}M\|v\|_2^2, \|R_g(v)\|_2 \le \frac{1}{2}M\|v\|_2^2$ for $M := \max_{z \in U_\epsilon(\Sigma)} \left\{\|\nabla^2 h(z)\|_2, \|\nabla^2 g(z)\|_2\right\} < \infty$. Now, we set $w = \nabla J(y)v$, which satisfies $\|w\|_2 \ge \sigma \|v\|_2$ by the definition of $\sigma$. Then, we observe

$$\|h(x)\|_2 + \|g_{I_y}(x)\|_2 = \|w\|_2 - \left(\|w\|_2 - \|h(x)\|_2 - \|g_{I_y}(x)\|_2\right) \overset{(\circ)}{\ge} \sigma\|v\|_2 - M\|v\|_2^2$$

where $(\circ)$ comes from the following observation:

$$\|w\|_2 - (\|h(x)\|_2 + \|g_{I_y}(x)\|_2) \le \|w\|_2 - \|[h(x)^T, g_{I_y}(x)^T]^T\|_2 \le \|w - [h(x)^T, g_{I_y}(x)^T]^T\|_2$$
$$= \|[R_h(v)^T, R_g(v)^T]^T\|_2 \le \|R_h(v)\|_2 + \|R_g(v)\|_2$$
$$\le M\|v\|_2^2.$$

Now, choose small $0 < \hat{\epsilon} \le \epsilon$ such that $M\hat{\epsilon} \le \frac{\sigma}{2}$. Then whenever $x \in U_{\hat{\epsilon}}(\Sigma)$ so that $\|v\|_2 < \hat{\epsilon}$, we get

$$\|h(x)\|_2 + \|g_{I_{\pi(x)}}(x)\|_2 \ge \frac{\sigma}{2}\|v\|_2 = \frac{\sigma}{2}\|x - \pi(x)\|_2.$$

$\qquad \square$

**Lemma C.4.** *For every subset $I \subset [l]$, consider $\Sigma_I := \{y \in \Sigma \mid I_y = I\}$. For $y \in \Sigma_I$, define $\nabla J_I(y) := [\nabla h(y)^T, \nabla g_I(y)^T]^T \in \mathbb{R}^{(m+|I|) \times d}$ and $\sigma_y^I := \sigma_{min}(\nabla J_I(y)) > 0$. Then, $\inf_{y \in \Sigma_I} \sigma_y^I > 0$ holds.*

*Proof.* Suppose $\inf_{y \in \Sigma_I} \sigma_y^I = 0$, then there exists a sequence $y_k \in \Sigma_I$ with $\sigma_{min}(\nabla J_I(y_k)) \to 0$. Because $\Sigma$ is compact, the sequence has a subsequence of $y_k$ converging to $y_* \in \Sigma$ by Bolzano–Weierstrass theorem. In this case, there are two possibilities:

1. When $I_{y_*} = I$, $\sigma_{min}(J_I(y_*)) > 0$ by LICQ, contradicting the assumption $\inf_{y \in \Sigma_I} \sigma_y^I = 0$.

2. When $I_{y_*} = I \cup K$ for some non-empty set $K \subset [l]$, the LICQ condition implies $\nabla J_{I \cup K}(y_*) = [\nabla h(y_*)^T, \nabla g_I(y_*)^T, \nabla g_K(y_*)^T]^T$ has a full rank, thereby, imposing $\nabla J_I(y_*)$ to have full rank. This again implies $\inf_{y \in \Sigma_I} \sigma_y^I > 0$, a contradiction.

Note that $I_{y_*}$ does not deactivate already activated inequality index $i \in I$. This is because $g_i(y_k) = 0$ for $\forall i \in I$ and continuity of $g_i$ implies $g_i(y_*) = 0$. Thus, the previous argument on two possibilities completes the proof. $\qquad \square$

---

**Lemma C.5** (Entrance cutoff of $U_{\hat{\epsilon}}(\Sigma)$ in terms of $\|h(x)\|_2$ and $g(x)$). *Let $\Sigma := \{x \in \mathbb{R}^d \mid h(x) = 0, g(x) \le 0\}$. Assume $h, g$ are continuous and $h$ is coercive, i.e. $\|h(x)\|_2 \to$ whenever $\|x\|_2 \to \infty$. Then, there exists $\delta^* > 0$ such that $dist(x, \Sigma) < \hat{\epsilon}$ if $\|h(x)\|_2 < \delta^*$ and $g_i(x) < \delta^*$ for $i \in [l]$, where $\hat{\epsilon} > 0$ is defined in Lemma C.3.*

*Proof.* Define $S_{\hat{\epsilon}} := \{x \in \mathbb{R}^d \mid dist(x, \Sigma) \ge \hat{\epsilon}\}$ and let $\psi(x) := \max\{\|h(x)\|_2, g_1(x), ..., g_l(x)\}$. Assume $\inf_{x \in S_{\hat{\epsilon}}} \psi(x) = 0$. In this case, there must be a sequence $\{x_k\}_{k \in \mathbb{N}} \subset S_{\hat{\epsilon}}$ with $\|h(x_k)\|_2 \to 0$ and $g_i(x_k) \le \psi(x_k) \to 0$ for $\forall i \in [l]$.

This implies $x_k \in h^{-1}(\bar{B}(0,1))$, $\forall k \ge K$ and for some $K \in \mathbb{N}$. Since $h^{-1}(\bar{B}(0,1))$ is a compact set due to the properness of $h$ (Lemma C.2), there exists a subsequence $\{x_{k_j}\}$ of $\{x_k\}$ converging to some $x_*$ from the Bolzano-Weierstrass theorem. Subsequently, the continuity of $h$ implies $h(x_*) = \lim_{j \to \infty} h(x_{k_j}) = 0, g_i(x) \le 0, \forall i \in [l]$, therefore, $x_* \in \Sigma$.

However, because $S_{\hat{\epsilon}}$ is closed and every $x_k \in S_{\hat{\epsilon}}$, it should satisfy $x_* \in S_{\hat{\epsilon}} \Rightarrow dist(x_*, \Sigma) \ge \hat{\epsilon}$, which is a contradiction that $x_* \in \Sigma \Leftrightarrow dist(x, \Sigma) = 0$. Therefore, $\delta^* := \inf_{x \in S_{\hat{\epsilon}}} \psi(x) > 0$ holds and $dist(x, \Sigma) < \hat{\epsilon}$ if $\psi(x) < \delta^*$. $\qquad \square$

---

**Theorem C.2** (Recoverable tubular neighborhood with boundary). *Let $\Sigma := \{x \in \mathbb{R}^d \mid h(x) = 0, g(x) \le 0\}$ be a compact Riemannian manifold with boundary. Assume LICQ condition on $\Sigma$. Then, there exists a **recoverable tubular neighborhood** of $\Sigma$ with width $\delta$, $\hat{U}_\delta(\Sigma) := \{x \in \mathbb{R}^d \mid \|h(x)\|_2 < \delta, g(x) < \delta\} \subset U_{\hat{\epsilon}}(\Sigma)$ for some $\delta > 0$ such that*

1. *The nearest-point projection map $\pi : \hat{U}_\delta(\Sigma) \to \Sigma, \pi(x) = \operatorname{argmin}_{y \in \Sigma} \|x - y\|_2$ is well-defined.*

2. *The following recovery map $\zeta(y, p, q_{I_y})$ is well-defined*

   $$\zeta(y, p, q_{I_y}) : \Sigma \times B_m(0, \delta) \times B_{|I_y|}(0, \delta) \to \hat{U}_\delta(\Sigma), \quad \zeta(y, p, q_{I_y}) = y + \nabla J(y)^T L(y, p, q_{I_y})$$

   *where $I_y$ is the index set of active inequalities at $y \in \Sigma$, $J(y) := [h(y), g_{I_y}(y)] \in \mathbb{R}^{m+|I_y|}$, and $L : \Sigma \times B_m(0, \delta) \times B_{|I_y|}(0, \delta) \to \mathbb{R}^{m+|I_y|}$ is the function such that*

   $$h(\zeta(y, p, q_{I_y})) = p, \quad \begin{cases} g_i(\zeta(y, p, q_{I_y})) = q_i, & i \in I_y \\ g_i(\zeta(y, p, q_{I_y})) < 0, & i \notin I_y, \end{cases} \quad \pi(\zeta(y, p, q_{I_y})) = y$$

   $\forall (y, p, q_{I_y}) \in \Sigma \times B_m(0, \delta) \times B_{|I_y|}(0, \delta)$. *Furthermore, when $y \in int(\Sigma)$, $L = L(y, p)$ is a $C^1$ function on $\Sigma \times B_m(0, \delta)$.*

---

*Proof.* Let us first define $F((y, p, q_{I_y}), L) : \mathbb{R}^{(d+m+|I_y|)+(m+|I_y|)} \to \mathbb{R}^{m+|I_y|}$, $F((y, p, q_{I_y}), L) := (h(y + \nabla J(y)^T L) - p, g_{I_y}(y + \nabla J(y)^T L) - q_{I_y}) \in \mathbb{R}^{m+|I_y|}$ and consider the stratum $\Sigma_I := \{y \in \Sigma \mid I_y = I\}$ for each subset $I$ of inequality indices.

Then for $y \in \Sigma_I$, we observe that $F((y, 0, 0), 0) = 0$ and $\nabla_L F((y, 0, 0), 0) = \nabla J(y) \nabla J(y)^T$ which is invertible by the LICQ condition. Therefore, the implicit function theorem ensures the existence of $\delta_y > 0$, $U_y := \{(y', p', q_I') \in \Sigma_I \times \mathbb{R}^m \times \mathbb{R}^I \mid d_\Sigma(y, y') < \delta_y, \|p'\|_2 < \delta_y, \|q_I'\|_2 < \delta_y\}$

such that there exists a unique $C^1$ map $L : U_y \to \mathbb{R}^{m+|I|}$ satisfying $F((y, p, q_I), l(y, p, q_I)) = 0, \ \forall (y, p, q_I) \in U_y$.

Now, observe that $\cup_{y \in \Sigma_I} U_y$ is the open cover of $\Sigma_I \times \{0\}_m \times \{0\}_I$, which is a compact set. Therefore, using the finite subcovers of $\Sigma_I$, we can pick $\hat{\delta}_I > 0$ such that for $\forall y \in \Sigma_I, \|p\|_2 < \hat{\delta}_I$, and $\|g_I\|_2 < \hat{\delta}_I$, $L(y, p, q_I)$ is well-defined $C^1$ function on $\Sigma_I \times B_m(0, \hat{\delta}_I) \times B_{|I|}(0, \hat{\delta}_I)$ and the recovery map $\zeta(y, p, q_I) : \Sigma \times B_m(0, \hat{\delta}) \times B_{|I|}(0, \hat{\delta}) \to \mathbb{R}^d, \ \zeta(y, p, q_I) = y + \nabla J(y)^T L(y, p, q_I)$ is also well-defined $C^1$ map on $\Sigma_I \times B_m(0, \hat{\delta}_I) \times B_{|I|}(0, \hat{\delta}_I)$ such that $h(\zeta(y, p, q_I)) = p$ and $g_i(\zeta(y, p, q_I)) = q_i$ for $i \in I$, which comes from the property $F(y, p, q_I), l(y, p, q_I)) = 0$.

Furthermore, from the continuity of $g_i, i \notin I$, there exists $\gamma_I > 0$ such that $g_i(z) \le -\gamma_I$ for all $z \in B(y, \gamma_I)$ and the compactness of $\hat{U}_{\hat{\epsilon}}(\Sigma)$ gives $G$-Lipschitzness of $g_i$ on $U_{\hat{\epsilon}}(\Sigma)$ for all $i \in [l]$, for some $G > 0$. Also, we recall that $L(y, p, q_I)$ is $C^1$ map on $\left\{(y, p, q_I) \mid y \in \Sigma_I, \|p\|_2 < \hat{\delta}_I, \|g_I\|_2 < \hat{\delta}_I\right\}$ with $L(y, 0, 0) = 0$.

Hence, the Taylor expansion of $L(y, p, q_I)$ with boundedness of $\|\nabla_p L(y, p, q_I)\|_2, \|\nabla_{q_I} L(y, p, q_I)\|_2$ on $\left\{(y, p, q_I) \mid y \in \Sigma_I, \|p\|_2 < \hat{\delta}_I, \|g_I\|_2 < \hat{\delta}_I\right\}$ and boundedness of $\|\nabla J(y)\|_2$ on $\Sigma$ gives

$$\|L(y, p, q_I)\| \le C_I(\|p\|_2 + \|q_I\|_2) \quad \Rightarrow \quad \|\nabla J(y)^T L(y, p, q_I)\|_2 \le C_I'(\|p\|_2 + \|q_I\|_2)$$

for some $C_I, C_I' > 0$. Then, choosing $\tilde{\delta}_I := \min\left\{\hat{\delta}_I, \frac{\gamma_I}{4C_I'G}\right\}$ concludes that whenever $\|p\|_2 < \tilde{\delta}_I, \|q_I\|_2 < \tilde{\delta}_I$, the inequality

$$g_i(\zeta(y, p, q_I)) = g_i(y + \nabla J(y)^T L(y, p, q_I)) \le g_i(y) + G\|\nabla J(y)^T L(y, p, q_I)\|_2 \le -\frac{\gamma_I}{2} < 0$$

holds for all $i \notin I$. Finally, setting $\delta = \min\left\{\min_{I \subset [l]} \tilde{\delta}_I, \delta^*\right\}$ (where the constant $\delta^*$ comes from Lemma C.5) and $\hat{U}_\delta(\Sigma) := \left\{x \in \mathbb{R}^d \mid \|h(x)\|_2 < \delta, \|g(x)\|_2 < \delta\right\} \subset U_{\hat{\epsilon}}(\Sigma)$ enable well-defined nearest-point projection map $\pi$ on $\hat{U}_\delta(\Sigma)$. By applying $\pi$ to $\zeta(y, p, g_{I_y})$, we recover the formula $\pi(\zeta(y, p, q_{I_y})) = \pi(y + \nabla J(y)^T l(y, p, q_{I_y})) = y, \ \forall (y, p, q_{I_y}) \in \Sigma \times B_m(0, \delta) \times B_{|I_y|}(0, \delta)$. $\quad\square$

# D  Construction of SDE with Exponentially Fast Decaying Constraints

**Proposition 1** (Construction of OLLA and its closed form SDE). *Consider the following SDE:*

$$dX_t = q(X_t)dt + Q(X_t)dW_t \tag{12}$$

*where*

$$Q := \underset{\bar{Q} \in \mathbb{R}^{d \times d}}{\arg\min} \|\sqrt{2}I - \bar{Q}\|_F^2 \quad s.t \quad \begin{cases} \bar{Q}\nabla h_i = 0, \ \forall i \in [m], \\ \bar{Q}\nabla g_j = 0, \ \forall j \in I_x. \end{cases}$$

$$q := \underset{\bar{q} \in \mathbb{R}^d}{\arg\min} \|\bar{q} + \nabla f\|_2^2 \quad s.t \quad \begin{cases} \nabla h_i^T \bar{q} + \frac{1}{2}\mathsf{Tr}\left(\nabla^2 h_i QQ^T\right) + \alpha h_i = 0, & \forall i \in [m], \\ \nabla g_j^T \bar{q} + \frac{1}{2}\mathsf{Tr}\left(\nabla^2 g_j QQ^T\right) + \alpha(g_j + \epsilon) = 0, \ \forall j \in I_x \end{cases}$$

*Then, there exists a closed form SDE of* (12) *given by:*

$$dX_t = -[\Pi(X_t)\nabla f(X_t) + \alpha \nabla J(X_t)^T G^{-1}(X_t)J(X_t)]dt + \mathcal{H}(X_t)dt + \sqrt{2}\Pi(X_t), \quad (Ito)$$
$$dX_t = -[\Pi(X_t)\nabla f(X_t) + \alpha \nabla J(X_t)^T G^{-1}(X_t)J(X_t)]dt + \sqrt{2}\Pi(X_t) \circ dW_t, \quad (Strato.)$$

*where $\circ$ denotes the Stratonovich integral and*

$$\mathcal{H} := -\nabla J^T G^{-1}\left[\mathsf{Tr}\left(\nabla^2 h_1 \Pi\right), ..., \mathsf{Tr}\left(\nabla^2 h_m \Pi\right) \mathsf{Tr}\left(\nabla^2 g_{i_1} \Pi\right), ..., \mathsf{Tr}\left(\nabla^2 g_{i_{|I_x|}} \Pi\right)\right]^T$$

*is the related Ito-Stratonovich correction, or mean curvature of $\left\{x \in \mathbb{R}^d \mid h(x) = 0, g_{I_x}(x) = 0\right\}$.*

*Proof.* Define $J(x) := [h(x)^T, g_{I_x}^T + \epsilon\mathbb{1}_{|I_x|}]^T \in \mathbb{R}^{m+|I_x|}$ and $\nabla J(x) := [\nabla h(x)^T, \nabla g_{I_x}(x)^T]^T \in \mathbb{R}^{(m+|I_x|) \times d}$. Then, the Lagrangian function associated with the optimization problem for $Q$ becomes

$\mathcal{L}(\bar{Q}, \Lambda) := \|\sqrt{2}I_d - \bar{Q}\|_F^2 + \text{Tr}\left(\Lambda^T \nabla J \bar{Q}\right)$ where $\Lambda \in \mathbb{R}^{(m+|I_x|) \times d}$ is a Lagrangian multiplier. Then, the stationarity condition with respect to $\bar{Q}$ gives

$$0 = \frac{\partial L}{\partial \bar{Q}} = -2(\sqrt{2}I - \bar{Q}) + \Lambda^T \nabla J \quad \Rightarrow \quad \bar{Q} = \sqrt{2}I - \frac{1}{2}\Lambda^T \nabla J.$$

Also, the constraint condition implies

$$0 = \bar{Q}\nabla J^T = (\sqrt{2}I - \frac{1}{2}\Lambda^T \nabla J)\nabla J^T \quad \Rightarrow \quad \Lambda^T = 2\sqrt{2}\nabla J^T (\nabla J \nabla J^T)^{-1},$$

where $\nabla J \nabla J^T$ is invertible due to the LICQ condition. Therefore, we get optimal $Q = \Pi := \sqrt{2}\left(I - \nabla J^T(\nabla J \nabla J^T)^{-1}\nabla J\right)$. For the $q$ part, we set

$$b := \frac{1}{2}\left[\text{Tr}\left(\nabla^2 h_1 Q Q^T\right) + \alpha h_1, ..., \text{Tr}\left(\nabla^2 g_{i_{|I_x|}} Q Q^T\right) + \alpha(g_{i_{|I_x|}} + \epsilon)\right]^T$$

and the associated Lagrangian function $L(\bar{q}, \lambda) := \|\bar{q} + \nabla f(x)\|^2 + \lambda^T (\nabla J \bar{q} + b)$ with $\lambda \in \mathbb{R}^{m+|I_x|}$ being an Lagrangian multiplier. Then, the stationarity with respect to $\bar{q}$ gives

$$0 = \frac{\partial L}{\partial \bar{q}} = 2(\bar{q} + \nabla f) + \nabla J^T \lambda \quad \Rightarrow \quad \bar{q} = -\nabla f - \nabla J^T \lambda.$$

Again, the constraint condition implies

$$0 = \nabla J \bar{q} + b = -\nabla J \nabla f - (\nabla J \nabla J^T)\lambda + b \quad \Rightarrow \quad \lambda = (\nabla J \nabla J^T)^{-1}[b - \nabla J \nabla f]$$

using the invertibility of $\nabla J \nabla J^T$. By plugging this expression to $\bar{q}$, we recover the optimal $q := -\Pi \nabla f - \nabla J^T (\nabla J \nabla J^T)^{-1}b$. Therefore, the Ito version of closed form SDE is given by

$$dX_t = -[\Pi(X_t)\nabla f(X_t) + \nabla J(X_t)^T G(X_t)^{-1}b(X_t)]dt + \sqrt{2}\Pi(X_t)dW_t$$

where $G := \nabla J \nabla J^T$ is the associated Gram matrix. Also, some tensor-calculus computation (Equation 3.46 in [19]) gives

$$-\nabla J^T G^{-1}\left[\text{Tr}\left(\nabla^2 h_1 \Pi\right), ..., \text{Tr}\left(\nabla^2 h_m \Pi\right), \text{Tr}\left(\nabla^2 g_{i_1} \Pi\right), ..., \text{Tr}\left(\nabla^2 g_{i_{|I_x|}} \Pi\right)\right]^T = \nabla \Pi(x)\Pi(x)$$

which is the Ito-Stratonovich correction term. Furthermore, applying the technique in [19] (Remark 3.17), we can recover that this expression is equal to the mean curvature term $\mathcal{H}(x)$ of a manifold defined by $\Sigma_{I_x} := \left\{x \in \mathbb{R}^d \mid h(x) = 0, g_{I_x}(x) = 0\right\}$. Therefore, we get the following closed form expression of the SDE:

$$dX_t = -[\Pi(X_t)\nabla f(X_t) + \alpha \nabla J(X_t)^T G^{-1}(X_t)J(X_t)]dt + \mathcal{H}(X_t)dt + \sqrt{2}\Pi(X_t), \quad \text{(Ito)}$$

$$dX_t = -[\Pi(X_t)\nabla f(X_t) + \alpha \nabla J(X_t)^T G^{-1}(X_t)J(X_t)]dt + \sqrt{2}\Pi(X_t) \circ dW_t, \quad \text{(Strato.)}$$

where $\mathcal{H} = -\nabla J^T G^{-1}\left[\text{Tr}\left(\nabla^2 h_1 \Pi\right), ..., \text{Tr}\left(\nabla^2 h_m \Pi\right), \text{Tr}\left(\nabla^2 g_{i_1} \Pi\right), ..., \text{Tr}\left(\nabla^2 g_{i_{|I_x|}} \Pi\right)\right]^T$ is the associated Ito-Stratonovich correction term (or mean curvature term of $\Sigma_{I_x}$). $\qquad \square$

**Lemma 1** (Exponential decay of constraint functions). *The dynamics induced by* (12) *satisfies the following properties almost surely for* $\forall i \in [m], \forall j \in I_{X_0}$:

$$h_i(X_t) = h_i(X_0)e^{-\alpha t}, \quad t \geq 0 \tag{13}$$

*and*

$$\begin{cases} g_j(X_t) = -\epsilon + (g_j(X_0) + \epsilon)e^{-\alpha t}, & t \leq \frac{1}{\alpha}\ln\left(\frac{g_j(X_0) + \epsilon}{\epsilon}\right) \\ g_j(X_t) \leq 0, & t \geq \frac{1}{\alpha}\ln\left(\frac{g_j(X_0) + \epsilon}{\epsilon}\right) \end{cases}$$

*with* $g_j(X_t) \leq 0, \forall t \geq 0$ *for* $j \notin I_{X_0}$, *where* $I_x := \{k \in [l] \mid g_k(x) \geq 0\}$ *is the index set of active inequality constraints.*

*Proof.* Observe that, for each $k \in [m]$, the Stratonovich chain rule implies

$$dh_k(X_t) = \nabla h_k(X_t)^T \left[ (-\Pi(X_t)\nabla f(X_t) - \alpha \nabla J(X_t)^T G^{-1}(X_t)J(X_t))dt + \sqrt{2}\Pi(X_t) \circ dW_t \right]$$

$$\overset{(1)}{=} -\alpha \nabla h_k(X_t)^T \nabla J(X_t)^T G^{-1}(X_t)J(X_t)dt$$

$$= -\alpha \sum_{i,j=1}^{m+I_x} [G(X_t)]_{ki}[G^{-1}(X_t)]_{ij}h_j(X_t)dt = -\alpha h_k(X_t)dt,$$

where (1) holds due to the fact $\nabla h_k(x)^T \Pi(x) = 0$. By integrating both side with respect to $t$, we recover $h(X_t) = h(X_0)e^{-\alpha t}$, $t \geq 0$ almost surely. Repeating the same calculation for $g_j$ for $k \in I_x$, we obtain

$$dg_k(X_t) = -\alpha \sum_{i,j=1}^{m+I_x} [G(X_t)]_{ki}[G^{-1}(X_t)]_{ij}(g_j(X_t) + \epsilon)dt = -\alpha(g_k(X_t) + \epsilon)dt,$$

which again recovers $g_k(X_t) = -\epsilon + (g_k(X_0) + \epsilon)e^{-\alpha t}$. Furthermore, once $g_j(X_t) \leq 0$, it is instantaneously reflected into interior of $\Sigma$ whenever it hits the boundary $\partial\Sigma$. Therefore, $g_j(X_t) \leq 0$ holds for $\forall t \geq 0, j \in I_{X_0}$. $\square$

# E   Proof of Theoretical Results - Equality-constraint OLLA

Observe that when the constraints are only equality constraints, the equality-constraint OLLA (3) is given by

$$dX_t = -[\Pi(X_t)\nabla f(X_t) + \alpha \nabla h(X_t)G^{-1}(X_t)h(X_t)]dt + \sqrt{2}\Pi(X_t) \circ dW_t. \tag{14}$$

The high-level proof idea of Theorem 1 is to decompose the convergence analysis into two parts: (1) Convergence of $W_2$ distance between $\rho_t$ and $\tilde{\rho}_t$, (2) Convergence of $\mathsf{KL}^\Sigma$ between $\tilde{\rho}_t$ and $\rho_\Sigma$ where $\rho_t, \tilde{\rho}_t$ are the law of $X_t, Y_t(:= \pi(X_t))$ respectively and $\rho_\Sigma$ is the law of the target distribution, which satisfies $d\rho_\Sigma \propto \exp(-f(x))d\sigma_\Sigma$.

## E.1   Upper Bound of $W_2(\rho_t, \tilde{\rho}_t)$

**Lemma E.1** (Upper bound of $W_2(\rho_t, \tilde{\rho}_t)$). *Let $\rho_t$ be the law of $X_t$ which follows equality-constrained OLLA (14) and define $t_0 := \frac{1}{\alpha}\ln\left(\frac{1}{\delta}\right)$. For $t \geq t_0$, the law $\tilde{\rho}_t$ of $Y_t := \pi(X_t)$ is well-defined and it holds that*

$$W_2(\rho_t, \tilde{\rho}_t) \leq \frac{M_h}{\kappa}e^{-\alpha t}. \tag{15}$$

*Proof.* For $t \geq t_0$, observe that $\|X_t - Y_t\|_2 = \|X_t - \pi(X_t)\|_2 \leq \frac{1}{\kappa}\|h(X_t)\|_2 \leq \frac{M_h}{\kappa}e^{-\alpha t}$ by Lemma C.1 and Lemma 1. Then, by integrating both sides with respect to optimal coupling of $\rho_t$ and $\tilde{\rho}_t$, we get

$$W_2(\rho_t, \tilde{\rho}_t) \leq \left(\mathbb{E}\left[\|X_t - Y_t\|_2^2\right]\right)^{\frac{1}{2}} \overset{(\circ)}{\leq} \mathbb{E}\left[\|X_t - Y_t\|_2\right] \leq \frac{M_h}{\kappa}e^{-\alpha t},$$

where $(\circ)$ holds by Jensen's inequality. $\square$

## E.2 Upper Bound of $\mathsf{KL}^\Sigma(\tilde\rho_t||\rho_\Sigma)$

**Lemma E.2** (SDE representation of projected process). *Let $X_t$ be the stochastic process following the SDE:*

$$dX_t = b(X_t,t)dt + \sqrt{2}\Pi(X_t) \circ dW_t$$

*where $X_t \in \hat{U}_\delta(\Sigma)$, $\forall t \geq 0$. Then, the stochastic process $Y_t$ defined by $Y_t = \pi(X_t)$ follows the SDE below:*

$$dY_t = \Pi(Y_t)b(Y_t,t)dt + \sqrt{2}\Pi(Y_t) \circ dW_t + [\nabla\pi(X_t)b(X_t,t) - \nabla\pi(Y_t)b(Y_t,t)]\,dt$$
$$+ \sqrt{2}[\nabla\pi(X_t)\Pi(X_t) - \nabla\pi(Y_t)\Pi(Y_t)] \circ dW_t$$

*Proof.* From the the Stratonovich chain rule, we observe that

$$dY_t = d\pi(X_t) = \nabla\pi(X_t)b(X_t,t)dt + \sqrt{2}\nabla\pi(X_t)\Pi(X_t) \circ dW_t.$$

This expression can be re-written as follows

$$dY_t = \left[\nabla\pi(Y_t)b(Y_t,t)dt + \sqrt{2}\nabla\pi(Y_t)\Pi(Y_t) \circ dW_t\right] + [\nabla\pi(X_t)b(X_t,t) - \nabla\pi(Y_t)b(Y_t,t)]\,dt$$
$$+ \sqrt{2}[\nabla\pi(X_t)\Pi(X_t) - \nabla\pi(Y_t)\Pi(Y_t)] \circ dW_t$$
$$\overset{(\circ)}{=} \left[\Pi(Y_t)b(Y_t,t)dt + \sqrt{2}\Pi(Y_t) \circ dW_t\right] + [\nabla\pi(X_t)b(X_t,t) - \nabla\pi(Y_t)b(Y_t,t)]\,dt$$
$$+ \sqrt{2}[\nabla\pi(X_t)\Pi(X_t) - \nabla\pi(Y_t)\Pi(Y_t)] \circ dW_t$$

where $(\circ)$ holds because $\nabla\pi(y) = \Pi(y)$, $\Pi(y)^2 = \Pi(y)$ (idempotent) for $\forall y \in \Sigma$ and $Y_t \in \Sigma$. □

**Corollary E.1** (SDE representation of projected process from equality-constrained OLLA). *Let $X_t$ be the stochastic process following equality-constrained OLLA (14). Then, for $t \geq t_0(:= \frac{1}{\alpha}\ln\left(\frac{1}{\delta}\right))$, the projected process $Y_t := \pi(X_t)$ follows the following SDE:*

$$dY_t = [-\Pi(Y_t)\nabla f(Y_t) + b_N(Y_t,t)]\,dt + \sqrt{2}\Pi(Y_t)(I + A_N(Y_t,t)) \circ dW_t$$

*where $\|b_N(Y_t,t)\|_2 = C_{b_N}e^{-\alpha t}, \|A_N(Y_t,t)\| = C_{A_N}e^{-\alpha t}$ for $t \geq 0$ almost surely for some constant $C_{b_N}, C_{A_N} := \frac{C_{L_A}M_h}{\kappa} > 0$ with $C_{L_A}$ being the Lipschitz constant of $\nabla\pi(x)\Pi(x)$ on $\hat{U}_\delta(\Sigma)$*

*Proof.* By applying Lemma E.2 to the SDE (14), it holds that

$$dY_t = \Pi(Y_t)b(Y_t,t)dt + \sqrt{2}\Pi(Y_t) \circ dW_t + [\nabla\pi(X_t)b(X_t,t) - \nabla\pi(Y_t)b(Y_t,t)]\,dt$$
$$+ \sqrt{2}[\nabla\pi(X_t)\Pi(X_t) - \nabla\pi(Y_t)\Pi(Y_t)] \circ dW_t$$

for $b(x,t) = b(x) := -\left[\nabla\Pi(x)\nabla f(x) + \alpha\nabla h(x)^T G^{-1}(x)h(x)\right]$. By using Lemma 1, Theorem C.1, we can set $X_t = \zeta(Y_t, h(X_0)e^{-\alpha t})$ where $\zeta : \Sigma \times \mathbb{R}^m \to \hat{U}_\delta(\Sigma)$ is the recovery map. Now, since $X_t, Y_t \in \hat{U}_\delta(\Sigma)$ and the closure of $\hat{U}_\delta(\Sigma)$ is compact, $\nabla\pi(x)b(x)$ and $\nabla\pi(x)\Pi(x)$ is $C_{L_b}, C_{L_A}$-Lipschitz on $\hat{U}_\delta(\Sigma)$, respectively for some $C_{L_b}, C_{L_A} > 0$. Therefore, it holds that

$$\|b_N(Y_t,t)\|_2 \leq C_{L_b}\|\zeta(Y_t, h(X_0)e^{-\alpha t}) - Y_t\|_2 \leq \frac{C_{L_b}\|h(X_0)\|_2}{\kappa}e^{-\alpha t} \leq \frac{C_{L_b}M_h}{\kappa}e^{-\alpha t}$$

where $b_N(Y_t,t) := \nabla\pi(\zeta(Y_t, h(X_0)e^{-\alpha t})b(\zeta(Y_t, h(X_0)e^{-\alpha t})) - \nabla\pi(Y_t)b(Y_t)$ and the second last inequality comes from Lemma C.1. Similarly, we obtain the bound of $A_N(Y_t,t) := \nabla\pi(\zeta(Y_t, h(X_0)e^{-\alpha t}))\Pi(\zeta(Y_t, h(X_0)e^{-\alpha t})) - \nabla\pi(Y_t)\Pi(Y_t)$ as follows:

$$\|A_N(Y_t,t)\|_2 \leq C_{L_A}\|\zeta(Y_t, h(X_0)e^{-\alpha t}) - Y_t\|_2 \leq \frac{C_{L_A}\|h(X_0)\|_2}{\kappa}e^{-\alpha t} \leq \frac{C_{L_A}M_h}{\kappa}e^{-\alpha t}$$

Finally, we complete the proof by setting $C_{b_N} := \frac{C_{L_b}M_h}{\kappa}, C_{A_N} := \frac{C_{L_A}M_h}{\kappa}$ and observing that $\nabla\pi(x) = \Pi(\pi(x))\nabla\pi(x)$ for $\forall x \in \hat{U}_\delta(\Sigma)$, which implies $A_N(Y_t,t) = \Pi(Y_t)A_N(Y_t,t)$. □

The following theorem is a Fokker-Planck equation of a Stratonovich SDE defined on a Riemannian manifold. We will rely on this theorem to describe the time derivative of $\tilde\rho_t$.

**Theorem E.1** (Fokker-Planck equation on Riemannian manifold [51, 52]). *Let $X_t \in \Sigma$ be a stochastic process following the SDE:*

$$dX_t = V_0 dt + \sum_{k=1}^{d} V_k \circ dB_t^k,$$

*where $V_0, V_k$ are smooth vector fields on $\Sigma$ for each $k \in [d]$ and $B_t^k$ are kth components of Brownian motion $B_t$. Then, the law $\rho_t$ of the stochastic process $X_t$ satisfies the following Fokker-Planck equation:*

$$\partial_t \rho_t = -\text{div}_\Sigma(\rho_t V_0) + \frac{1}{2}\sum_{k=1}^{d} \text{div}_\Sigma(\text{div}_\Sigma(\rho_t V_k)V_k).$$

**Lemma E.3** (Upper bound of $\text{KL}^\Sigma(\tilde{\rho}_t || \rho_\Sigma)$). *Assume that $\rho_\Sigma$ satisfies the LSI condition with constant $\lambda_{LSI}$. Let $X_t$ be the stochastic process following equality-constrained OLLA (14) and $\tilde{\rho}_t$ be the law of $Y_t := \pi(X_t)$ after $t \geq t_{cut}$, $t_{cut} := \max\left\{\frac{1}{\alpha}\ln\delta, \frac{1}{\alpha}\ln(C_5)\right\}$. Then, for $\alpha \neq 2\lambda_{LSI}$, the following non-asymptotic convergence rate of $\text{KL}^\Sigma(\tilde{\rho}_t || \rho_\Sigma)$ can be obtained as follows*

$$\text{KL}^\Sigma(\tilde{\rho}_t || \rho_\Sigma) \leq \exp\left(-2\lambda_{LSI}(t - t_{cut}) - \frac{2\lambda_{LSI}C_5}{\alpha}(e^{-\alpha t} - e^{-\alpha t_{cut}})\right)\left[\text{KL}^\Sigma(\tilde{\rho}_{t_{cut}} || \rho_\Sigma)\right.$$

$$\left. + C_6 \int_{t_{cut}}^{t} \exp\left(2\lambda(s - t_{cut}) + \frac{2\lambda_{LSI}C_5}{\alpha}(e^{-\alpha s} - e^{-\alpha t_{cut}})\right)e^{-\alpha s}ds\right]$$

*In particular, if $\alpha > 2\lambda_{LSI}$, it becomes*

$$\text{KL}^\Sigma(\tilde{\rho}_t || \rho_\Sigma) \leq \exp\left(-2\lambda_{LSI}(t - t_{cut}) - \frac{2\lambda_{LSI}C_5}{\alpha}(e^{-\alpha t} - e^{-\alpha t_{cut}})\right)\left[\text{KL}^\Sigma(\tilde{\rho}_{t_{cut}} || \rho_\Sigma) + C_7\right]$$

*for some constants $C_5 = \mathcal{O}(1 + C_{A_N} + C_{A_N}^2), C_6, C_7 := \frac{C_6 e^{-\alpha t_{cut}}}{\alpha - 2\lambda_{LSI}} > 0$.*

*Proof.* By Theorem E.1, Corollary E.1, and the choice of $\nabla f = -\nabla \ln \rho_\Sigma$, we know that the projected process $Y_t$ is given by

$$dY_t = [-\Pi(Y_t)\nabla f(Y_t) + b_N(Y_t, t)]\,dt + \sqrt{2}\Pi(Y_t)(I + A_N(Y_t, t)) \circ dW_t$$

and its associated Fokker-Planck equation can be written as follows:

$$\partial_t \tilde{\rho}_t = -\text{div}_\Sigma(\tilde{\rho}_t(\nabla_\Sigma \ln \rho_\Sigma + b_N)) + \sum_{k=1}^{d} \text{div}_\Sigma(\text{div}_\Sigma(\tilde{\rho}_t(f_k + \delta_k))(f_k + \delta_k))$$

where $f_k = \Pi e_k, \delta_k = \Pi A_N e_k$, and $e_k$ is kth standard basis vector for $\mathbb{R}^d$. Now observe the following equations:

$$\partial_t \text{KL}^\Sigma(\tilde{\rho}_t || \rho_\Sigma) = \int_\Sigma \partial_t \tilde{\rho}_t \cdot \ln\left(\frac{\tilde{\rho}_t}{\rho_\Sigma}\right) d\sigma_\Sigma + \partial_t \int \tilde{\rho}_t d\sigma_\Sigma = \int_\Sigma \partial \tilde{\rho}_t \ln\left(\frac{\tilde{\rho}_t}{\rho_\Sigma}\right) d\sigma_\Sigma$$

$$= \underbrace{\int_\Sigma \left[-\text{div}_\Sigma(\tilde{\rho}_t \nabla_\Sigma \ln \rho_\Sigma) + \sum_{k=1}^{d} \text{div}_\Sigma(\text{div}_\Sigma(\tilde{\rho}_t f_k)f_k)\right] \ln\left(\frac{\tilde{\rho}_t}{\rho_\Sigma}\right) d\sigma_\Sigma}_{\text{Term (1)}}$$

$$+ \underbrace{\int_\Sigma [-\text{div}_\Sigma(\tilde{\rho}_t b_N)] \ln\left(\frac{\tilde{\rho}_t}{\rho_\Sigma}\right) d\sigma_\Sigma}_{\text{Term (2)}}$$

$$+ \underbrace{\int_\Sigma \sum_{k=1}^{d} [\text{div}_\Sigma(\text{div}_\Sigma(\tilde{\rho}_t \delta_k)f_k + \text{div}_\Sigma(\tilde{\rho}_t f_k)\delta_k + \text{div}_\Sigma(\tilde{\rho}_t \delta_k)\delta_k] \ln\left(\frac{\tilde{\rho}_t}{\rho_\Sigma}\right) d\sigma_\Sigma}_{\text{Term (3)}}.$$

**Analysis of Term (1) - induced by the main SDE.** From integration by parts, we obtain

$$\text{Term (1)} = \int_\Sigma \tilde{\rho}_t \langle \nabla_\Sigma \ln \rho_\Sigma, \nabla_\Sigma \ln \left( \frac{\tilde{\rho}_t}{\rho_\Sigma} \right) \rangle d\sigma_\Sigma - \sum_{k=1}^d \int_\Sigma \text{div}_\Sigma(\tilde{\rho}_t f_k) \langle f_k, \nabla_\Sigma \ln \left( \frac{\tilde{\rho}_t}{\rho_\Sigma} \right) \rangle d\sigma_\Sigma$$

$$= \int_\Sigma \tilde{\rho}_t \langle \nabla_\Sigma \ln \rho_\Sigma, \nabla_\Sigma \ln \left( \frac{\tilde{\rho}_t}{\rho_\Sigma} \right) \rangle d\sigma_\Sigma - \sum_{k=1}^d \int_\Sigma \langle \nabla_\Sigma \tilde{\rho}_t, f_k \rangle \langle f_k, \nabla_\Sigma \ln \left( \frac{\tilde{\rho}_t}{\rho_\Sigma} \right) \rangle d\sigma_\Sigma$$

$$- \sum_{k=1}^d \int_\Sigma \tilde{\rho}_t \text{div}_\Sigma(f_k) \langle f_k, \nabla_\Sigma \ln \left( \frac{\tilde{\rho}_t}{\rho_\Sigma} \right) \rangle d\sigma_\Sigma$$

$$\overset{(\triangle)}{=} \int_\Sigma \tilde{\rho}_t \langle \nabla_\Sigma \ln \rho_\Sigma, \nabla_\Sigma \ln \left( \frac{\tilde{\rho}_t}{\rho_\Sigma} \right) \rangle d\sigma_\Sigma - \int_\Sigma \tilde{\rho}_t \langle \nabla_\Sigma \ln \tilde{\rho}_t, \nabla_\Sigma \ln \left( \frac{\tilde{\rho}_t}{\rho_\Sigma} \right) \rangle d\sigma_\Sigma$$

$$= - \int_\Sigma \tilde{\rho}_t \| \nabla_\Sigma \ln \left( \frac{\tilde{\rho}_t}{\rho_\Sigma} \right) \|_2^2 d\sigma_\Sigma = -I^\Sigma(\tilde{\rho}_t \| \rho_\Sigma)$$

where $(\triangle)$ holds using Lemma E.4 (the third term = 0) and the fact that $\tilde{\rho}_t \nabla_\Sigma \ln \tilde{\rho}_t = \nabla_\Sigma \tilde{\rho}_t$.

**Analysis of Term (2) - induced by the noise drift** $b_N$. Again, using the integration by parts, we observe that

$$|\text{Term (2)}| = \left| \int_\Sigma \text{div}_\Sigma(\tilde{\rho}_t b_N) \ln \left( \frac{\tilde{\rho}_t}{\rho_\Sigma} \right) d\sigma_\Sigma \right| = \left| \int_\Sigma \tilde{\rho}_t \langle b_N, \nabla_\Sigma \ln \left( \frac{\tilde{\rho}_t}{\rho_\Sigma} \right) \rangle d\sigma_\Sigma \right|$$

$$\leq \int_\Sigma \tilde{\rho}_t \| b_N \|_2 \| \nabla_\Sigma \ln \left( \frac{\tilde{\rho}_t}{\rho_\Sigma} \right) \|_2 d\sigma_\Sigma \leq C_{b_N} e^{-\alpha t} \int_\Sigma \tilde{\rho}_t \cdot 1 \cdot \| \nabla_\Sigma \ln \left( \frac{\tilde{\rho}_t}{\rho_\Sigma} \right) \|_2 d\sigma_\Sigma$$

$$\overset{(\square)}{\leq} C_{b_N} e^{-\alpha t} \int_\Sigma \tilde{\rho}_t \left[ \frac{C_{b_N}}{4} + \frac{1}{C_{b_N}} \| \nabla_\Sigma \ln \left( \frac{\tilde{\rho}_t}{\rho_\Sigma} \right) \|_2^2 \right] d\sigma_\Sigma = e^{-\alpha t} I^\Sigma(\tilde{\rho}_t \| \rho_\Sigma) + \frac{C_{b_N}^2}{4} e^{-\alpha t},$$

where $(\square)$ inequality holds using the AM-GM inequality.

**Analysis of Term (3) - induced by noise diffusion** $A_N$. To analyze Term (3), we apply integration by parts and the chain rule of the divergence, i.e., $\text{div}_\Sigma(\tilde{\rho}_t a_k) = \langle \nabla_\Sigma \tilde{\rho}_t, a_k \rangle + \tilde{\rho}_t \text{div}_\Sigma(a_k)$ for a vector field $a_k$ on $\Sigma$:

$$\text{Term (3)} = - \sum_{k=1}^d \int_\Sigma \tilde{\rho}_t \langle (\text{div}_\Sigma \delta_k) \delta_k, \nabla_\Sigma \ln \left( \frac{\tilde{\rho}_t}{\rho_\Sigma} \right) \rangle d\sigma_\Sigma - \sum_{k=1}^d \int_\Sigma \tilde{\rho}_t \langle (\text{div}_\Sigma \delta_k) f_k, \nabla_\Sigma \ln \left( \frac{\tilde{\rho}_t}{\rho_\Sigma} \right) \rangle d\sigma_\Sigma$$

$$- \sum_{k=1}^d \int_\Sigma \tilde{\rho}_t \langle (\text{div}_\Sigma f_k) \delta_k, \nabla_\Sigma \ln \left( \frac{\tilde{\rho}_t}{\rho_\Sigma} \right) \rangle d\sigma_\Sigma - \sum_{k=1}^d \int_\Sigma \langle \nabla_\Sigma \tilde{\rho}_t, \delta_k \rangle \langle f_k, \nabla_\Sigma \ln \left( \frac{\tilde{\rho}_t}{\rho_\Sigma} \right) \rangle d\sigma_\Sigma$$

$$- \sum_{k=1}^d \int_\Sigma \langle \nabla_\Sigma \tilde{\rho}_t, f_k \rangle \langle \delta_k, \nabla_\Sigma \ln \left( \frac{\tilde{\rho}_t}{\rho_\Sigma} \right) \rangle d\sigma_\Sigma - \sum_{k=1}^d \int_\Sigma \langle \nabla_\Sigma \tilde{\rho}_t, \delta_k \rangle \langle \delta_k, \nabla_\Sigma \ln \left( \frac{\tilde{\rho}_t}{\rho_\Sigma} \right) \rangle d\sigma_\Sigma.$$

Now, note that $\sum_{k=1}^d \delta_k f_k^T = \Pi A_N \Pi$, $\sum_{k=1}^d f_k \delta_k^T = \Pi A_N^T \Pi$, and $\sum_{k=1}^d \delta_k \delta_k^T = \Pi A_N A_N^T \Pi$. Then it holds that

$$\left| - \sum_{k=1}^d \int_\Sigma \langle \nabla_\Sigma \tilde{\rho}_t, \delta_k \rangle \langle f_k, \nabla_\Sigma \ln \left( \frac{\tilde{\rho}_t}{\rho_\Sigma} \right) \rangle d\sigma_\Sigma \right| = \left| \int_\Sigma \tilde{\rho}_t (\nabla_\Sigma \ln \tilde{\rho}_t)^T \Pi A_N \Pi (\nabla_\Sigma \ln \left( \frac{\tilde{\rho}_t}{\rho_\Sigma} \right)) d\sigma_\Sigma \right|$$

$$\leq C_{A_N} e^{-\alpha t} \int_\Sigma \tilde{\rho}_t \| \nabla_\Sigma \ln \tilde{\rho}_t \|_2 \| \nabla_\Sigma \ln \left( \frac{\tilde{\rho}_t}{\rho_\Sigma} \right) \|_2 d\sigma_\Sigma$$

$$\overset{(\times)}{\leq} C_{A_N} e^{-\alpha t} I^\Sigma(\tilde{\rho}_t \| \rho_\Sigma) + C_{A_N} C_3 e^{-\alpha t} \int_\Sigma \tilde{\rho}_t \| \nabla_\Sigma \ln \left( \frac{\tilde{\rho}_t}{\rho_\Sigma} \right) \|_2 d\sigma_\Sigma$$

$$\overset{(\circ)}{\leq} C_{A_N} e^{-\alpha t} I^\Sigma(\tilde{\rho}_t \| \rho_\Sigma) + C_{A_N} C_3 e^{-\alpha t} \int_\Sigma \tilde{\rho}_t \left[ \frac{C_3}{4} + \frac{1}{C_3} \| \nabla_\Sigma \ln \left( \frac{\tilde{\rho}_t}{\rho_\Sigma} \right) \|_2^2 \right] d\sigma_\Sigma$$

$$\leq 2 C_{A_N} e^{-\alpha t} I^\Sigma(\tilde{\rho}_t \| \rho_\Sigma) + \frac{C_{A_N} C_3^2}{4} e^{-\alpha t},$$

where $C_3 := \max_{x\in\Sigma}\|\nabla_\Sigma \ln\rho_\Sigma\|_2 < \infty$ by compactness of $\Sigma$, and $(\times)$ holds using the triangle inequality; $\|\nabla_\Sigma \ln\tilde{\rho}_t\|_2 \leq \|\nabla_\Sigma \ln\left(\frac{\tilde{\rho}_t}{\rho_\Sigma}\right)\|_2 + \|\nabla_\Sigma \ln\rho_\Sigma\|_2$. Also, $(\circ)$ comes from the AM-GM inequality. By following the same reasoning, the following inequalities are obtained:

$$\left|-\sum_{k=1}^{d}\int_\Sigma \langle\nabla_\Sigma\tilde{\rho}_t, f_k\rangle\langle\delta_k, \nabla_\Sigma\ln\left(\frac{\tilde{\rho}_t}{\rho_\Sigma}\right)\rangle d\sigma_\Sigma\right| \leq 2C_{A_N}e^{-\alpha t}I^\Sigma(\tilde{\rho}_t\|\rho_\Sigma) + \frac{C_{A_N}C_3^2}{4}e^{-\alpha t}$$

$$\left|-\sum_{k=1}^{d}\int_\Sigma \langle\nabla_\Sigma\tilde{\rho}_t, \delta_k\rangle\langle\delta_k, \nabla_\Sigma\ln\left(\frac{\tilde{\rho}_t}{\rho_\Sigma}\right)\rangle d\sigma_\Sigma\right| \leq 2C_{A_N}^2 e^{-2\alpha t}I^\Sigma(\tilde{\rho}_t\|\rho_\Sigma) + \frac{C_{A_N}^2 C_3^2}{4}e^{-2\alpha t}$$

$$\leq 2C_{A_N}^2 e^{-\alpha t}I^\Sigma(\tilde{\rho}_t\|\rho_\Sigma) + \frac{C_{A_N}^2 C_3^2}{4}e^{-\alpha t}$$

Next, we observe that the following terms decay exponentially fast :

$$\left|\sum_{k=1}^{d}\int_\Sigma \tilde{\rho}_t\langle\mathsf{div}_\Sigma(f_k)\delta_k, \nabla_\Sigma\ln\left(\frac{\tilde{\rho}_t}{\rho_\Sigma}\right)\rangle d\sigma_\Sigma\right| \leq \int_\Sigma \tilde{\rho}_t\|\sum_{k=1}^{d}\mathsf{div}_\Sigma(f_k)\delta_k\|_2\|\nabla_\Sigma\ln\left(\frac{\tilde{\rho}_t}{\rho_\Sigma}\right)\|_2 d\sigma_\Sigma$$

$$\stackrel{(\oplus)}{\leq} \int_\Sigma \tilde{\rho}_t\|\Pi A_N\|_2\|\mathsf{div}_\Sigma(\Pi)\|_2\|\nabla_\Sigma\ln\left(\frac{\tilde{\rho}_t}{\rho_\Sigma}\right)\|_2 d\sigma_\Sigma$$

$$\leq C_{A_N}C_4 e^{-\alpha t}\int_\Sigma \tilde{\rho}_t 1\cdot\|\nabla_\Sigma\ln\left(\frac{\tilde{\rho}_t}{\rho_\Sigma}\right)\|_2 d\sigma_\Sigma$$

$$\leq C_{A_N}C_4 e^{-\alpha t}\int_\Sigma \tilde{\rho}_t\left[\frac{C_{A_N}C_4}{4} + \frac{1}{C_{A_N}C_4}\|\nabla_\Sigma\ln\left(\frac{\tilde{\rho}_t}{\rho_\Sigma}\right)\|_2^2\right]d\sigma_\Sigma$$

$$\leq e^{-\alpha t}I^\Sigma(\tilde{\rho}_t\|\rho_\Sigma) + \frac{C_{A_N}^2 C_4^2}{4}e^{-\alpha t},$$

where $C_4 := \max_{x\in\Sigma}\|\mathsf{div}_\Sigma(\Pi(x))\|_2 < \infty$, $(\oplus)$ holds because $\sum_{k=1}^{d}\mathsf{div}_\Sigma(f_k)\delta_k = \mathsf{div}_\Sigma(\Pi)^T\Pi A_N$ once $\mathsf{div}_\Sigma(\Pi)$ is given by $(\mathsf{div}_\Sigma\Pi)_k := \mathsf{div}_\Sigma(f_k)$ for each $k\in[d]$. Similarly, we obtain the following bound by using Lemma E.5 :

$$\left|\sum_{k=1}^{d}\int_\Sigma \tilde{\rho}_t\langle\mathsf{div}_\Sigma(\delta_k)f_k, \nabla_\Sigma\ln\left(\frac{\tilde{\rho}_t}{\rho_\Sigma}\right)\rangle d\sigma_\Sigma\right| \leq \int_\Sigma \tilde{\rho}_t\|\Pi\|_2\|\mathsf{div}(\Pi A_N)\|_2\|\nabla_\Sigma\ln\left(\frac{\tilde{\rho}_t}{\rho_\Sigma}\right)\|d\sigma_\Sigma$$

$$\stackrel{\text{Lem}E.5}{\leq} C_{div}e^{-\alpha t}\int_\Sigma \tilde{\rho}_t\|\nabla_\Sigma\ln\left(\frac{\tilde{\rho}_t}{\rho_\Sigma}\right)\|_2 d\sigma_\Sigma \leq e^{-\alpha t}I^\Sigma(\tilde{\rho}_t\|\rho_\Sigma) + \frac{C_{div}^2}{4}e^{-\alpha t}$$

and

$$\left|\sum_{k=1}^{d}\int_\Sigma \tilde{\rho}_t\langle\mathsf{div}_\Sigma(\delta_k)\delta_k, \nabla_\Sigma\ln\left(\frac{\tilde{\rho}_t}{\rho_\Sigma}\right)\rangle d\sigma_\Sigma\right| \leq \int_\Sigma \tilde{\rho}_t\|\Pi A_N\|_2\|\mathsf{div}(\Pi A_N)\|_2\|\nabla_\Sigma\ln\left(\frac{\tilde{\rho}_t}{\rho_\Sigma}\right)\|d\sigma_\Sigma$$

$$\stackrel{\text{Lem}E.5}{\leq} C_{div}C_{A_N}e^{-2\alpha t}\int_\Sigma \tilde{\rho}_t\|\nabla_\Sigma\ln\left(\frac{\tilde{\rho}_t}{\rho_\Sigma}\right)\|d\sigma_\Sigma \leq e^{-\alpha t}I^\Sigma(\tilde{\rho}_t\|\rho_\Sigma) + \frac{C_{div}^2 C_{A_N}^2}{4}e^{-\alpha t},$$

where $\mathsf{div}_\Sigma(\Pi A_N)$ is similarly defined by $(\mathsf{div}_\Sigma\Pi A_N)_k := \mathsf{div}_\Sigma(\delta_k)$ for each $k\in[d]$.

**Applying Gronwall-type inequality.** By summing all the bounds, we arrive at

$$\partial_t\mathsf{KL}^\Sigma(\tilde{\rho}_t\|\rho_\Sigma) \stackrel{\text{LSI}}{\leq} -2\lambda_{\text{LSI}}(1 - C_5 e^{-\alpha t})\mathsf{KL}^\Sigma(\tilde{\rho}_t\|\rho_\Sigma) + C_6 e^{-\alpha t},$$

with $C_5 := (4 + 4C_{A_N} + 2C_{A_N}^2)$, $C_6 := \left(\frac{C_{b_N}^2}{4} + \frac{C_{A_N}C_3^2}{2} + \frac{C_{A_N}^2 C_3^2}{4} + \frac{C_{A_N}^2 C_4^2}{4} + \frac{C_{div}^2}{4} + \frac{C_{div}^2 C_{A_N}^2}{4}\right)$.

Therefore, the Grönwall-type inequality gives for $t > t_{\text{cut}}, t_{\text{cut}} := \max\left\{\frac{1}{\alpha}\ln\delta, \frac{1}{\alpha}\ln(C_5)\right\}$:

$$\mathsf{KL}^\Sigma(\tilde{\rho}_t\|\rho_\Sigma) \leq \exp\left(-2\lambda_{\text{LSI}}(t - t_{\text{cut}}) - \frac{2\lambda_{\text{LSI}}C_5}{\alpha}(e^{-\alpha t} - e^{-\alpha t_{\text{cut}}})\right)[\mathsf{KL}^\Sigma(\tilde{\rho}_{t_{\text{cut}}}\|\rho_\Sigma)$$

$$+ C_6\int_{t_{\text{cut}}}^{t}\exp\left(2\lambda_{\text{LSI}}(s - t_{\text{cut}}) + \frac{2\lambda_{\text{LSI}}C_5}{\alpha}(e^{-\alpha s} - e^{-\alpha t_{\text{cut}}})\right)e^{-\alpha s}ds].$$

In particular, if $\alpha > 2\lambda_{\text{LSI}}$, it holds that

$$\mathsf{KL}^\Sigma(\tilde\rho_t||\rho_\Sigma) \le \exp\left(-2\lambda_{\text{LSI}}(t - t_{\text{cut}}) - \frac{2\lambda_{\text{LSI}}C_5}{\alpha}(e^{-\alpha t} - e^{-\alpha t_{\text{cut}}})\right)[\mathsf{KL}^\Sigma(\tilde\rho_{t_{\text{cut}}}||\rho_\Sigma) + C_7]$$

where $C_7 := \frac{e^{-\alpha t_{\text{cut}}}}{\alpha - 2\lambda_{\text{LSI}}}$ from the fact that

$$\int_{t_{\text{cut}}}^\infty \exp\left(2\lambda_{\text{LSI}}(s - t_{\text{cut}}) + \frac{2\lambda_{\text{LSI}}C_5}{\alpha}(e^{-\alpha s} - e^{-\alpha t_{\text{cut}}})\right)e^{-\alpha s}ds \le \int_{t_{\text{cut}}}^\infty \exp(2\lambda_{\text{LSI}}(s - t_{\text{cut}}))e^{-\alpha s}ds$$

$$= \frac{e^{-\alpha t_{\text{cut}}}}{\alpha - 2\lambda_{\text{LSI}}} < \infty.$$

$\square$

**Lemma E.4.** *Let $\{f_k\}_{k=1}^d$ be a set of vectors defined by $f_k = \Pi(x)e_k$, where $\Pi(x)$ is the orthogonal projector onto $T_x\Sigma$ and $e_k$ is the kth standard basis vector of $\mathbb{R}^d$. Then, it holds that*

$$\sum_{k=1}^d (\mathsf{div}_\Sigma f_k)f_k = 0.$$

*Proof.* Recalling that $\Pi(x) = I - \nabla h(x)^T(\nabla h(x)\nabla h(x)^T)^{-1}\nabla h(x)$, we define $N(x) = \nabla h(x)^T(\nabla h(x)\nabla h(x)^T)^{-\frac{1}{2}} \in \mathbb{R}^{d\times m}$ so that $N(x)^T N(x) = I_m$ and $\Pi(x) = I - N(x)N(x)^T$. If we let the columns of $N(x)$ to be $\{n_1(x), ..., n_m(x)\}$, then these produce an orthonormal basis of $N_x\Sigma$. This is because $\text{Im}(N(x)) = \text{Im}(\nabla h(x)^T)$ (from the invertibility $(\nabla h(x)\nabla h(x)^T)^{-\frac{1}{2}}$) implies $\{n_1(x), ...n_m(x)\}$ span $N_x\Sigma$ and $N(x)^T N(x) = I$ guarantees the orthonormality.

Next, we define a vector field $F(x)$ by $F(x) = \Pi(x)\mathsf{div}_\Sigma(\Pi(x))$ where $(\mathsf{div}_\Sigma\Pi(x))_k := \mathsf{div}_\Sigma(f_k(x))$ for each $k \in [d]$. With this definition, we have $\mathsf{div}_\Sigma\Pi = -\mathsf{div}_\Sigma(NN^T) = -\sum_{k=1}^m \mathsf{div}_\Sigma(n_k n_k^T)$. Now observe that for $l \in [d]$,

$$(\mathsf{div}_\Sigma(n_k n_k^T))_l = \mathsf{Tr}\left(\Pi\nabla((n_k n_k^T)_l)\right) = \sum_{i,j=1}^d \Pi_{ij}\partial_j(n_k n_k^T)_{il} = \sum_{i,j}[\Pi_{ij}\partial_j n_{ki}n_{kl} + \Pi_{ij}n_{ki}\partial_j n_{kl}]$$

$$= (\mathsf{div}_\Sigma n_k)n_{kl} + \sum_{j=1}^d \underbrace{(n_k\Pi)_j}_{=0}\partial_j n_{kl} = (\mathsf{div}_\Sigma n_k)n_{kl}$$

where $n_{kl}$ is the $l$th component of $n_k$. From this fact, we have the following result:

$$\mathsf{div}_\Sigma\Pi = -\sum_{k=1}^d \mathsf{div}_\Sigma(n_k n_k^T) = -\sum_{k=1}^d (\mathsf{div}_\Sigma n_k)n_k \implies F = \Pi\mathsf{div}_\Sigma(\Pi) = 0.$$

Finally, the definition of $F$ gives $\sum_{k=1}^d \mathsf{div}_\Sigma(f_k)f_k = F$, which is zero by the above argument. $\square$

**Lemma E.5.** *Let $\tilde\rho_t$ be the law of the projected process $Y_t$ of $X_t$, where $X_t$ follows equality-constrained OLLA. Define $\delta_k(t, x) := \Pi(x)A_N(x, t)e_k$ and denote $\mathsf{div}_\Sigma(\Pi A_N)$ as a vector in $\mathbb{R}^d$ such that $(\mathsf{div}_\Sigma\Pi A_N)_k := \mathsf{div}_\Sigma(\delta_k)$ for each $k \in [d]$. Then, it holds almost surely*

$$\|\mathsf{div}_\Sigma(\Pi(Y_t)A_N(Y_t, t))\|_2 \le C_{div}e^{-\alpha t}$$

*for $t \ge t_0(:= \frac{1}{\alpha}\ln\left(\frac{1}{\delta}\right))$ and some constant $C_{div} > 0$.*

*Proof.* First, for each $k \in [d]$, observe that $\nabla\delta_k = \nabla\Pi(y)A_N(y, t)e_k + \Pi(y)\nabla A_N(y, t)e_k$ and

$$\mathsf{div}_\Sigma(\delta_k(y)) = \underbrace{\mathsf{Tr}\left(\Pi(y)\nabla\Pi(y)A_N(y, t)e_k\right)}_{\text{Term (1)}} + \underbrace{\mathsf{Tr}\left(\Pi(y)\nabla A_N(y, t)e_k\right)}_{\text{Term (2)}} \qquad (16)$$

where, for a matrix-valued function $G(y)$, $\nabla G(y)$ is the third-order tensor defined by $(\nabla G_{ij}(y))_{ijk} = \frac{\partial G_{ij}(y)}{\partial y_k}$ for $i, j, k \in [d]$, and the gradient $\nabla$ is taken over $y$.

For the Term (1), we know that when $y = Y_t, t \geq t_0$,

$$|\text{Term (1)}| \leq \|\Pi(\nabla\Pi)(A_N e_k)\|_F \leq \|(\nabla\Pi)(A_N e_k)\|_F \leq K_1 \|A_N e_k\|_2 \leq K_1 C_{A_N} e^{-\alpha t} \quad a.s.$$

where $K_1 := \sup_{y \in \Sigma, \|v\|_2 = 1} \|\nabla\Pi(y)v\|_F < \infty$.

For the Term (2), recall that $A_N(y, t) := \nabla\pi(\zeta(y, h(X_0)e^{-\alpha t}))\Pi(\zeta(y, h(X_0)e^{-\alpha t})) - \nabla\pi(y)\Pi(y)$ conditionally on $X_0$, from [Corollary E.1](). For the notational convenience, we define $\eta(y, t) := \zeta(y, h(X_0)e^{-\alpha t})$. Then, it follows that

$$\begin{aligned}
\nabla A_N(y, t) &= \nabla^2\pi(\eta(y, t))\nabla\eta(y, t)\Pi(\eta(y, t)) + \nabla\pi(\eta(y, t))\nabla\Pi(\eta(y, t))\nabla\eta(y, t) \\
&\quad - \nabla^2\pi(y)\Pi(y) - \nabla\pi(y)\nabla\Pi(y) \\
&= \nabla^2\pi(\eta(y, t))(\nabla\eta(y, t) - I)\Pi(\eta(y, t)) + \nabla\pi(\eta(y, t))\nabla\Pi(\eta(y, t))(\nabla\eta(y, t) - I) \\
&\quad + \nabla^2\pi(\eta(y, t))\Pi(\eta(y, t)) - \nabla^2\pi(y)\Pi(y) + \nabla\pi(\eta(y, t))\nabla\Pi(\eta(y, t)) - \nabla\pi(y)\nabla\Pi(y).
\end{aligned}$$

At this moment, from the recovery map $\zeta$ in [Theorem C.1](), the integral form of the remainder gives

$$\|\nabla\eta(y, t) - I\|_2 = \|\nabla(\zeta(y, p) - \zeta(y, 0))\|_2 = \|\nabla_y\zeta(y, p) - \nabla_y\zeta(y, 0)\|_2 \leq K_2\|p\|_2,$$

where $p := h(X_0)e^{-\alpha t}$, $K_2 := \sup_{(y,p) \in \Sigma \times B(0,\delta)} \|\nabla_p\nabla_y\zeta(y, p)\|_2 < \infty$.

By combining these results with the previous expression of $\nabla A_N(y, t)$, we get

$$\|\nabla A_N(Y_t, t)\|_2 \leq D_4 e^{-\alpha t}$$

for some $D_4 > 0$, using the boundedness of $\|\nabla^2\pi\|_2, \|\nabla\pi\|_2, \|\nabla\Pi\|_2$ on $\hat{U}_\delta$, the Lipschitzness of $(\nabla^2\pi)\Pi, \nabla\pi\nabla\Pi$ on $\hat{U}_\delta$, and the contraction of $\|\eta(Y_t, t) - Y_t)\|_2 \leq \frac{M_h}{\kappa} e^{-\alpha t}$. Finally, when $y = Y_t$, we get the following upper bound of Term (2) by applying the previous result:

$$|\text{Term (2)}| \leq \|\Pi\nabla A_N e_k\|_F \overset{(\circ)}{\leq} \sqrt{d-m}\|\nabla A_N e_k\|_2 = \sqrt{d-m}D_4 e^{-\alpha t} = D_5 e^{-\alpha t} \quad a.s.$$

where $(\circ)$ comes from the fact that $\|\Pi(y)\|_F^2 = \text{Tr}(\Pi(y)) = \text{rank}(\Pi(y)) = d - m$ and $D_5 := \sqrt{d-m}D_4$. Therefore, by combining results for Term (1) and Term (2), we obtain

$$|\text{div}_\Sigma(\delta_k(Y_t))| \leq D_6 e^{-\alpha t} \quad \Rightarrow \quad \|\text{div}_\Sigma(\Pi(Y_t)A_N(Y_t, t))\|_2 = \sqrt{\sum_{k=1}^{d}(\text{div}_\Sigma(\delta_k(Y_t)))^2} \leq D_7 e^{-\alpha t}$$

for $t \geq t_0$ and $D_6 := K_1 C_{A_N} + D_5, D_7 := \sqrt{d}D_6$. Because the final result holds without any dependency on $X_0$, the result holds almost surely without conditioning on $X_0$. $\quad\square$

**Lemma E.6** (LSI implies Talagrand inequality [43, 53, 19]). *For the probability measures $\mu, \nu$ defined on a smooth complete Riemannian manifold $\Sigma$, define the $W_2^\Sigma$ distance between $\mu$ and $\nu$ in $\Sigma$ by*

$$W_2^\Sigma(\mu, \nu) := \left( \inf_{\pi \in \Pi(\mu, \nu)} \int_{\Sigma \times \Sigma} d_\Sigma(p, q)^2 \pi(dp, dq) \right)^{\frac{1}{2}}$$

*where $\Pi(\mu, \nu)$ denotes the set of coupling probability measures of $\mu, \nu$, and $d_\Sigma$ denotes the geodesic distance on $\Sigma$ so that for $p, q \in \Sigma$*

$$d_\Sigma(p, q) := \inf \left\{ \left( \int_0^1 \|\dot{\gamma}(t)\|_g^2 dt \right)^{\frac{1}{2}} \mid \gamma \in C^1([0, 1], \Sigma), \gamma(0) = p, \gamma(1) = q \right\}$$

*with $\|\cdot\|_g$ being the induced metric on $\Sigma$. Then, the probability $\nu$ is said to satisfy the Talagrand inequality $(T)$ with constant $\lambda_T > 0$ if for all probability measures $\mu$ with $\mu \ll \nu$, it holds that*

$$W_2^\Sigma(\mu, \nu) \leq \sqrt{\frac{2}{\lambda_T} \mathsf{KL}^\Sigma(\mu || \nu)}.$$

*Particularly, if $\nu$ satisfies a Logarithmic Sobolev Inequality (LSI) with constant $\lambda_{LSI}$, then $\nu$ satisfies the Talagrand inequality with constant $\lambda_{LSI}$.*

**Theorem 1** (Convergence result for equality-constrained OLLA). *Suppose assumptions (C1) to (C4) hold. Let $X_t$ be the stochastic process following the equality-constrained OLLA (14) and let $\rho_t, \tilde{\rho}_t$ be the law of $X_t$ and its projection $Y_t = \pi(X_t)$ on $\Sigma$ for $t \geq t_{cut}$, $t_{cut} := \max\left\{ \frac{1}{\alpha} \ln \delta, \frac{1}{\alpha} \ln C_5 \right\}$, respectively. Then, for all $t \geq t_{cut}$, it holds that*

$$W_2(\rho_t, \rho_\Sigma) \leq \frac{M_h}{\kappa} e^{-\alpha t} + \sqrt{\frac{2}{\lambda_{LSI}} \mathsf{KL}^\Sigma(\tilde{\rho}_t || \rho_\Sigma)}$$

*where*

$$\mathsf{KL}^\Sigma(\tilde{\rho}_t || \rho_\Sigma) \leq \exp\left( -2\lambda_{LSI}(t - t_{cut}) - \frac{2\lambda_{LSI} C_5}{\alpha}(e^{-\alpha t} - e^{-\alpha t_{cut}}) \right) [\mathsf{KL}^\Sigma(\tilde{\rho}_{t_{cut}} || \rho_\Sigma)$$

$$+ C_6 \int_{t_{cut}}^t \exp\left( 2\lambda(s - t_{cut}) + \frac{2\lambda_{LSI} C_5}{\alpha}(e^{-\alpha s} - e^{-\alpha t_{cut}}) \right) e^{-\alpha s} ds]$$

*In particular, if $\alpha > 2\lambda_{LSI}$, it holds that*

$$\mathsf{KL}^\Sigma(\tilde{\rho}_t || \rho_\Sigma) \leq \exp\left( -2\lambda_{LSI}(t - t_{cut}) - \frac{2\lambda_{LSI} C_5}{\alpha}(e^{-\alpha t} - e^{-\alpha t_{cut}}) \right) [\mathsf{KL}^\Sigma(\tilde{\rho}_{t_{cut}} || \rho_\Sigma) + C_7]$$

*for some constants $C_5 = \mathcal{O}\left( 1 + \frac{C_{L_A} M_h}{\kappa} + \left( \frac{C_{L_A} M_h}{\kappa} \right)^2 \right), C_6, C_7 := \frac{C_6 e^{-\alpha t_{cut}}}{\alpha - 2\lambda_{LSI}} > 0$ with $C_{L_A}$ being the Lipschitz constant of $\nabla \pi(x) \Pi(x)$ on $\hat{U}_\delta(\Sigma)$.*

*Proof.* First, observe that $d_2(p, q) = \|p - q\|_2 \leq d_\Sigma(p, q), \forall p, q \in \Sigma$ because $\Sigma$ is the submanifold of $\mathbb{R}^d$ with Euclidean metric. Thus, $W_2(\tilde{\rho}_t, \rho_\Sigma) \leq W_2^\Sigma(\tilde{\rho}_t, \rho_\Sigma)$ holds and we have

$$W_2(\rho_t, \rho_\Sigma) \overset{\triangle - \text{ineq}}{\leq} W_2(\rho_t, \tilde{\rho}_t) + W_2(\tilde{\rho}_t, \rho_\Sigma) \leq W_2(\rho_t, \tilde{\rho}_t) + W_2^\Sigma(\tilde{\rho}_t, \rho_\Sigma).$$

Now, recall that $\Sigma := \left\{ x \in \mathbb{R}^d \mid h(x) = 0 \right\}$ is a smooth compact and connected Riemannian manifold. Therefore, it is complete by the Hopf-Rinow theorem. Thus, Lemma E.6 implies

$$W_2^\Sigma(\tilde{\rho}_t, \rho_\Sigma) \leq \sqrt{\frac{2}{\lambda_{\text{LSI}}} \mathsf{KL}^\Sigma(\tilde{\rho}_t || \rho_\Sigma)} \Rightarrow W_2(\rho_t, \rho_\Sigma) \leq W_2(\rho_t, \tilde{\rho}_t) + \sqrt{\frac{2}{\lambda_{\text{LSI}}} \mathsf{KL}^\Sigma(\tilde{\rho}_t || \rho_\Sigma)}.$$

Hence, we conclude the proof by borrowing the results of Lemma E.1 and Lemma E.3 $\qquad \square$

# F   Proof of Theoretical Results - Inequality-constrained OLLA

In this section, we analyze the non-asymptotic convergence rate of inequality-constrained OLLA. Note that Proposition 1 gives

$$dX_t = -\nabla f(X_t)dt + \sqrt{2}dW_t, \qquad\qquad\qquad \text{if } g(X_t) < 0$$
$$dX_t = -\Pi(X_t)\nabla f(X_t)dt - \alpha\nabla g_{I_x}^T G(X_t)^{-1}(g_{I_x} + \epsilon\mathbb{1}_{I_x})dt + \sqrt{2}\Pi(X_t)\circ dW_t, \quad \text{otherwise}$$

as the closed form SDE of inequality-constrained OLLA.

## F.1   Convergence Result for Inequality-constrained OLLA

**Lemma F.1** (Boundary behavior of $\rho_t$ of inequality-constrained OLLA). *Let $X_t$ be the stochastic process following the inequality-constrained OLLA and $\rho_t$ be the law of $X_t$. Also, denote $J_t$ be the probability current density defined by $\partial_t \rho_t = -\nabla \cdot J_t$. Then, for $t \geq t_{cut}$, $t_{cut} := \frac{1}{\alpha}\ln\left(\frac{M_g+\epsilon}{\epsilon}\right)$, it holds that*

$$\langle n(x), J_t(x)\rangle = 0, \quad \rho_t(x) = 0, \quad \forall x \in \partial\Sigma,$$

*where $n$ is the unit normal vector of $\Sigma$ and $\sigma_{\partial\Sigma}$ is the surface measure of $\partial\Sigma$.*

*Proof.* From the Fokker-Planck equation of the inequality-constrained OLLA, we know that

$$J_t = q\rho_t - \frac{1}{2}\nabla \cdot [QQ^T\rho_t] = \left[q - \frac{1}{2}\nabla \cdot (QQ^T)\right]\rho_t - \frac{1}{2}QQ^T\nabla\rho_t,$$

where the last equality comes from the chain rule of the matrix divergence. Then, for each $\nabla g_j$, $j \in I_x$, observe that

$$\nabla g_j^T(\nabla \cdot (QQ^T)) = \sum_{k=1}^d (\partial_k g_j)\left[\nabla \cdot (QQ^T)\right]_k = \sum_{i,k=1}^d (\partial_k g_j)\partial_i(QQ^T)_{ik}$$

$$\overset{(\circ)}{=} \sum_{i=1}^d \partial_i\left[\underbrace{\sum_{k=1}^d (QQ^T)_{ik}\partial_k g_j}_{=0}\right] - \sum_{i,k=1}^d (QQ^T)_{ik}\partial_i\partial_k g_j = -\mathsf{Tr}\left(\nabla^2 g_j QQ^T\right)$$

holds for $x \in \partial\Sigma$, where $(\circ)$ holds due to the $\nabla g_k^T Q = 0$ condition of Proposition 1 and $Q^2 = Q$. Therefore, for each $j \in I_x$, we have

$$\langle J_t, \nabla g_j\rangle = \nabla g_j^T J_t = \left[\nabla g_j^T q + \frac{1}{2}\mathsf{Tr}\left(\nabla^2 g_j QQ^T\right)\right]\rho_t \overset{(\triangle)}{=} -\alpha(g+\epsilon)\rho_t = -\alpha\epsilon\rho_t \leq 0,$$

where $(\triangle)$ holds from the $\nabla g_j q + \frac{1}{2}\mathsf{Tr}\left(\nabla^2 g_j QQ^T\right) + \alpha(g+\epsilon) = 0$ condition of Proposition 1. Finally, we conclude the proof by observing the following:

$$0 \overset{(+)}{=} \frac{d}{dt}\int_\Sigma \rho_t d\sigma_\Sigma = -\int_\Sigma \nabla \cdot J_t d\sigma_\Sigma = -\int_{\partial\Sigma} n^T J_t d\sigma_{\partial\Sigma} = \int_{\partial\Sigma} \underbrace{\alpha\epsilon\rho_t}_{\geq 0} d\sigma_{\partial\Sigma} \Rightarrow \rho_t = 0,$$

where $n$ is the outward unit normal vector of $\partial\Sigma$ and $(+)$ holds because $\text{supp}(\rho_t) \subset \Sigma$ for $t \geq t_{cut}$ implies $\int_\Sigma \rho_t d\sigma = 1$ for $t \geq t_{cut}$. $\qquad\square$

**Theorem 2** (Convergence result for inequality-constrained OLLA). *Assume that $\rho_\Sigma$ satisfies the LSI condition with constant $\lambda_{LSI}$. Let $X_t$ be the stochastic process following the inequality-constrained OLLA and let $\rho_t$ be the law of $X_t$. Then, for $t \geq t_{cut}$, $t_{cut} := \frac{1}{\alpha} \ln \left( \frac{M_g + \epsilon}{\epsilon} \right)$, the following holds*

$$W_2(\rho_t, \rho_\Sigma) \leq \sqrt{\frac{2}{\lambda_{LSI}} KL^\Sigma(\rho_t || \rho_\Sigma)},$$

*where*

$$KL^\Sigma(\rho_t || \rho_\Sigma) \leq e^{-2\lambda_{LSI}(t - t_{cut})} KL^\Sigma(\rho_{t_{cut}} || \rho_\Sigma).$$

*Proof.* Observe that

$$\partial_t KL^\Sigma(\rho_t || \rho_\Sigma) = \int_\Sigma \partial_t \rho_t \ln \left( \frac{\rho_t}{\rho_\Sigma} \right) d\sigma_\Sigma + \partial_t \int \rho_t d\sigma_\Sigma = \int_\Sigma \partial_t \rho_t \ln \left( \frac{\rho_t}{\rho_\Sigma} \right) d\sigma_\Sigma$$

$$= \int_\Sigma (-\nabla \cdot J_t) \ln \left( \frac{\rho_t}{\rho_\Sigma} \right) d\sigma_\Sigma = \int_\Sigma J_t \nabla \ln \left( \frac{\rho_t}{\rho_\Sigma} \right) d\sigma_\Sigma - \int_{\partial \Sigma} \underbrace{\langle J_t, n \rangle}_{=0} \ln \left( \frac{\rho_t}{\rho_\Sigma} \right) d\sigma_{\partial \Sigma}$$

$$\stackrel{(\circ)}{=} \int_\Sigma J_t \nabla \ln \left( \frac{\rho_t}{\rho_\Sigma} \right) d\sigma_\Sigma \stackrel{(\triangle)}{=} -I^\Sigma(\rho_t || \rho_\Sigma) \leq -2\lambda_{LSI} KL_\Sigma(\rho_t || \rho_\Sigma),$$

where $(\circ)$ holds due to Lemma F.1 for $t \geq t_{cut}$ and $(\triangle)$ comes from the fact that $J_t = \nabla \ln \rho_\Sigma(x) \rho_t - \rho_t \nabla \ln \rho_t = -\rho_t \nabla \ln \left( \frac{\rho_t}{\rho_\Sigma} \right)$ almost everywhere in $\Sigma$. Therefore, we recover the upper bound of $KL^\Sigma(\rho_t || \rho_\Sigma)$ by applying the Gronwall-type inequality as follows:

$$KL^\Sigma(\rho_t || \rho_\Sigma) \leq e^{-2\lambda_{LSI}(t - t_{cut})} KL^\Sigma(\rho_{t_{cut}} || \rho_\Sigma).$$

Also, we recall that $\Sigma := \left\{ x \in \mathbb{R}^d \mid g(x) \leq 0 \right\}$ is a smooth compact and connected Riemannian manifold, thereby it is complete by the Hopf-Rinow theorem. Thus, Lemma E.6 implies

$$W_2(\rho_t, \rho_\Sigma) \leq W_2^\Sigma(\rho_t, \rho_\Sigma) \leq \sqrt{\frac{2}{\lambda_{LSI}} KL^\Sigma(\rho_t || \rho_\Sigma)}.$$

Hence, we conclude the proof by applying the previous upper bound of $KL^\Sigma(\rho_t || \rho_\Sigma)$. $\qquad\square$

# G   Proof of Theoretical Results - Mixed-constrained OLLA

## G.1   Upper Bound of $W_2(\rho_t, \tilde{\rho}_t)$

**Lemma G.1** (Upper bound of $W_2(\rho_t, \tilde{\rho}_t)$). *Let $\rho_t$ be the law of $X_t$ which follows mixed-constrained OLLA (12) and define $t_1 := \max \left\{ \frac{1}{\alpha} \ln \left( \frac{M_g + \epsilon}{\epsilon} \right), \frac{1}{\alpha} \ln \left( \frac{M_h}{\delta} \right) \right\}$. For $t \geq t_1$, the law $\tilde{\rho}_t$ of $Y_t := \pi(X_t)$ is well-defined and it holds that*

$$W_2(\rho_t, \tilde{\rho}_t) \leq \frac{M_h}{\kappa} e^{-\alpha t} \tag{17}$$

*Proof.* For $t \geq t_1$, observe that

$$\|X_t - \pi(X_t)\|_2 \leq \frac{1}{\kappa} \left( \|h(X_t)\|_2 + \underbrace{\|g_{I_{\pi(X_t)}}(X_t)\|_2}_{=0} \right) \leq \frac{M_h}{\kappa} e^{-\alpha t}$$

by Lemma C.3 and Lemma 1. Then, by integrating both side with respect to optimal coupling of $\rho_t$ and $\tilde{\rho}_t$, we get

$$W_2(\rho_t, \tilde{\rho}_t) \leq \left( \mathbb{E} \left[ \|| X_t - Y_t ||\|_2^2 \right] \right)^{\frac{1}{2}} \stackrel{(\circ)}{\leq} \mathbb{E} \left[ \|X_t - Y_t\|_2 \right] \leq \frac{M_h}{\kappa} e^{-\alpha t}$$

where $(\circ)$ holds by Jensen's inequality. $\qquad\square$

## G.2 Upper Bound of $\mathsf{KL}^\Sigma(\tilde{\rho}_t || \rho_\Sigma)$

**Corollary G.1** (SDE representation of projected process from mixed-constrained OLLA). *Let $X_t$ be the stochastic process following mixed-constrained OLLA* (12). *Define $t_1 :=$* $\max\left\{ \frac{1}{\alpha} \ln\left(\frac{M_g + \epsilon}{\epsilon}\right), \frac{1}{\alpha} \ln\left(\frac{M_h}{\delta}\right) \right\}$ *and assume $Y_t = \pi(X_t) \in int(\Sigma)$ for $t \geq t_1$ via the assumption* *(M1). Then, the projected process $Y_t$ follows the following SDE:*

$$dY_t = [-\Pi(Y_t)\nabla f(Y_t) + b_N(Y_t, t)] dt + \sqrt{2}\Pi(Y_t)(I + A_N(Y_t, t)) \circ dW_t,$$

*where $\|b_N(Y_t, t)\|_2 = \tilde{C}_{b_N} e^{-\alpha t}, \|A_N(Y_t, t)\| = \tilde{C}_{A_N} e^{-\alpha t}$ for $t \geq t_1$ almost surely for some constant $\tilde{C}_{b_N}, \tilde{C}_{A_N} := \frac{\tilde{C}_{L_A} M_h}{\kappa} > 0$ with $\tilde{C}_{L_A}$ being the Lipschitz constant of $\nabla\pi(x)\Pi(x)$ on $\hat{U}_\delta(\Sigma)$.*

*Proof.* By applying Lemma E.2 to the SDE (3), it holds that

$$dY_t = \Pi(Y_t)b(Y_t, t)dt + \sqrt{2}\Pi(Y_t) \circ dW_t + [\nabla\pi(X_t)b(X_t, t) - \nabla\pi(Y_t)b(Y_t, t)] dt$$
$$+ \sqrt{2}[\nabla\pi(X_t)\Pi(X_t) - \nabla\pi(Y_t)\Pi(Y_t)] \circ dW_t$$

for $b(x, t) = b(x) := -[\Pi(x)\nabla f(x) + \alpha\nabla J(x)^T G^{-1}(x)J(x)]$. Now note that $\pi(X_t) \in int(\Sigma)$ for $t \geq t_1$ and the recovery map $\zeta$ (Theorem C.2) is $C^1$ for $\pi(x) \in int(\Sigma), x \in \hat{U}_\delta(\Sigma)$. Therefore, we can set $X_t = \zeta(Y_t, h(X_0)e^{-\alpha t})$ by using Lemma 1 and Theorem C.2. Again, since $X_t, Y_t \in \hat{U}_\delta(\Sigma)$ and the closure of $\hat{U}_\delta(\Sigma)$ is compact, $\nabla\pi(x)b(x)$ and $\nabla\pi(x)\Pi(x)$ is $\tilde{C}_{L_b}, \tilde{C}_{L_A}$-Lipschitz on $\hat{U}_\delta(\Sigma)$, respectively for some $\tilde{C}_{L_b}, \tilde{C}_{L_A} > 0$. Therefore, it holds that

$$\|b_N(Y_t, t)\|_2 \leq \tilde{C}_{L_b}\|\zeta(Y_t, h(X_0)e^{-\alpha t}) - Y_t\|_2 \leq \frac{\tilde{C}_{L_b}\|h(X_0)\|_2}{\kappa}e^{-\alpha t} \leq \frac{\tilde{C}_{L_b}M_h}{\kappa}e^{-\alpha t},$$

where $b_N(Y_t, t) := \nabla\pi(\zeta(Y_t, h(X_0)e^{-\alpha t}))b(\zeta(Y_t, h(X_0)e^{-\alpha t})) - \nabla\pi(Y_t)b(Y_t)$ and the last inequality comes from Lemma C.1. Similarly, we obtain the bound of $A_N(Y_t, t) :=$ $\nabla\pi(\zeta(Y_t, h(X_0)e^{-\alpha t}))\Pi(\zeta(Y_t, h(X_0)e^{-\alpha t})) - \nabla\pi(Y_t)\Pi(Y_t)$ as follows:

$$\|A_N(Y_t, t)\|_2 \leq \tilde{C}_{L_A}\|\zeta(Y_t, h(X_0)e^{-\alpha t}) - Y_t\|_2 \leq \frac{\tilde{C}_{L_A}\|h(X_0)\|_2}{\kappa}e^{-\alpha t} \leq \frac{\tilde{C}_{L_A}M_h}{\kappa}e^{-\alpha t}.$$

Hence, we complete the proof by setting $C_{b_N} := \frac{\tilde{C}_{L_b}M_h}{\kappa}, C_{A_N} := \frac{\tilde{C}_{L_A}M_h}{\kappa}$ and noting that $\nabla\pi(x) = \Pi(\pi(x))\nabla\pi(x)$ for $\pi(x) \in int(\Sigma), x \in \hat{U}_\delta(\Sigma)$, which implies $A_N(Y_t, t) = \Pi(Y_t)A_N(Y_t, t)$. $\square$

**Lemma G.2** (Upper bound of $\mathsf{KL}^\Sigma(\tilde{\rho}_t||\rho_\Sigma)$). *Assume that $\rho_\Sigma$ satisfies the LSI condition with constant $\lambda_{LSI}$. Let $X_t$ be the stochastic process following mixed-constrained OLLA (12) and $\tilde{\rho}_t$ be the law of $Y_t := \pi(X_t)$ after $t \geq t_{cut}, t_{cut} := \max\left\{\frac{1}{\alpha}\ln\left(\frac{M_g+\epsilon}{\epsilon}\right), \frac{1}{\alpha}\ln\left(\frac{M_h}{\delta}\right), \frac{1}{\alpha}\ln(\tilde{C}_5)\right\}$. Suppose*

- *(Regularity of $\Sigma_p$)*    $\Sigma_p := \pi(\{x \in \mathbb{R}^d \mid h(x) = p, g(x) \leq 0\}) \subset int(\Sigma)$ for $0 < \|p\|_2 \leq \delta$.

- *(Regularity of $\partial\Sigma_p$)*    *The boundary velocity $v_p^b$ of $\partial\Sigma_p$ satisfies $\sup_{x\in\partial\Sigma_p}\|v_p^b\|_2 \leq V\|p\|_2^\beta$ for some $V > 0, \beta > 0$. Also, assume $M_\Sigma := \sup_{\|p\|_2<\delta} \sigma_{\partial\Sigma_p}(\partial\Sigma_p) < \infty$.*

- *(Bound on $\rho_t, \rho_\Sigma$)*    $G_1 := \sup_{t\geq 0, x\in\Sigma} \tilde{\rho}_t < \infty$ and $0 < G_2 \leq \rho_\Sigma \leq G_3$ for $x \in \Sigma$.

*Then, for $\alpha \neq 2\lambda_{LSI}$, the following non-asymptotic convergence rate of $\mathsf{KL}^\Sigma(\tilde{\rho}_t||\rho_\Sigma)$ can be obtained as follows*

$$\mathsf{KL}^\Sigma(\tilde{\rho}_t||\rho_\Sigma) \leq \exp\left(-2\lambda_{LSI}(t - t_{cut}) - \frac{2\lambda_{LSI}\tilde{C}_5}{\alpha}(e^{-\alpha t} - e^{-\alpha t_{cut}})\right) \times$$

$$[\mathsf{KL}^\Sigma(\tilde{\rho}_{t_{cut}}||\rho_\Sigma) + \int_{t_{cut}}^t \exp\left(2\lambda_{LSI}(s - t_{cut}) + \frac{2\lambda_{LSI}\tilde{C}_5}{\alpha}(e^{-\alpha s} - e^{-\alpha t_{cut}})\right) \times$$

$$\left[\left(\tilde{C}_6 + \alpha G_4 G_5 M_h\right)e^{-\alpha s} + G_4 V M_h^\beta e^{-\alpha\beta s}\right] ds].$$

*In particular, if $\alpha > 2\lambda_{LSI}$, $\beta \geq 1$, the inequality becomes*

$$\mathsf{KL}^\Sigma(\tilde{\rho}_t||\rho_\Sigma) \leq \exp\left(-2\lambda_{LSI}(t - t_{cut}) - \frac{2\lambda_{LSI}\tilde{C}_5}{\alpha}(e^{-\alpha t} - e^{-\alpha t_{cut}})\right)[\mathsf{KL}^\Sigma(\tilde{\rho}_{t_{cut}}||\rho_\Sigma) + \tilde{C}_7 + \tilde{C}_8]$$

*for some constants $\tilde{C}_5 = \mathcal{O}(1 + \tilde{C}_{A_N} + \tilde{C}_{A_N}^2), G_4, G_5, G_6, \tilde{C}_6, \tilde{C}_7 > 0$, and $\tilde{C}_7 := (\tilde{C}_6 + \alpha G_4 G_5 M_h)\frac{e^{-\alpha t_{cut}}}{\alpha - 2\lambda_{LSI}}$ and $\tilde{C}_8 := (G_6 V M_h^\beta)\frac{e^{-\alpha\beta t_{cut}}}{\alpha\beta - 2\lambda_{LSI}}$.*

*Proof.* First, Corollary G.1, and the choice of $\nabla f = -\nabla \ln \rho_\Sigma$ gives the following SDE of the projected process $Y_t$ of $X_t$ for $t \geq t_{\text{cut}}$:

$$dY_t = [-\Pi(Y_t)\nabla f(Y_t) + b_N(Y_t, t)] dt + \sqrt{2}\Pi(Y_t)(I + A_N(Y_t, t)) \circ dW_t$$

where $b_N(x, t) := \nabla\pi(\zeta(x, h(X_0)e^{-\alpha t}))b(\zeta(x, h(X_0)e^{-\alpha t})) - \nabla\pi(x)b(x)$, $A_N(x, t) := \nabla\pi(\zeta(x, h(X_0)e^{-\alpha t}))\Pi(\zeta(x, h(X_0)e^{-\alpha t})) - \nabla\pi(x)\Pi(x)$ for $x \in \Sigma$, conditionally on $X_0$.

Hence, from Corollary G.1, its associated Fokker-Planck equation can be written as follows:

$$\partial_t\tilde{\rho}_t = -\mathsf{div}_\Sigma(\tilde{\rho}_t(\nabla_\Sigma \ln \rho_\Sigma + b_N)) + \sum_{k=1}^d \mathsf{div}_\Sigma\left(\mathsf{div}_\Sigma(\tilde{\rho}_t(f_k + \delta_k))(f_k + \delta_k)\right).$$

Defining $\Sigma_t := \pi(\{x \in \mathbb{R}^d \mid h(x) = h(X_0)e^{-\alpha t}, g(x) \leq 0\})$ conditionally on $X_0$, we observe

$$\partial_t\mathsf{KL}^\Sigma(\tilde{\rho}_t||\rho_\Sigma) = \partial_t \int_{\Sigma_t} \tilde{\rho}_t \ln\left(\frac{\tilde{\rho}_t}{\rho_\Sigma}\right) d\sigma_\Sigma$$

$$= \underbrace{\int_{\Sigma_t} \partial_t\tilde{\rho}_t \ln\left(\frac{\tilde{\rho}_t}{\rho_\Sigma}\right) d\sigma_\Sigma}_{\text{Term (1)}} + \underbrace{\int_{\partial\Sigma_t} \tilde{\rho}_t\left[\ln\left(\frac{\tilde{\rho}_t}{\rho_\Sigma}\right) - 1\right]\langle v_t^b, n_t\rangle d\sigma_{\partial\Sigma_t}}_{\text{Term (2)}} + \underbrace{\partial_t \int_{\Sigma_t} \tilde{\rho}_t d\sigma_\Sigma}_{=0}$$

where the last equality holds from the Leibniz integral rule with $v_t^b$ being the velocity vector of the boundary of $\Sigma_t$ and $n_t$ being the outward unit normal vector of $\partial\Sigma_t$. Therefore, the expression of

$\partial_t \tilde{\rho}_t$ implies that Term (1) becomes

$$\text{Term (1)} = \underbrace{\int_{\Sigma_t} \left\langle \left[ \tilde{\rho}_t(\nabla_\Sigma \ln \rho_\Sigma + b_N) - \sum_{k=1}^d \mathsf{div}_\Sigma(\tilde{\rho}_t(f_k + \delta_k))(f_k + \delta_k) \right], \nabla_\Sigma \ln\left(\frac{\tilde{\rho}_t}{\rho_\Sigma}\right) \right\rangle d\sigma_\Sigma}_{\text{Term (1-1)}}$$

$$\underbrace{- \int_{\partial\Sigma_t} \left\langle \left[ \tilde{\rho}_t(\nabla_\Sigma \ln \rho_\Sigma + b_N) - \sum_{k=1}^d \mathsf{div}_\Sigma(\tilde{\rho}_t(f_k + \delta_k))(f_k + \delta_k) \right] \ln\left(\frac{\tilde{\rho}_t}{\rho_\Sigma}\right), n_t \right\rangle d\sigma_{\partial\Sigma_t}}_{\text{Term (1-2)}}$$

by integration by parts, where $f_k := \Pi e_k, \delta_k := \Pi A_N \delta_k$. Now, we observe that Term (1-1) can be bounded as

$$\text{Term (1-1)} \le -(1 - \tilde{C}_5 e^{-\alpha t}) I^\Sigma(\tilde{\rho}_t || \rho_\Sigma) + \tilde{C}_6 e^{-\alpha t}$$

following the same proof of Lemma E.3 with different constants $\tilde{C}_5 = \mathcal{O}(1 + \tilde{C}_{A_N} + \tilde{C}_{A_N}^2), \tilde{C}_6 > 0$ (note that we ignored the integrand at $\partial\Sigma$ which is measure zero with respect to $d\sigma_\Sigma$).

For the analysis of Term (1-2), we first observe the following fact:

$$\int_{\partial\Sigma_t} \left| \tilde{\rho}_t \ln\left(\frac{\tilde{\rho}_t}{\rho_\Sigma}\right) \right| d\sigma_{\partial\Sigma_t} \le G_3 \max\left\{ \frac{1}{e}, \left| \frac{G_1}{G_2} \ln\left(\frac{G_1}{G_2}\right) \right| \right\} \sigma_{\partial\Sigma_t}(\partial\Sigma_t)$$

$$\le G_3 M_\Sigma \max\left\{ \frac{1}{e}, \left| \frac{G_1}{G_2} \ln\left(\frac{G_1}{G_2}\right) \right| \right\} := G_4$$

from the assumptions of $G_1 := \sup_{t \ge 0, x \in \Sigma} \tilde{\rho}_t < \infty, 0 < G_2 \le \rho_\Sigma \le G_3$, and the regularity of $\partial\Sigma_t$ such that $\sup_{t \ge t_{\text{cut}}} \sigma_{\partial\Sigma_t}(\partial\Sigma_t) \le M_\Sigma < \infty$.

Next, from Corollary G.1, we note that the following holds conditionally on $X_0$:

$$f_k(x) + \delta_k(x) = \Pi(x)(I + A_N(t, x))e_k = \nabla\pi(\zeta(x, h(X_0)e^{-\alpha t}))\Pi(\zeta(x, h(X_0)e^{-\alpha t}))e_k.$$

Because $\Pi(\zeta(x, h(X_0)e^{-\alpha t}))e_k$ is a tangent vector on $\{x \in \mathbb{R}^d \mid h(x) = h(X_0)e^{-\alpha t}, g(x) \le 0\}$, $f_k(x) + \delta_k(x) = \nabla\pi(\zeta(x, h(X_0)e^{-\alpha t}))\Pi(\zeta(x, h(X_0)e^{-\alpha t}))e_k$ becomes a tangent vector of $\Sigma_t$. Similarly, it also becomes a tangent vector of $\partial\Sigma_t$ on the boundary because $\Pi$ is the orthogonal projector induced by $h$ and active $g$. Hence, $\langle f_k + \delta_k, n_t \rangle = 0$ holds, where $n_t$ is the outward unit normal vector of $\partial\Sigma_t$.

Therefore, Term (1-2) becomes

$$|\text{Term (1-2)}| \overset{(1)}{\le} \alpha G_5 M_h e^{-\alpha t} \int_{\partial\Sigma_t} \left| \tilde{\rho}_t \ln\left(\frac{\tilde{\rho}_t}{\rho_\Sigma}\right) \right| d\sigma_{\partial\Sigma_t} \le \alpha G_4 G_5 M_h e^{-\alpha t},$$

where (1) holds because Corollary G.1 gives

$$\nabla_\Sigma \ln \rho_\Sigma(x) + b_N(x, t) = \nabla\pi(\zeta(x, h(X_0)e^{-\alpha t}))b(\zeta(t, h(X_0)e^{-\alpha t}))$$

with $b(x) := -\left[ \Pi(x)\nabla f(x) + \alpha\nabla J(x)^T G^{-1}(x)J(x) \right]$ and, therefore,

$$|\langle \nabla_\Sigma \ln \rho_\Sigma(x) + b_N(x, t), n_t(x) \rangle| \le \alpha G_5 \|J(\zeta(x, h(X_0)e^{-\alpha t}))\|_2 \le \alpha G_5 M_h e^{-\alpha t}$$

for $x \in \partial\Sigma_t$ with $G_5 := \sup_{x \in \Sigma, \|p\|_2 < \delta} \|\nabla\pi(\zeta(x, p))\nabla J(\zeta(x, p))^T G^{-1}(\zeta(x, p))\|_2 < \infty$.

Also, similarly, Term (2) is bounded as follows:

$$|\text{Term (2)}| \le \sup_{x \in \partial\Sigma_t} \|v_t^b\|_2 \int_{\partial\Sigma_t} \left( \left| \tilde{\rho}_t \ln\left(\frac{\tilde{\rho}_t}{\rho_\Sigma}\right) \right| + |\tilde{\rho}_t| \right) d\sigma_{\partial\Sigma_t} \le G_6 V M_h^\beta e^{-\alpha\beta t}$$

with $G_6 := G_4 + G_1 M_\Sigma$. Therefore, combining the results with the LSI condition gives the following inequality:

$$\partial_t \mathsf{KL}^\Sigma(\tilde{\rho}_t || \rho_\Sigma) \le -2\lambda_{\text{LSI}}(1 - \tilde{C}_5 e^{-\alpha t})\mathsf{KL}^\Sigma(\tilde{\rho}_t || \rho_\Sigma) + \left( \tilde{C}_6 + \alpha G_4 G_5 M_h \right) e^{-\alpha t} + G_6 V M_h^\beta e^{-\alpha\beta t}$$

where the last inequality comes from the LSI condition. Finally, applying the Grönwall-type inequality recovers the following inequality:

$$\mathsf{KL}^{\Sigma}(\tilde{\rho}_t||\rho_{\Sigma}) \leq \exp\left(-2\lambda_{\mathrm{LSI}}(t - t_{\mathrm{cut}}) - \frac{2\lambda_{\mathrm{LSI}}\tilde{C}_5}{\alpha}(e^{-\alpha t} - e^{-\alpha t_{\mathrm{cut}}})\right)[\mathsf{KL}^{\Sigma}(\tilde{\rho}_{t_{\mathrm{cut}}}||\rho_{\Sigma})$$

$$+ \int_{t_{\mathrm{cut}}}^{t} \exp\left(2\lambda_{\mathrm{LSI}}(s - t_{\mathrm{cut}}) + \frac{2\lambda_{\mathrm{LSI}}\tilde{C}_5}{\alpha}(e^{-\alpha s} - e^{-\alpha t_{\mathrm{cut}}})\right) \times$$

$$\left[\left(\tilde{C}_6 + \alpha G_4 G_5 M_h\right)e^{-\alpha s} + G_6 V M_h^{\beta} e^{-\alpha\beta s}\right]ds].$$

Also, similarly in the last argument of Lemma E.3, we observe that if $\alpha > 2\lambda_{\mathrm{LSI}}$ and $\beta \geq 1$

$$\int_{t_{\mathrm{cut}}}^{\infty} \exp\left(2\lambda_{\mathrm{LSI}}(s - t_{\mathrm{cut}}) + \frac{2\lambda_{\mathrm{LSI}}\tilde{C}_5}{\alpha}(e^{-\alpha s} - e^{-\alpha t_{\mathrm{cut}}})\right)e^{-\alpha s}ds \leq \int_{t_{\mathrm{cut}}}^{\infty} \exp\left(2\lambda_{\mathrm{LSI}}(s - t_{\mathrm{cut}})\right)e^{-\alpha s}ds$$

$$\leq \frac{e^{-\alpha t_{\mathrm{cut}}}}{\alpha - 2\lambda_{\mathrm{LSI}}} < \infty$$

and

$$\int_{t_{\mathrm{cut}}}^{\infty} \exp\left(2\lambda_{\mathrm{LSI}}(s - t_{\mathrm{cut}}) + \frac{2\lambda_{\mathrm{LSI}}\tilde{C}_5}{\alpha}(e^{-\alpha s} - e^{-\alpha t_{\mathrm{cut}}})\right)e^{-\alpha\beta s}ds \leq \frac{e^{-\alpha\beta t_{\mathrm{cut}}}}{\alpha\beta - 2\lambda_{\mathrm{LSI}}} < \infty.$$

Therefore, there exists $\tilde{C}_7, \tilde{C}_8 < \infty$ such that

$$\mathsf{KL}^{\Sigma}(\tilde{\rho}_t||\rho_{\Sigma}) \leq \exp\left(-2\lambda_{\mathrm{LSI}}(t - t_{\mathrm{cut}}) - \frac{2\lambda_{\mathrm{LSI}}\tilde{C}_5}{\alpha}(e^{-\alpha t} - e^{-\alpha t_{\mathrm{cut}}})\right)[\mathsf{KL}^{\Sigma}(\tilde{\rho}_{t_{\mathrm{cut}}}||\rho_{\Sigma}) + \tilde{C}_7 + \tilde{C}_8],$$

where $\tilde{C}_7 := (\tilde{C}_6 + \alpha G_4 G_5 M_h)\frac{e^{-\alpha t_{\mathrm{cut}}}}{\alpha - 2\lambda_{\mathrm{LSI}}}$ and $\tilde{C}_8 := (G_6 V M_h^{\beta})\frac{e^{-\alpha\beta t_{\mathrm{cut}}}}{\alpha\beta - 2\lambda_{\mathrm{LSI}}}$. □

**Theorem 3** (Convergence result for mixed-constrained OLLA). *Assume that $\rho_\Sigma$ satisfies the LSI condition with constant $\lambda_{LSI}$. Let $X_t$ be the stochastic process following mixed-constrained OLLA (3) and $\tilde{\rho}_t$ be the law of $Y_t := \pi(X_t)$ after $t \geq t_{cut}, t_{cut} := \max\left\{ \frac{1}{\alpha}\ln\left(\frac{M_g+\epsilon}{\epsilon}\right), \frac{1}{\alpha}\ln\left(\frac{M_h}{\delta}\right), \frac{1}{\alpha}\ln(\tilde{C}_5)\right\}$. Suppose*

- *(Regularity of $\Sigma_p$)* $\quad \Sigma_p := \pi(\{x \in \mathbb{R}^d \mid h(x) = p, g(x) \leq 0\}) \subset int(\Sigma)$ *for $0 < \|p\|_2 \leq \delta$.*

- *(Regularity of $\partial\Sigma_p$)* $\quad$ *The boundary velocity $v_p^b$ of $\partial\Sigma_p$ satisfies $\sup_{x \in \partial\Sigma_p}\|v_p^b\|_2 \leq V\|p\|_2^\beta$ for some $V > 0, \beta > 0$. Also, assume $M_\Sigma := \sup_{\|p\|_2 < \delta} \sigma_{\partial\Sigma_p}(\partial\Sigma_p) < \infty$.*

- *(Bound on $\rho_t, \rho_\Sigma$)* $\quad G_1 := \sup_{t \geq 0, x \in \Sigma} \tilde{\rho}_t < \infty$ *and $0 < G_2 \leq \rho_\Sigma \leq G_3$ for $x \in \Sigma$.*

*Then, for $\alpha \neq 2\lambda_{LSI}$ the following non-asymptotic convergence rate of $W_2(\rho_t, \rho_\Sigma)$ can be obtained as follows*

$$W_2(\rho_t, \rho_\Sigma) \leq \frac{M_h}{\kappa}e^{-\alpha t} + \sqrt{\frac{2}{\lambda_{LSI}}\mathsf{KL}^\Sigma(\tilde{\rho}_t\|\rho_\Sigma)}$$

*where*

$$\mathsf{KL}^\Sigma(\tilde{\rho}_t\|\rho_\Sigma) \leq \exp\left(-2\lambda_{LSI}(t - t_{cut}) - \frac{2\lambda_{LSI}\tilde{C}_5}{\alpha}(e^{-\alpha t} - e^{-\alpha t_{cut}})\right) \times$$

$$[\mathsf{KL}^\Sigma(\tilde{\rho}_{t_{cut}}\|\rho_\Sigma) + \int_{t_{cut}}^t \exp\left(2\lambda_{LSI}(s - t_{cut}) + \frac{2\lambda_{LSI}\tilde{C}_5}{\alpha}(e^{-\alpha s} - e^{-\alpha t_{cut}})\right) \times$$

$$\left[\left(\tilde{C}_6 + \alpha G_4 G_5 M_h\right)e^{-\alpha s} + G_6 V M_h^\beta e^{-\alpha\beta s}\right]ds]$$

*In particular, if $\alpha > 2\lambda_{LSI}$ and $\beta \geq 1$, the previous bound simplifies to*

$$\mathsf{KL}^\Sigma(\tilde{\rho}_t\|\rho_\Sigma) \leq \exp\left(-2\lambda_{LSI}(t - t_{cut}) - \frac{2\lambda_{LSI}\tilde{C}_5}{\alpha}(e^{-\alpha t} - e^{-\alpha t_{cut}})\right)[\mathsf{KL}^\Sigma(\tilde{\rho}_{t_{cut}}\|\rho_\Sigma) + \tilde{C}_7 + \tilde{C}_8]$$

*for some constants $\tilde{C}_5 = \mathcal{O}\left(1 + \frac{\tilde{C}_{L_A}M_h}{\kappa} + \left(\frac{\tilde{C}_{L_A}M_h}{\kappa}\right)^2\right), G_4, G_5, G_6, \tilde{C}_6, \tilde{C}_7 > 0$, and $\tilde{C}_7 := (\tilde{C}_6 + \alpha G_4 G_5 M_h)\frac{e^{-\alpha t_{cut}}}{\alpha - 2\lambda_{LSI}}$ and $\tilde{C}_8 := (G_6 V M_h^\beta)\frac{e^{-\alpha\beta t_{cut}}}{\alpha\beta - 2\lambda_{LSI}}$, with $\tilde{C}_{L_A}$ being the Lipschitz constant of $\nabla\pi(x)\Pi(x)$ on $\hat{U}_\delta(\Sigma)$.*

*Proof.* We note that $\tilde{\rho}_t(x) = 0$ on $\partial\Sigma$ for $t \geq t_{cut}$ holds from the regularity of the $\Sigma_p$ assumption. Therefore, by the same approach in Theorem 1 and Lemma E.6, it holds that

$$W_2^\Sigma(\tilde{\rho}_t, \rho_\Sigma) \leq \sqrt{\frac{2}{\lambda_{LSI}}\mathsf{KL}^\Sigma(\tilde{\rho}_t\|\rho_\Sigma)} \Rightarrow W_2(\rho_t, \rho_\Sigma) \leq W_2(\rho_t, \tilde{\rho}_t) + \sqrt{\frac{2}{\lambda_{LSI}}\mathsf{KL}^\Sigma(\tilde{\rho}_t\|\rho_\Sigma)}.$$

Therefore, we conclude the proof by combining the results of Lemma G.1 and Lemma G.2. $\qquad\square$

# H  Experiment Settings and Supplementary Results

**Settings.**  The first two experiments were executed on a desktop with an AMD Ryzen 9 7900X CPU (12 cores) with 32 GB RAM. Runs were implemented in WSL2 (Ubuntu) environment (CPU-only), using the Python and the PyTorch [54] framework.

## H.1  Experiment Settings and Supplementary Results for Synthetic 2D Examples

**Experiment Settings.**  In this experiment, we compare four samplers (OLLA, OLLA-H, CLangevin, CHMC) on the following synthetic 2D examples:

1. **(Star)**  a star-shaped equality manifold with uniform density:

$$f(x) = 0, \ h(x) = \sqrt{x_1^2 + x_2^2} - (1.5 + 0.3\cos(5\theta)), \ \theta = \arctan 2(x_2, x_1).$$

2. **(Two Lobes)**  a two-lobe inequality manifold (from [18]) with uniform density:

$$f(x) = 0, \ g(x) = -\ln q(x) - 2, \ q(x) = \frac{e^{-2(x_1-3)^2} + e^{-2(x_1+3)^2}}{e^{2(\|x\|_2-3)^2}}.$$

3. **(Quadratic Poly)**  a quadratic curve defined by mixed polynomial equality and inequality under a standard Gaussian target:

$$f(x) = \frac{1}{2}\|x\|_2^2, \ h(x) = x_1^4 x_2^2 + x_1^2 + x_2 - 1, \ g(x) = x_1^3 - x_2^3 - 1.$$

4. **(Mixture Gaussian)**  a nine-Gaussian mixture restricted by a seven-lobe manifold:

$$f(x) = -\ln\left(\sum_{i=1}^{9} \exp\left(-5\|x - c_i\|_2^2\right)\right), \ h(x) = \sqrt{x_1^2 + x_2^2} - (3 + \cos(7\theta)).$$

$$g(x) = (x_1 - 2)^2 - 5x_1 x_2^3 + 0.5x_2^5 - 40, \ \theta = \arctan 2(x_2, x_1).$$

with $\{c_i\}_{i=1}^{9} = \{-2, 0, 2\}^2$.

where $x = [x_1, x_2]^T \in \mathbb{R}^2$. For each 2D example, we run 200 independent chains for $K = 5000$ steps each. From each chain, we retain only the state at step $K$, yielding 200 samples per sampler.

To provide a fixed target distribution for distance calculation, we generate 200 samples using CGHMC. This reference is held constant across all comparisons. For each sampler, we compute $W_2^2$, energy distance as well as the mean constraint violations $\mathbb{E}[|h(x)|]$ and $\mathbb{E}[\max g(x)^+]$, over the 200 samples.

The hyperparameter setup is provided in Table 6. To mitigate ill-conditioning in the Newton solver of CLangevin, we add Tikhonov regularization with Tikhonov matrix $\Gamma = \sqrt{\lambda}I$, $\lambda = 1.0$ for the Mixture Gaussian example, and $\lambda = 0.1$ for the

Table 6: Hyperparameter settings for 2D synthetic examples ($\Delta t = 5 \times 10^{-4}$)

| Method | Hyperparameters |
|---|---|
| OLLA | $\alpha = 200, \ \epsilon = 1$ |
| OLLA–H | $\alpha = 200, \ \epsilon = 1, \ N = 5$ |
| CLangevin | $L = 3, \ \tau = 10^{-4}, \lambda = \{1, 0.1\}$ |
| CHMC | $\gamma = 1, \ L = 3, \ \tau = 10^{-4}, \lambda = 0$ |
| CGHMC | $\gamma = 1, \ L = 3, \ \tau = 10^{-4}, \lambda = 0$ |

other three. For the CLangevin, CHMC, CGHMC, $X_0$ is initialized exactly on $\Sigma$ to ensure the stability of algorithms while OLLA and OLLA-H have noisy initialization $X_0 = Y_0 + \mathcal{N}(0, I), Y_0 \in \Sigma$ and $X_t$ progressively approaches to $\Sigma$ by the landing mechanism.

**Remark 8.**  The results shown in Figure 2 and Figure 3 were obtained under a different setup described above: a larger step size $\Delta t$ was used; all methods were initialized from the same initialization point $X_0 \in \Sigma$, rather than via random sampling near $\Sigma$; and the regularization parameter $\lambda$ in CLangevin was increased for improved numerical stability under large step size.

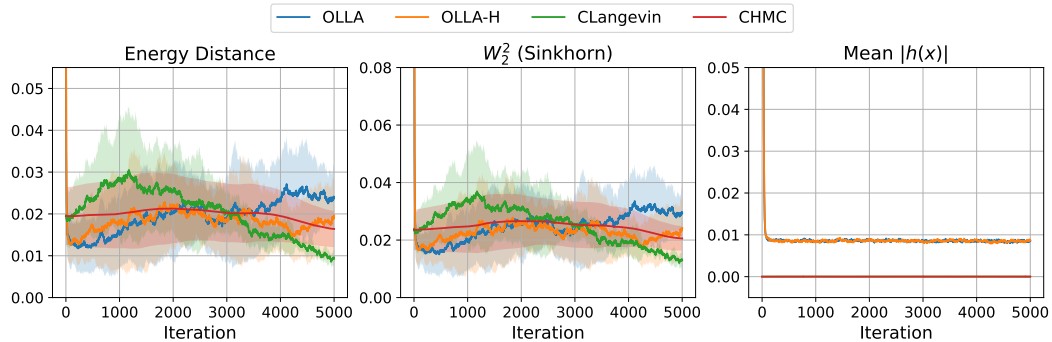

Figure 6: Convergence diagnostics on the Star example (equality-only case). (1) energy distance to CGHMC samples (left), (2) $W_2^2$ distance to CGHMC samples (center), and (3) mean of $|h(x)|$ (right)

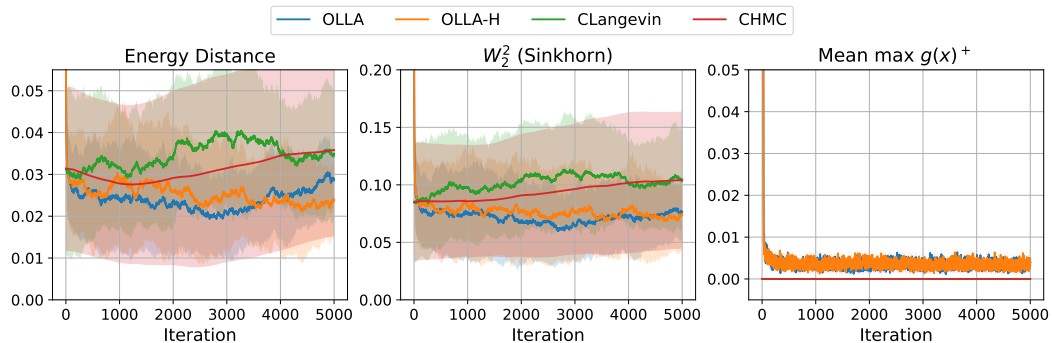

Figure 7: Convergence diagnostics on the Two Lobes example (inequality-only case). (1) energy distance to CGHMC samples (left), (2) $W_2^2$ distance to CGHMC samples (center), and (3) mean of $\max g(x)^+$ (right)

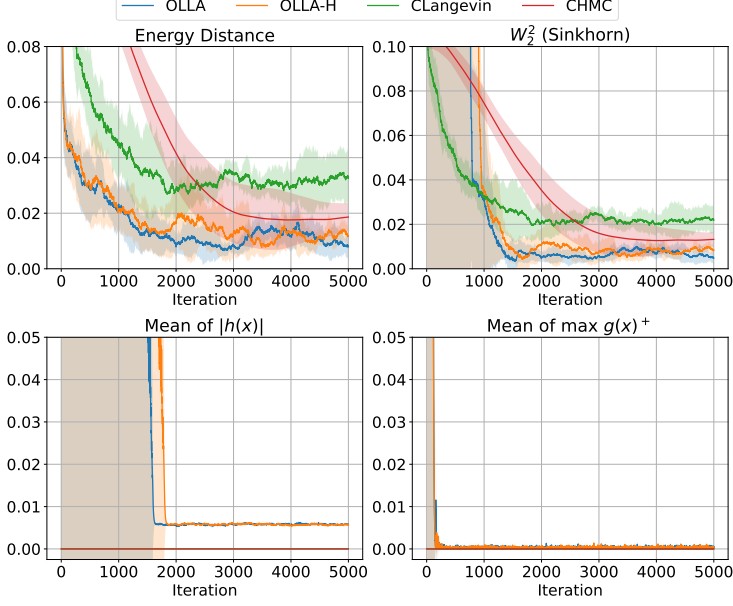

Figure 8: Convergence diagnostics on the Quadratic Poly example (mixed-case). (1) energy distance to CGHMC samples (top left) and (2) $W_2^2$ distance to CGHMC samples (top right). (3) mean of $|h(x)|$ (bottom left) and (4) mean of $\max g(x)^+$ (bottom right)

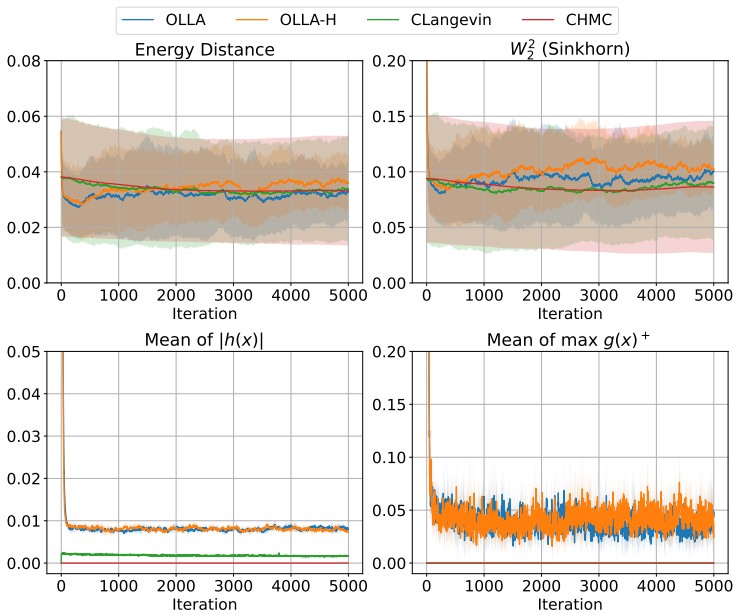

Figure 9: Convergence diagnostics on the Mixture Gaussian example (mixed-case). (1) energy distance to CGHMC samples (top left) and (2) $W_2^2$ distance to CGHMC samples (top right). (3) mean of $|h(x)|$ (bottom left) and (4) mean of $\max g(x)^+$ (bottom right)

**Supplementary Results - Sampling Accuracy & Constraint Violation.** In addition to the Mixture Gaussian example (Figure 3), we evaluated OLLA and OLLA-H on three further 2D geometries to verify that the distributional accuracy observed in Figure 3 of the main text generalize to other settings. Across all three additional examples, OLLA and OLLA-H closely match the convergence behavior of CLangevin and CHMC in both energy distance and $W_2^2$ metrics, while maintaining constraint violations at low levels without requiring explicit projection steps. These trends are illustrated in Figure 6 (Star), Figure 7 (Two Lobes), Figure 8 (Quadratic Poly), and Figure 9 (Mixture Gaussian under the hyperparameter setup of Table 6). These additional experiments confirm that our landing-based sampler can provide relatively accurate, constraint-respecting samples across diverse manifold geometries.

**Supplementary Results - Effect of Hyperparameters $\alpha$ and $\epsilon$.** In Table 7, we report how varying the landing rate $\alpha$ (with $\epsilon = 1$) affects sampling performance on the Mixture Gaussian example. For both OLLA and OLLA–H, increasing $\alpha$ from 1 to 500 leads to consistent reductions in energy distance, $W_2^2$, and the average constraint violation $\mathbb{E}[|h(x)|]$. However, setting $\alpha$ too large causes sampling failures and numerical errors (e.g. $\alpha = 5000$).

Similarly, Table 8 examines the effect of boundary repulsion rate $\epsilon$ (with $\alpha = 100$). Across the full range of $\epsilon$ tested, sampling accuracy—as measured by both $W_2^2$ and energy distance—remains essentially invariant, exhibiting smaller variation than when $\alpha$ is changed. In contrast, inequality-constraint enforcement steadily improves as $\epsilon$ increases: the average violation $\mathbb{E}[g(x)^+]$ declines monotonically, reflecting stronger repulsion. Only when $\epsilon$ becomes exceedingly large does one observe a degradation in equality-constraint enforcement and occasional numerical instabilities, mirroring the breakdown seen at an extreme $\alpha$ value.

Table 7: Effect of $\alpha$ on energy distance, $W_2^2$, $\mathbb{E}[|h(x)|]$, and $\mathbb{E}[g(x)^+]$ on the Mixture Gaussian example with $\epsilon = 1$ (top: OLLA, bottom: OLLA-H)

| $\alpha$ | energy distance | $W_2^2$ | $\mathbb{E}[|h(x)|]$ | $\mathbb{E}[g(x)^+]$ |
|---|---|---|---|---|
| 1 | $0.121_{\pm 0.025}$ | $0.363_{\pm 0.064}$ | $0.682_{\pm 0.017}$ | $1.113_{\pm 0.351}$ |
| 10 | $0.066_{\pm 0.019}$ | $0.200_{\pm 0.035}$ | $0.130_{\pm 0.001}$ | $0.234_{\pm 0.032}$ |
| 100 | $0.053_{\pm 0.016}$ | $0.159_{\pm 0.032}$ | $0.017_{\pm 0.001}$ | $0.045_{\pm 0.007}$ |
| 200 | $0.040_{\pm 0.012}$ | $0.121_{\pm 0.019}$ | $0.008_{\pm 0.001}$ | $0.054_{\pm 0.009}$ |
| 500 | $0.033_{\pm 0.011}$ | $0.104_{\pm 0.020}$ | $0.004_{\pm 0.000}$ | $0.021_{\pm 0.018}$ |
| 700 | $0.044_{\pm 0.012}$ | $0.132_{\pm 0.019}$ | $0.003_{\pm 0.000}$ | $0.016_{\pm 0.011}$ |
| 5000 | NaN (results unavailable) | | | |
| 1 | $0.104_{\pm 0.019}$ | $0.333_{\pm 0.077}$ | $0.643_{\pm 0.032}$ | $1.102_{\pm 0.337}$ |
| 10 | $0.059_{\pm 0.018}$ | $0.181_{\pm 0.038}$ | $0.129_{\pm 0.011}$ | $0.189_{\pm 0.023}$ |
| 100 | $0.052_{\pm 0.017}$ | $0.156_{\pm 0.026}$ | $0.015_{\pm 0.001}$ | $0.055_{\pm 0.024}$ |
| 200 | $0.052_{\pm 0.018}$ | $0.153_{\pm 0.037}$ | $0.009_{\pm 0.000}$ | $0.039_{\pm 0.015}$ |
| 500 | $0.041_{\pm 0.013}$ | $0.124_{\pm 0.027}$ | $0.004_{\pm 0.000}$ | $0.013_{\pm 0.009}$ |
| 700 | $0.037_{\pm 0.011}$ | $0.110_{\pm 0.019}$ | $0.002_{\pm 0.000}$ | $0.007_{\pm 0.006}$ |
| 5000 | NaN (results unavailable) | | | |

Table 8: Effect of $\epsilon$ on energy distance, $W_2^2$, $\mathbb{E}[|h(x)|]$, and $\mathbb{E}[g(x)^+]$ on the Mixture Gaussian example with $\alpha = 100$ (top: OLLA, bottom: OLLA-H)

| $\epsilon$ | energy distance | $W_2^2$ | $\mathbb{E}[|h(x)|]$ | $\mathbb{E}[g(x)^+]$ |
|---|---|---|---|---|
| 0.1 | $0.048_{\pm 0.014}$ | $0.151_{\pm 0.026}$ | $0.014_{\pm 0.001}$ | $0.082_{\pm 0.017}$ |
| 1 | $0.033_{\pm 0.004}$ | $0.108_{\pm 0.011}$ | $0.017_{\pm 0.002}$ | $0.067_{\pm 0.027}$ |
| 5 | $0.040_{\pm 0.006}$ | $0.123_{\pm 0.018}$ | $0.016_{\pm 0.001}$ | $0.040_{\pm 0.015}$ |
| 10 | $0.036_{\pm 0.016}$ | $0.112_{\pm 0.034}$ | $0.017_{\pm 0.001}$ | $0.019_{\pm 0.006}$ |
| 50 | $0.038_{\pm 0.014}$ | $0.112_{\pm 0.029}$ | $0.018_{\pm 0.001}$ | $0.003_{\pm 0.004}$ |
| 200 | $0.041_{\pm 0.020}$ | $0.119_{\pm 0.057}$ | $0.018_{\pm 0.002}$ | $0.006_{\pm 0.012}$ |
| 10000 | $0.066_{\pm 0.022}$ | $0.137_{\pm 0.124}$ | $0.033_{\pm 0.020}$ | $0.000_{\pm 0.000}$ |
| 0.1 | $0.039_{\pm 0.012}$ | $0.126_{\pm 0.014}$ | $0.013_{\pm 0.001}$ | $0.073_{\pm 0.026}$ |
| 1 | $0.048_{\pm 0.018}$ | $0.142_{\pm 0.029}$ | $0.016_{\pm 0.001}$ | $0.048_{\pm 0.014}$ |
| 5 | $0.035_{\pm 0.011}$ | $0.111_{\pm 0.031}$ | $0.018_{\pm 0.001}$ | $0.027_{\pm 0.007}$ |
| 10 | $0.044_{\pm 0.014}$ | $0.127_{\pm 0.026}$ | $0.018_{\pm 0.001}$ | $0.027_{\pm 0.007}$ |
| 50 | $0.047_{\pm 0.013}$ | $0.134_{\pm 0.019}$ | $0.018_{\pm 0.002}$ | $0.010_{\pm 0.016}$ |
| 200 | $0.040_{\pm 0.007}$ | $0.117_{\pm 0.016}$ | $0.018_{\pm 0.001}$ | $0.001_{\pm 0.001}$ |
| 10000 | $0.073_{\pm 0.027}$ | $1.111_{\pm 1.830}$ | $0.083_{\pm 0.117}$ | $0.000_{\pm 0.000}$ |

## H.2 Experiment Settings and Supplementary Results for High-dimensional Manifold with Large Number of Constraints

**Experiment Settings.** In this high-dimensional experiment, we construct a synthetic "stress-test" manifold in $\mathbb{R}^d$ by imposing $m-1$ linear equality and $l$ spherical inequality constraints inside a bounding sphere. Concretely, we generate $m-1$ random hyperplanes $h_i(x) = a_i^T x - b_i$ (with $a_i \sim \mathcal{N}(0, I_d), b_i \sim \mathcal{N}(0, 0.1^2)$) for $i \in [m-1]$, together with the sphere constraint $h_m(x) = \|x\|_2^2 - R^2$ (with $R = 5$), and $l$ spherical obstacles $g_j(x) = r^2 - \|x - c_j\|_2^2$ (with $r = 1$ and obstacle centers $c_j \sim \mathcal{N}(0, \sqrt{R/2}I_d)$) for $j \in [l]$. All randomness is fixed via a seed across experiments.

For each choice of ambient dimension $d$, the number of equality constraints $m$, and the number of obstacles $l$, we run one single chain of each algorithm for 1000 iterations. We discard the first 200 iterations as burn-in and then retain every 5th iterate, yielding 160 post-burn-in samples. Similar to 2D synthetic experiments, the baseline samplers (CLangevin, CHMC, and CGHMC) are initialized exactly on $\Sigma$ so that $X_0 \in \Sigma$, where as OLLA and OLLA-H start from noisy initialization $Y_0 = X_0 + \mathcal{N}(0, I_d), X_0 \in \Sigma$.

Table 9: Hyperparameter settings for high-dimensional manifold example ($\Delta t = 1 \times 10^{-2}$)

| Method | Hyperparameters |
|--------|-----------------|
| OLLA | $\alpha = 200, \ \epsilon = 1$ |
| OLLA–H | $\alpha = 200, \ \epsilon = 1, \ N = 5$ |
| CLangevin | $L = 5, \ \tau = 10^{-4}, \lambda = 0.1$ |
| CHMC | $\gamma = 1, \ L = 5, \ \tau = 10^{-4}, \lambda = 0$ |
| CGHMC | $\gamma = 1, \ L = 5, \ \tau = 10^{-4}, \lambda = 0$ |

We measure performance along two complementary criteria. First, we report the computational cost per effective sample size (ESS), defined as

$$\text{CPU time / ESS} := \frac{\text{total CPU runtime (s)}}{\min_{1 \le i \le d} \text{ESS}_i},$$

where $\text{ESS}_i$ is the uni-variate ESS in coordinate $i$. Second, we assess the estimation accuracy via the sample means of representative test functions—e.g. $P(x_1 > 0)$ and some complicated test functions. To isolate the effect of each problem parameter, we vary one element of $\{d, m, l\}$ at a time while holding the others fixed.

**Remark 9.** The results shown in Figure 4 and Figure 5 were obtained under a different setup as described above: a larger step size $\Delta t$ was used and the regularization parameter $\lambda$ in CLangevin, CHMC, and CGHMC was increased for improved numerical stability under large step size.

**Supplementary Results - Scaling under Dimension and the Number of Inequality Constraints** In Figure 10, we plot the performance of samplers as the ambient dimension $d$ increases from 50 to 700 (with $m = l = 5$). Across all $d$, OLLA and OLLA-H produce virtually identical estimates of our benchmark test functions similar to the baselines. In contrast, CPU time per ESS of OLLA-H stays essentially flat (on the order of $\approx 0.05$s/sample), whereas OLLA grows roughly linearly (reaching $\approx 1.1$s/sample at $d = 700$), Also, the CPU runtime plot exhibits OLLA-H achieves the lowest wall-clock times across all dimensions.

Similarly, Figure 11 shows results with dimension fixed at $d = 100$ and the number of inequality constraints $l$ varying form 10 to 60 (with $m = 5$). Again, OLLA and OLLA-H approximately match baseline methods in estimating the mean of test functions, and OLLA-H demonstrates dramatically lower CPU time per ESS and CPU runtime than both OLLA and the other baselines.

**Supplementary Results - Scaling under the Number of Equality Constraints** In Figure 12, we show how sampling performance changes as the number of equality constraints $m$ grows from 10 to 60 (with $d = 100, l = 5$, fixed $\alpha = 200$). Although OLLA-H continues to show the lowest CPU time/ESS (and total CPU time) among the methods, its sampling accuracy gradually degrades as $m$ increases, drifting away from the baseline values. Especially, we observe that the equality constraint violation worsens, and one must compensate by increasing $\alpha$, reducing $\Delta t$ with a longer chain to suppress the equality constraint violation. Each of these solutions may lead to increase of computational cost, and $\alpha$ cannot be driven arbitrarily high due to its induced discretization instabilities (Table 7). Equality-only baselines therefore achieve more accurate estimates-albeit at higher cost-indicating that large $m$ regimes are particularly challenging to OLLA-H compared to the high-dimensional or many inequality constraints settings.

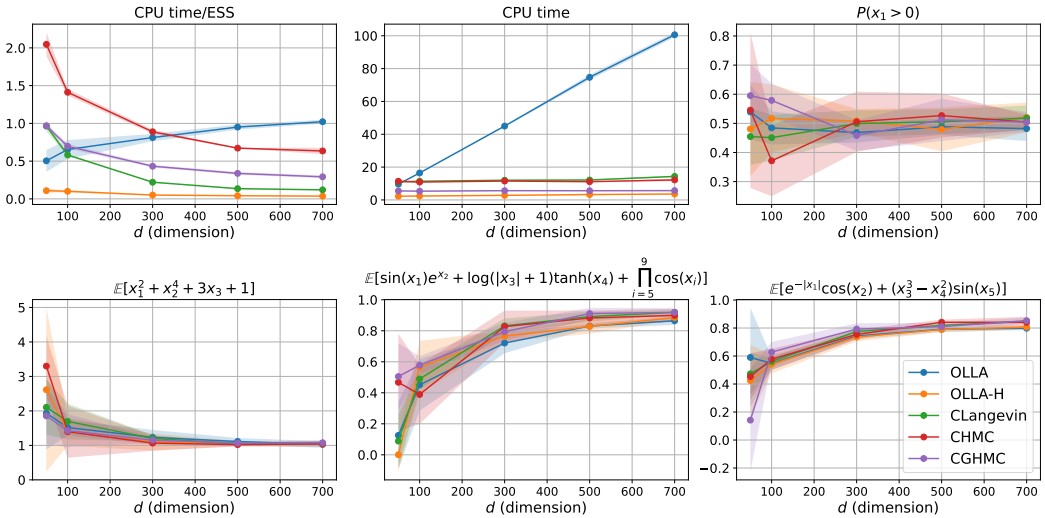

Figure 10: Scaling under the change of ambient dimension $d$. (1) CPU time per ESS (top left), (2) total CPU runtime (top center), (3) estimates of test functions (others) as the dimension $d$ increases from 50 to 700 (with $m = l = 5$).

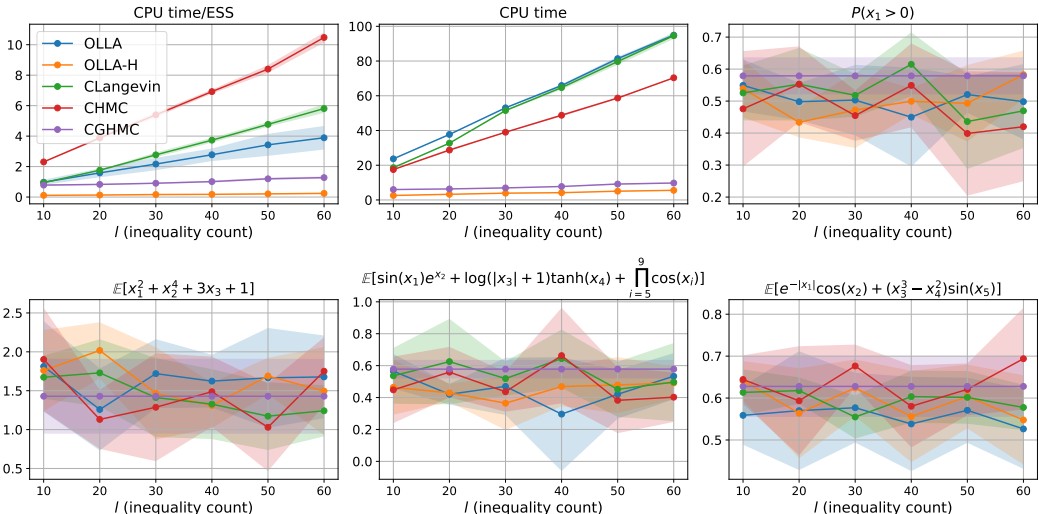

Figure 11: Scaling under the change of inequality count $l$. (1) CPU time per ESS (top left), (2) total CPU runtime (top center) (3) estimates of test functions (others) as the number of inequalities $l$ increases from 10 to 60 (with $d = 100, m = 5$). The average inequality violation is maintained at $\mathbb{E}[g(x)^+] = 0.000 \pm 0.000$ for all samplers.

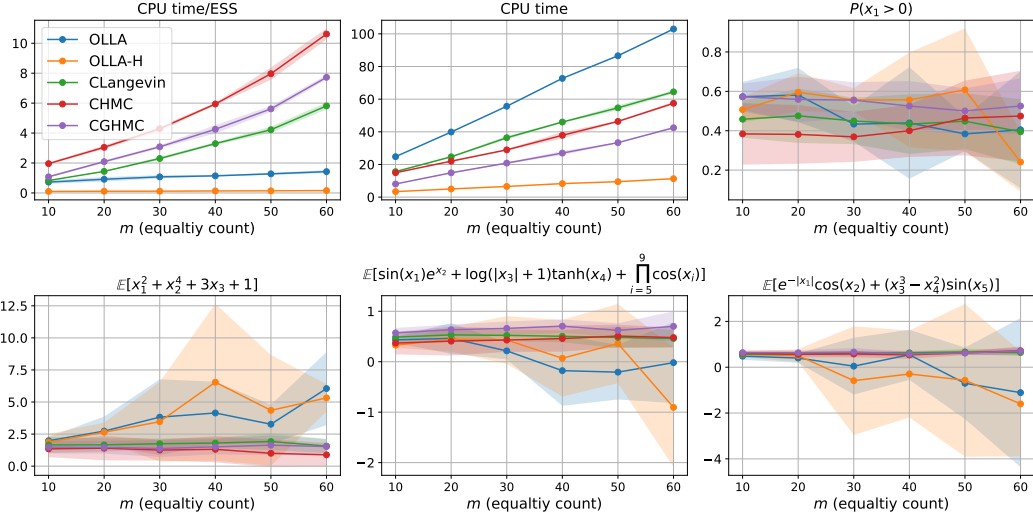

Figure 12: Scaling under the change of equality count $m$. (1) CPU time per ESS (top left), (2) total CPU runtime (top center), (3) estimates of test functions (others) as the number of inequalities $m$ increases from 10 to 60 (with $d = 100, l = 5$).

### H.3 Experiment Settings for Molecular system and Bayesian logistic regression task

**Settings.** These two experiments were executed on a Linux machine equipped with Intel Xeon Gold 6226 CPU (24 cores) and 192 GB RAM.

**Experiment Settings (Molecular System).** This experiment models a polymer chain of $N_{\text{atoms}}$ atoms in $\mathbb{R}^3$, so the state space dimension is $d = 3N_{\text{atoms}}$. Let $p \in \mathbb{R}^d$ be the flattened vector of atom positions, and let $P_k \in \mathbb{R}^3$ denote the position of the $k$-th atom (where $p$ is reshaped into an $N_{\text{atoms}} \times 3$ matrix).

The **equality constraints** $h(p) = 0$ enforce fixed bond lengths ($l_b$) between adjacent atoms and fixed angles ($\theta_a$) between consecutive bonds:

1. Bond length constraints ($k = 0, \dots, N_{\text{atoms}} - 2$):

$$h_{\text{bond},k}(p) = \|P_k - P_{k+1}\|_2^2 - l_b^2 = 0$$

2. Bond angle constraints ($k = 1, \dots, N_{\text{atoms}} - 2$): Let $v_1^k = P_{k-1} - P_k$ and $v_2^k = P_{k+1} - P_k$.

$$h_{\text{angle},k}(p) = \frac{v_1^k \cdot v_2^k}{\|v_1^k\|_2 \|v_2^k\|_2} - \cos(\theta_a) = 0$$

with $l_b = 1.0$ and $\theta_a = 109.5°$.

The **inequality constraints** ($g(p) \leq 0$) enforce steric hindrance, ensuring a minimum distance ($r_{\min}$) between non-adjacent atoms ($|i - j| \geq 2$):

$$g_{ij}(p) = r_{\min}^2 - \|P_i - P_j\|_2^2 \leq 0 \quad \text{for } j \geq i + 2$$

with $r_{\min} = 1.0$.

The **potential function** $f(p)$ models the energy of the polymer configuration and includes terms for torsion angles and non-bonded interactions based on the Weeks-Chandler-Andersen (WCA) potential. It is calculated as $f(p) = \beta(U_{\text{tor}}(p) + U_{\text{nb}}(p))$, where $\beta$ is an inverse temperature parameter (default: $\beta = 1.0$).

The torsion potential, $U_{\text{tor}}$, depends on the dihedral angles $\phi_k$ (for $k = 1, \dots, N_{\text{atoms}} - 3$) formed by consecutive bonds, where $\phi_k$ denotes the dihedral angle between atoms $P_{k-1}, P_k, P_{k+1}, P_{k+2}$.

The potential is a sum over modes $m \in M$ (default: $M = \{1, 3\}$) with corresponding force constants $k_m$ (default: $k_1 = 0.5, k_3 = 0.2$) and phase shifts $\delta_m$ (default: $\delta_1 = 0.0, \delta_3 = 0.0$):

$$U_{\text{tor}}(p) = \sum_{k=1}^{N_{\text{atoms}}-3} \sum_{m \in M} k_m(1 + \cos(m\phi_k - \delta_m)).$$

The non-bonded WCA potential, $U_{\text{nb}}$, models repulsion between atoms $i$ and $j$ that are not directly bonded and are separated by at least two bonds ($|i - j| \geq 3$). Let $R_{ij} = \|P_i - P_j\|_2$ be the distance between atoms $i$ and $j$. With the steric minimum distance $r_{\text{min}}$, its associated length scale $\sigma = r_{\text{min}}/2^{1/6}$, and the potential cutoff is $r_c = 2^{1/6}\sigma$. The interaction energy is given by:

$$U_{\text{LJ}}(R_{ij}) = 4\epsilon_{\text{WCA}} \left[ \left( \frac{\sigma}{R_{ij}} \right)^{12} - \left( \frac{\sigma}{R_{ij}} \right)^6 \right] + \epsilon_{\text{WCA}}$$

where $\epsilon_{\text{WCA}}$ is the energy scale (default: 1.0). The total non-bonded potential is the sum over eligible pairs ($j \geq i + 3$) where the distance is less than the cutoff:

$$U_{\text{nb}}(p) = \sum_{i=0}^{N_{\text{atoms}}-4} \sum_{j=i+3}^{N_{\text{atoms}}-1} U_{\text{LJ}}(R_{ij}) \cdot \mathbb{I}(R_{ij} < r_c)$$

where $\mathbb{I}(\cdot)$ is the indicator function.

We vary $N_{\text{atoms}} \in \{5, 10, 15, 20, 30\}$, corresponding to $d \in \{15, 30, 45, 60, 90\}$. We run a single chain for $K = 5000$ steps, discard the first 1000 as burn-in, and thin by 5. To measure the accuracy of sampling, we use the radius of gyration squared ($R_g^2$) as a test function, which is defined as:

$$R_g^2(p) = \frac{1}{N_{\text{atoms}}} \sum_{k=0}^{N_{\text{atoms}}-1} \|P_k - P_{cm}\|_2^2, \quad \text{where } P_{cm} = \frac{1}{N_{\text{atoms}}} \sum_{k=0}^{N_{\text{atoms}}-1} P_k$$

Table 10: Hyperparameter settings for the Molecular System example ($\Delta t = 1 \times 10^{-5}$)

| Method | Hyperparameters |
|---|---|
| OLLA-H | $\alpha = 500, \epsilon = 1.0, N \in \{0, 5\}$ |
| CLangevin | $L = 30, \tau = 10^{-4}, \lambda = 0.5$ |
| CHMC | $\gamma = 1.0, L = 30, \tau = 10^{-4}, \lambda = 0.0$ |
| CGHMC | $\gamma = 1.0, L = 30, \tau = 10^{-4}, \lambda = 0.0$ |

**Experiment Settings (Bayesian logistic regression).** This experiment involves sampling the posterior distribution of weights $\theta \in \mathbb{R}^d$ for a two-layer Bayesian neural network applied to the German Credit dataset [41]. Let $\sigma(\cdot)$ denote the sigmoid function and $\hat{p}_{\text{logit}}(\theta, x, a)$ be the network's output probability for input features $x$ and sensitive attribute $a$.

The neural network consists of an input layer, two hidden layers with ReLU activation (sizes $H_1 = 32$, $H_2 = 16$), and a final linear output layer combined with a bias term $b_0$ and a term $\alpha \cdot a$ dependent on the sensitive attribute:

$$h_1 = \text{ReLU}(W_1 x + b_1) \in \mathbb{R}^{H_1}$$
$$h_2 = \text{ReLU}(W_2 h_1 + b_2) \in \mathbb{R}^{H_2}$$
$$\hat{p}_{\text{logit}}(\theta, x, a) = w_3^T h_2 + \alpha a + b_0 \in \mathbb{R}$$

where the parameters constituting $\theta$ have dimensions: $W_1 \in \mathbb{R}^{H_1 \times \text{input\_dim}}$, $b_1 \in \mathbb{R}^{H_1}$, $W_2 \in \mathbb{R}^{H_2 \times H_1}$, $b_2 \in \mathbb{R}^{H_2}$, $w_3 \in \mathbb{R}^{H_2}$, $\alpha \in \mathbb{R}$, and $b_0 \in \mathbb{R}$. So, the total dimension $d = H_1 \cdot \text{input\_dim} + H_1 + H_2 \cdot H_1 + H_2 + H_2 + 2$ varies based on the input dimension, which itself depends on the feature hashing dimension used for categorical features. In particular, the parameter size of $\theta$ can be changed by adjusting hashing dimension.

The **potential function** is the negative log-posterior $f(v) = -(\log P(\mathcal{D}|\theta) + \log P(\theta))$, where $\mathcal{D}$ is the training data, $P(\mathcal{D}|\theta)$ is the log-likelihood using the sigmoid of the logits, and $\log P(\theta)$ is the log-prior based on an isotropic Gaussian distribution with precision $10^{-3}$.

The **equality constraints** ($h(\theta) = 0$) enforce fairness via demographic parity on True Positive Rate (TPR) and False Positive Rate (FPR) between groups defined by the sensitive attribute (default: gender) $a \in \{0, 1\}$:

$$h_{\text{TPR}}(v) = \mathbb{E}_{x,y|a=1,y=1}[\sigma(\hat{p}_{\text{logit}}(v, x, 1))] - \mathbb{E}_{x,y|a=0,y=1}[\sigma(\hat{p}_{\text{logit}}(v, x, 0))] = 0$$
$$h_{\text{FPR}}(v) = \mathbb{E}_{x,y|a=1,y=0}[\sigma(\hat{p}_{\text{logit}}(v, x, 1))] - \mathbb{E}_{x,y|a=0,y=0}[\sigma(\hat{p}_{\text{logit}}(v, x, 0))] = 0.$$

These expectations are estimated using averages over the training data subsets corresponding to each sensitive attribute $a$ and true label $y$.

The **inequality constraints** ($g(\theta) \le 0$) enforce monotonicity on selected features (default: duration, credit amount, existing credit, age) by requiring the gradient of the logit with respect to these features to have a specific sign (or be close to zero, within a margin $\delta = 1.0$) at a subset of anchor data points $\mathcal{D}_{\text{anchor}}$. Let $S^+ = \{\text{duration, credit amount, existing credits}\}$ and $S^- = \{\text{age}\}$. The constraints are formulated as:

$$g_{\text{mono}}(v) = \max \left( \max_{j \in S^+, x_i \in \mathcal{D}_{\text{anchor}}} \left\{ -\frac{\partial \hat{p}_{\text{logit}}(v, x_i, a_i)}{\partial x_{ij}} \right\}, \max_{k \in S^-, x_i \in \mathcal{D}_{\text{anchor}}} \left\{ \frac{\partial \hat{p}_{\text{logit}}(v, x_i, a_i)}{\partial x_{ik}} \right\} \right) - \delta.$$

The dimension $d$ varies based on feature hashing as $d \in \{706, 1986, 4994, 9986, 49986, 100002\}$. We run a single chain for $K = 200$ steps, discard the first 40 as burn-in, and thin by 2. Also, we use the test Negative Log-Likelihood (NLL) as the metric.

Table 11: Hyperparameter settings for the Bayesian logistic regression task

| Method | Hyperparameters |
| --- | --- |
| OLLA-H | $\alpha = 100, \epsilon = 1.0, N \in \{0, 5\}, \Delta t = 5 \times 10^{-4}$ |
| CLangevin | $L = 10, \tau = 1.0, \lambda = 0.5, \Delta t = 5 \times 10^{-4}$ |
| CHMC | $\gamma = 1.0, L = 10, \tau = 1.0, \lambda = 0.5, \Delta t = 5 \times 10^{-3}$ |
| CGHMC | $\gamma = 1.0, L = 10, \tau = 1.0, \lambda = 0.5, \Delta t = 5 \times 10^{-3}$ |

