# OpenReview forum: "Fast Non-Log-Concave Sampling under Nonconvex Equality and Inequality Constraints with Landing"
_NeurIPS.cc/2025/Conference — NeurIPS 2025 poster_

### Official Review · Reviewer_AMyy · 2025-06-30

**Clarity:** 3
**Significance:** 3
**Originality:** 3
**Rating:** 5
**Confidence:** 2

**Summary:**

This paper introduces Overdamped Langevin with Landing (OLLA), a novel projection-free sampling algorithm for distributions defined by complex, and potentially non-convex, equality and inequality constraints. By decomposing Langevin dynamics into tangential and normal components and enforcing constraint satisfaction via an exponential "landing" drift, OLLA provably converges exponentially fast in 2-Wasserstein distance under a log-Sobolev inequality (LSI) assumption. The authors also present a practical Euler–Maruyama discretization (OLLA-H) using a Hutchinson estimator for curvature corrections, achieving only $O(N)$ gradient costs per step.

**Questions:**

Do the analysis and convergence guarantees extend naturally to high dimensions? If not, could the authors discuss the challenges in making this dependence explicit?

**Ethical Concerns:**

["NO or VERY MINOR ethics concerns only"]

**Final Justification:**

My concerns have been adequately addressed, and I have updated my score accordingly. The additional theoretical insights and experimental results provided in the rebuttal helped clarify my key concerns. I encourage the authors to include these additions in the revised version to strengthen the final paper.

**Limitations:**

There are no limitations or potential negative societal impacts of their work.

**Paper Formatting Concerns:**

There are no paper formatting concerns.

**Quality:**

4

**Strengths And Weaknesses:**

**Strengths**

* The authors provide a unified framework to handle arbitrary smooth equality and inequality constraints without explicit projections.
* The authors provide rigorous non-asymptotic exponential convergence guarantees in the 2-Wasserstein distance, leveraging LSI constants.
* Practical discretization via Hutchinson trace estimation reduces per-step cost to $O(N)$ gradient evaluations.
* The paper is generally well-structured.

**Weaknesses**
* The theoretical convergence rate has a similar mathematical form to that of unconstrained Langevin algorithms under good conditions, depending on an LSI constant. Intuitively, one might expect that constraining the sampling domain would improve the LSI constant and lead to faster convergence, especially as the domain diameter shrinks [1-3]. The paper does not provide such an analysis or an estimate of the LSI constant in terms of the domain geometry. It is unclear how difficult such an analysis would be. Could authors comment on that, and if possible, offer a theoretical advantage in convergence rate?
* The experimental evaluation is confined to synthetic tasks. As it is a paper with a strong theoretical focus, this is understandable.
* Typos: line 210: "Metropolis-Hasting correction" -> "Metropolis-Hastings correction"

**References**

[1] Reflection Couplings and Contraction Rates for Diffusions, Probability Theory and Related Fields, 2016

[2] Sampling from a Log-Concave Distribution with Projected Langevin Monte Carlo, Discrete & Computational Geometry, 2018

[3] Projected Stochastic Gradient Langevin Algorithms for Constrained Sampling and Non-Convex Learning, COLT, 2021

---

> ### Author Rebuttal · Authors · 2025-07-31
>
> We appreciate the reviewer for the constructive feedback. Below, we provide itemized responses to address each of the comments, and we hope they clarify your questions and concerns.
>
> ---
> >Weakness (1): The theoretical convergence rate has a similar mathematical form to that of unconstrained Langevin algorithms under good conditions, depending on an LSI constant. Intuitively, one might expect that constraining the sampling domain would improve the LSI constant and lead to faster convergence, especially as the domain diameter shrinks [1-3]. The paper does not provide such an analysis or an estimate of the LSI constant in terms of the domain geometry. It is unclear how difficult such an analysis would be. Could authors comment on that, and if possible, offer a theoretical advantage in convergence rate?
>
> **Answer.** Thank you for providing insightful feedback. We agree that we missed this analysis in our paper. Below we (1) state a geometric lower bound for the LSI constant on $\Sigma$, (2) explain how shrinking diameter improves the on-manifold mixing rate and how this interacts with the landing rate $\alpha$ in our bound.
>
> **Geometric control of $\lambda\_{\text{LSI}}.$** First, we bring up the following informal theorem (Theorem 7.3., [2]):
> * Theorem (Informal) Let $\Sigma$ be a compact Riemannian manifold with diameter $D = \sup_{x,y \in \Sigma} d_\Sigma(x,y)$ and non-negative Ricci curvature. Then, $\lambda_{LSI} \geq \frac{\lambda_1}{1+2D\sqrt{\lambda_1}}$, or $\lambda_{LSI} \geq \frac{\pi^2}{(1+2\pi)D^2}$ holds, where $\lambda_1$ is the first eigenvalue of the Laplace-Beltrami operator.
>
> This implies if the constraints shrink the diameter $D$ (or increase $\lambda_1$), the lower bound of $\lambda\_{\text{LSI}}$ increases, so the on-manifold decay of $\textsf{KL}^\Sigma(\rho_t || \rho_\Sigma)$ acceraltes (dominated by exponential rate $-2\lambda\_{\text{LSI}})$
>
> **How this yields a theoretical advantage in our bound.** Our non-asymptotic bound reads, for sufficently large enough $t \geq t_{cut}$,
> \begin{equation}
>     W_2(\rho_t, \rho_\Sigma) \leq \frac{M_h}{\kappa} e^{-\alpha t} + \sqrt{\frac{2}{\lambda\_{\text{LSI}}} \textsf{KL}^{\Sigma} (\tilde{\rho}_t || \rho\_{\Sigma}) }
> \end{equation}
> With $\textsf{KL}^{\Sigma} (\tilde{\rho}_t || \rho\_{\Sigma}) = \mathcal O(e^{-2(\lambda\_{\text{LSI}} (t-t\_{cut}))})$, the second term decays at a scale $\mathcal O(e^{-\lambda\_{\text{LSI}} t})$. Thus the effective rate is controlled by $\min \\{\alpha, \lambda\_{\text{LSI}} \\}$. By this analysis, we can observe that stronger constraints that reduce $D$ make $\lambda\_{\text{LSI}}$ larger, so, provided $\alpha$ is not the bottleneck, the overall convergence rate in $W_2$ improves.
>
> ---
> >Question (1): Do the analysis and convergence guarantees extend naturally to high dimensions? If not, could the authors discuss the challenges in making this dependence explicit?
>
> **Answer.** Our convergence guarantees have some dependency on dimension $d$. The rates $\alpha, \lambda\_{\text{LSI}}$ are dimension-free, but the explicit $d$-dependence enters through the additive constant, which we can make explicit and a little sharper than now.
>
> **Where the dimension dependency enters.** To simplify the setup, we illustrate this using Theorem 1 (Same thing can be applied to mixed scenario). In theorem 1, there exists a constant $C_7:= \frac{C_6 e^{-\alpha t_{cut}}}{\alpha - 2\lambda\_{\text{LSI}}}$ (there is an errata; missing $C_6$), and $C_6$ depends on the constant $C_{div}$ by $C_6 = \mathcal O (C_{div}^2)$, which again depends on $\mathcal O (\sqrt{d(d-m)})$ by the proof of Lemma E.5 in appendix. However, we revised the proof by using better linear algebra techniques and were able to conclude that we can reduce the dependence of $C_{div}$ by $\mathcal O (\sqrt{d-m})$.
>
> Therefore, we can say our convergence rate depends on $\mathcal O (\sqrt{d-m})$ (due to the square-root on $\textsf{KL}^\Sigma$) in the equality-only case (same analysis can be applied to mixed cases). We will incorporate this refinement and reflect this dependency in our paper.
>
> ---
> >Weakness (2): The experimental evaluation is confined to synthetic tasks.
>
> **Answer.** We appreciate this concern. For the related answers, please refer to our responses to tmLS and 74bE, where we added two real-world benchmarks with full setup and results.
>
> ---
> >Suggestion (1): Typos in Line 210: "Metropolis-Hasting correction" $\rightarrow$ "Metropolis-Hastings correction".
>
> **Answer.**  Thank you for pointing this out. We will fix this.
>
> ---
> Thank you again for your positive review and helpful feedback. We'd appreciate the opportunity to discuss further if you have any additional questions or suggestions.
>
>
> ### Reference
> [2] M. Ledoux (1999). Concentration of measure and logarithmic Sobolev inequalities. Séminaire de Probabilités 33.

---

### Official Review · Reviewer_74bE · 2025-07-01

**Clarity:** 3
**Significance:** 3
**Originality:** 3
**Rating:** 5
**Confidence:** 4

**Summary:**

This paper proposes OLLA, a projection-free Langevin dynamics method for sampling under nonconvex equality/inequality constraints. By combining tangential noise with normal drift ("landing"), OLLA enforces constraints without projections or slack variables. Theoretical analysis shows exponential convergence in Wasserstein distance, while a practical variant (OLLA-H) uses trace estimation for scalability. Experiments demonstrate superior efficiency over baselines in high dimensions.

**Questions:**

* How should practitioners choose $(\alpha, \epsilon)$ for real-world problems? Are there heuristics or adaptive schemes?
* How does OLLA behave when constraints violate LICQ (e.g., degenerate gradients)?
* The paper notes degraded performance for large $m$. Is there a theoretical or empirical threshold beyond which OLLA becomes impractical?

**Ethical Concerns:**

["NO or VERY MINOR ethics concerns only"]

**Final Justification:**

The authors answered my questions and addressed my concern, and thus I would like to keep my positive score.

**Limitations:**

Yes

**Quality:**

4

**Strengths And Weaknesses:**

Strengths:
* This paper introduces a novel SDE design with a landing term for constraint enforcement, combined with rigorous non-asymptotic convergence analysis for three constraint scenarios.
* This paper addresses a critical gap in constrained sampling, especially for nonconvex sets where projections fail, which also has practical impact for Bayesian inference, generative modeling, and other ML applications with constraints.
* This is the first method to unify equality/inequality constraints in Langevin dynamics without projections.

Weaknesses:
* Critical assumptions are deferred to the appendix, making it harder to evaluate the method’s applicability upfront. This obscures potential limitations, such as sensitivity to constraint degeneracy or nonsmoothness, which should be highlighted in the main text.
* Hyperparameters $(\alpha, \epsilon)$ are sensitive but lack principled selection guidelines.
* Limited to synthetic experiments; real-world benchmarks would strengthen the impact.

---

> ### Author Rebuttal · Authors · 2025-07-31
>
> We appreciate the reviewer for the constructive feedback. Below, we provide itemized responses to address each of the comments, and we hope they clarify your questions and concerns.
>
> ---
> >Weakness (3): Limited to synthetic experiments; real-world benchmarks would strengthen the impact.
>
> **Answers.** We added two real-world tests that demonstrate the advantages of OLLA-H under complicated potentials and domain constraints. For the experimental results, please refer to the results we provided to reviewer tmLS.
>
> **Experiment (A): Molecular/Polymer system.**  In this experiment, we have $N$ atoms $P = (P_0, ..., P_{N-1}) \in (\mathbb R^3)^N$, $(d = 3N)$ and we sample from the complicated potential $f(P) = \sum_{i <j, |i-j|>1} \phi_{nb}(P_i, P_j) + \sum_{k=1}^{N-3} \phi_{tors}(P_{i-1}, P_i, P_{i+1}, P_{i+2})$, where $\phi_{nb}$ is 12-degree piece-wise polynomials (e.g. nonbonded pair potential), $\phi_{tors}$ is highly nonlinear function involving periodic cosine function (e.g. Periodic dihedral torsion potential).
>
> The equality constraints are defined by $h_k^{bond}(P) = ||P_k-P_{k+1}||_2^2 - l^2 = 0, \ k =0, ..., N-2$, and $h_k^{ang}(P) = (P\_{k-1} - P\_{k}) \cdot (P\_{k+1} - P\_{k}) - l^2 \cos(\theta), k=1,...,N-2$, and the inequality constraints are given by $g\_{ij}(P) = r\_{min}^2 - ||P_i-P_j||_2^2$ for $|i-j| > 1$.
>
> **Experiment (B): UCI German Credit: Bayesian 2-layer logistic regression with fairness + monotonicity constraints.** In this experiment, we use a two-layer ReLU neural network with hashed categoricals (to incorporate categorical data). The potential function $f(\theta)$ is the posterior distribution with an isotropic Gaussian prior.
>
> The equality constraints are $h_{TPR}(\theta) = \mathbb E [p_\theta(X,A) \mid Y=1, A=1] - \mathbb E [p_\theta(X,A) \mid Y=1, A=0]$, $h_{FPR}(\theta) = \mathbb E [p_\theta (X,A) \mid Y=0, A=1] - \mathbb E [p_\theta(X,A) \mid Y=0, A=0]$, where $Y$ is the binary outcome label (e.g. good/bad credit), $A$ is the sensitive attribute (e.g. gender), and $p_\theta (X,A)$ is the predicted probability based on $X,A$.
>
> The inequality constraints are set to be $g_+(\theta) = \max_{r,j \in S_+} \\{ -\partial_{x_j} l_\theta (z_r)\\}$, $g_-(\theta) = \max_{r,j \in S_-} \\{\partial_{x_j} l_\theta (z_r)\\}$, where $S_+$ is the index set of monotone-increasing features, $S_-$ is the index set of monotone-decreasing features, $l_\theta(z)$ is logit value based on $z = (x,a)$, and $z$ is anchor points to check monotonicity.
>
> **Summarized Results.** In scenarios where the set of active inequality constraints is sparse (i.e.$|I_x|$ is small) or the manifold geometry is especially intricate so that each projection requires many Newton iterations, OLLA-H proves more computationally efficient than other baseline algorithms.
>
> ---
>
> >Weakness (2) & Question (1): Hyperparameters $(\alpha, \epsilon)$ are sensitive but lack principled selection guidelines. How should practitioners choose for real-world problems? Are there heuristics or adaptive schemes?
> >Question (3): The paper notes degraded performance for large $m$. Is there a
> >theoretical or empirical threshold beyond which OLLA becomes impractical?
>
>
> **Answers.** Below we give a principled rule and a practical recipe.
>
> **Landing rate $\alpha$.** First, we observe that a monotonically decreasing upper bound of $W_2$ can be achieved whenever $\alpha > 2\lambda_{LSI}$ (from theorem 3). However, $\lambda_{LSI}$ is unknown in most cases. So, we rely on the following informal theorem (Theorem 7.3., [2]):
> * Theorem (Informal) Let $\Sigma$ be a compact Riemannian manifold with diameter $D = \sup_{x,y \in \Sigma} d_\Sigma(x,y)$ and non-negative Ricci curvature. Then, $\lambda_{LSI} \geq \frac{\lambda_1}{1+2D\sqrt{\lambda_1}}$, or $\lambda_{LSI} \geq \frac{\pi^2}{(1+2\pi)D^2}$ holds, where $\lambda_1$ is the first eigenvalue of the Laplace-Beltrami operator.
>
> Therefore, we start searching reasonable $\alpha$ at a scale of $\mathcal O(1/D^2)$, which not only decreases equality constraints fast but also maintains numerical stability of discretization.
>
> **Step size $\Delta t$.** For choosing a reasonable step size, we note the previous analysis on the decay of $h(X_k)$. Because $\mathbb E [h(X_k)]$ decays exponentially fast with base $(1-\alpha \Delta t)$, we choose a reasonable $\Delta t > 0$ such that $\alpha \Delta t < 1$ with some step size searching.
>
> **Boundary repulsion rate $\epsilon$.** This term plays a crucial role in satisfying the inequality constraints. Too small $\epsilon$ will make trajectories sticky near the boundary of $\Sigma$, and too large $\epsilon$ will make aggressive repulsion on the boundary so that the sample overshoots.
> In practice, we monitor the inequality violation during sampling and adjust $\epsilon$ so that the sample does not overshoot and show sticky behavior at the boundary. Fortunately, it turns out that $\epsilon$ is less sensitive than $\alpha$ (Table 6 in Appendix), and it is often not hard to choose a good $\epsilon$.
>
> **Adaptive method for choosing $\alpha.$**
> In our high-dimension stress test, we observed that keeping $\alpha$ fixed while increasing the number of equalities $m$ slows the convergence rate (Figure 12, Appendix H). In contrast to this, this phenomenon was improved by adaptively choosing $\alpha$; increasing $\alpha$ with $m$ mitigates this problem (described in Section 5, Appendix H.2). A plausible explantation is that, as $m$ grows, the feasible manifolds's diameter $D$ shrinks, so the requirement $\alpha > 2\lambda_{LSI}$ effectively calls for a larger $\alpha$.
>
> **Threshold hold of $m$ beyond which OLLA-H becomes ineffective.**
> However, when $\alpha$ is increased, we also need to reduce the step size to keep $\alpha \Delta t <1$ to ensure discretization stability. If we combine the previous two criteria to choose $\alpha$, we obtain a two-sided bound $\mathcal 1/D^2 \lesssim \alpha \leq 1/\Delta t$.
>
> This implies that if the lower bound $D^{-2}$ grows with $m$ (as observed in our stress test), then a fixed $\Delta t$ will eventually violate $\alpha \Delta t < 1$. Consequently, $\Delta t$ must be reduced as $m$ increases. At that point, to cover the same physical time horizon $T$, the step count $K = T/\Delta t$ increases, starting to make OLLA-H ineffective compared to other baselines.
>
> ---
> >Question (2): How does OLLA behave when constraints violate LICQ (e.g., degenerate gradients)?
>
> **Answers.** LICQ is used in the theory to guarantee a uniformly well-conditioned normal space (also invertibility of Gram matrix $G$). In practice, OLLA/OLLA-H remains well-defined and typically stable under LICQ failures (e.g., degenerated/colinear constraints) by using a Moore-Penrose pseudoinverse $G^\dagger$ (Remark 4, Appendix B).
>
> To understand how this can mitigate the problem, we recall the induced definition of orthogonal projector $\Pi(x):= I-\nabla J(x)^T G(x)^\dagger \nabla J(x)$, which again becomes a idempotent projector onto $\text{Null}(\nabla J(x)) = T_x\Sigma$. Hence, $\Pi(x)$ always projects onto the tangent space $T_x\Sigma$.
>
> * Example (redundant gradient): Let $h(x,y) = x^2+y^2 - 1$, $g(x,y) = x^2 + y^2 -\frac{3}{2}$.
> In this example, $\nabla h, \nabla g$ are colinear, making $G$ singular. However, $\Pi(x):= I-\nabla J(x)^T G(x)^\dagger \nabla J(x)$ still projects onto the circle's tangent line, which is precisely the intended behavior of our algorithm.
> ---
>
> >Weakness (1): Critical assumptions are deferred to the appendix, making it harder to evaluate the method’s applicability upfront. This obscures potential limitations, such as sensitivity to constraint degeneracy or nonsmoothness, which should be highlighted in the main text.
>
> **Answers.** Thank you for pointing this out. This point was also raised by reviewer cZ9Z. We will move the key assumptions to the main text with short intuitions and reasonable examples to understand them. The following response briefly highlights the major changes; for detailed assumptions and clarifications, please see our reply to reviewer cZ9Z.
>
> **Reason to introduce new assumption (M1'').** One of our original mixed-case assumption (M1) asked the projected manifold $\Sigma_p := \pi( \\{h(x) =p , g(x) \leq 0\\})$ to lie in $\text{int} (\Sigma)$ for small enough $||p||_2>0$, which can be strong in practice. To mitigate this, we propose the following assumption (M1'')
> * Assumption (M1''): Define $p_t$ = $P(\pi(X_t) \notin \text{int}(\Sigma))$. we assume that after sufficently large time $t_{bd}$, there exist $\alpha_{bd}, C_{bd}, \epsilon_{bd} >0$ such that $p_t \leq C_{bd}e^{-\alpha_{bd}t}$.
> * Corollary (informal): Under assumptions (C1)-(C3) and (M1'')-(M3), for $t \geq t_{bd}$ and $\alpha > 2\lambda_{LSI}, \beta \geq 1$, it holds that
> \begin{equation}
>     W_2(\rho_t, \rho_\Sigma) \leq \frac{M_h}{\kappa} e^{-\alpha t} + \sqrt{\frac{2}{\lambda_{LSI}} \textsf{KL}^{\Sigma} (\tilde{\rho}\_{t}^{\Sigma\_{in}} || \rho\_{\Sigma})  + C\_{bd}e^{-\alpha\_{bd}t}\cdot \text{diam}\_{\Sigma} (\Sigma)^2}
> \end{equation}
> where $\textsf{KL}^{\Sigma}(\tilde{\rho}_{t}^{\Sigma\_{in}} || \rho\_{\Sigma}) = \mathcal O(\exp(-2\lambda\_{LSI} t))$ as in Theorem 3.
>
> This implies when inequalities and initial distributions are nice (e.g. (M1) holds or $\Sigma_p$ contracts to $\Sigma$ quickly), the last term (including $\text{diam}\_{\Sigma} (\Sigma)$) decays rapidy and the overall rate can be improved; otherwise that term becomes the bottleneck.
>
> ---
> Thank you again for your positive review and helpful feedback. We'd appreciate the opportunity to discuss further if you have any additional questions or suggestions.
>
>
> ### Reference
> [2] M. Ledoux (1999). Concentration of measure and logarithmic Sobolev inequalities. Séminaire de Probabilités 33.

---

> > ### Comment · Reviewer_74bE · 2025-08-04
> >
> > Thanks for the detailed response. I acknowledge the rebuttal and would like to keep my current score.

---

> > > ### Author Response · Authors · 2025-08-08
> > >
> > > Thank you for your constructive feedback, questions, and your time. We will reflect the outcomes of these discussions in the final version.

---

### Official Review · Reviewer_cZ9Z · 2025-07-02

**Clarity:** 2
**Significance:** 3
**Originality:** 3
**Rating:** 4
**Confidence:** 2

**Summary:**

This paper studies the problem of sampling from a density satisfying log-Sobolev inequality with nonlinear equality and inequality constraines, even when the feasible set is nonconvex. The authors propose a novel framework, OLLA, which modifies Langevin dynamics to enforce constraints via a landing mechanism. By operating on the tangent space of the constraint manifold, OLLA avoids costly projections. The method achieves exponential convergence under some assumptions and is efficiently discretized using Hutchinson trace estimation. Empirical results demonstrate that OLLA matches or exceeds the performance of existing constrained sampling algorithms in both accuracy and computational efficiency, particularly in high-dimensional settings.

**Questions:**

How should one select the landing rate and step size?

Since there is no theoretical analsysis for OLLA-H, how is the Hutchinson trace estimator affect the convergence rate?

Could you clarify the assumptions underlying the main theorems and illustrate concrete settings in which they hold?

**Ethical Concerns:**

["NO or VERY MINOR ethics concerns only"]

**Final Justification:**

The authors' rebuttal addressed my concerns. I will keep my positive score.

**Limitations:**

yes

**Quality:**

3

**Strengths And Weaknesses:**

Strengths:

This paper propose a novel sampling framework combined with projection free landing mechanism in optimization and shows exponential convergence guarantees in 2-Wasserstein distance under mild LICQ and log-Sobolev assumptions. To make the algorithm more efficient they further modify the algorithm by using Hutchinson trace estimation, yielding low per-step cost even in high dimensions.

Weakness:

The stated assumptions are unclear and may not hold in practical scenarios, and the algorithm is highly sensitive to its hyperparameters, yet the authors provide no guidance on how to select them.

---

> ### Author Rebuttal · Authors · 2025-07-31
>
> We appreciate the reviewer for the constructive feedback. Below, we provide itemized responses to address each of the comments, and we hope they clarify your questions and concerns.
>
> ---
> >Weakness (1): The stated assumptions are unclear and may not hold in practical scenarios.
> >Question (3): Could you clarify the assumptions underlying the main theorems and illustrate concrete settings in which they hold?
>
> **Answer.** Below we restate them concisely, provide settings where they hold, and for the mixed case, we propose a milder replacement (M1'') of the strong assumption (M1) together with a corollary.
>
> **Core assumptions (C1) - (C3)**
> First, we recall our notations $\Sigma := \\{x \in \mathbb R^d \mid h(x) =0 , \ g(x) \leq 0\\}$
> * Assumption (C1)-LICQ: For every $y \in \Sigma$, $\\{\nabla h_i(y)\\}\_{i \in [m]}  \cup \\{\nabla g_j(y)\\}\_{j \in I_y}$ is linearly independent.
>
> This assumption guarantees a uniformly well-conditioned normal space and underlies the projection/landing analysis.
>
> * Assumption (C2)-Regularity of $\Sigma$: $\Sigma$ is assumed to be compact and connected. Also, $\nabla h(x) \neq 0$ on $\Sigma$ and $h$ is coercive ($||h(x)||_2 \rightarrow \infty$ as $||x||_2 \rightarrow \infty$). When there are only inequality constraints, we assume $\text{dim}(\Sigma) = d$.
>
> This assumption are required for the existence of $\lambda_{LSI}$ (from C4 in Appendix) and linkage between constraint violation and normal displacement: smaller $||h(x)||_2, \ g(x)^+$ implies smaller $||x-\pi(x)||_2$ within its recoverable tubular neighborhood.
>
> * Assumption (C3)-Bounded initialization: For the initial law $\rho_0$, we assume $M_h := \sup\_{x_0 \in \text{supp}(\rho_0)} ||h(x)||_2 < \infty, M_g :=\sup\_{x_0 \in \text{supp}(\rho_0)} ||g(x)||_2 < \infty$.
>
> This ensures a finite landing time (almost surely) into the recoverable tubular neighborhood (for equality constraints) or the constrained domain $\Sigma$ (for inequality constraints).
>
> **Examples satisfying (C1)-(C3).** Some of classical manifolds can satisfy assumptions (C1)-(C3). For example, sphere $S^{d-1} := \\{x \in \mathbb R ^d \mid ||x||_2 = 1\\}$, Stiefel manifold $St(n, p) := \\{X \in \mathbb R^{n \times p} \mid X^T X = I_p\\}$ (with $n > p$ ), Tori $T^k$.
>
> **Mixed-case only assumptions (M1) - (M3)**
> Our mixed-case analysis introduced the assumption (M1) to control how the projected manifold $\Sigma_p$ intersects the interior of $\Sigma$.
> * Assumption (M1)-Regularity of $\Sigma_p$: The projected manifold $\Sigma_p := \pi ( \\{ x \in \mathbb R^d \mid h(x) = p, g(x) \leq 0 \\})$ lies inside the interior of $\Sigma$ for $0 < ||p||_2 < \delta$, where $\delta$ is the width of recoverable tubular neighborhood $\hat{U}\_{\delta}(\Sigma)$.
>
> We agree that (M1) can be strong in many practical geometries. To understand why this is assumed. We can think of the following examples. We set $h(x,y) = 0$ and define the equality-only projection $\pi^h(x,y) = (x,0)$ (the nearest-point projection onto $\\{y=0\\}$). Also, we write $\Sigma:= \\{(x,0) \in \mathbb R^2 \mid g(x,0) \leq 0\\}$, $\Sigma_p := \pi^h ( \\{(x,y) \in \mathbb R^2 \mid y =p, g(x,y) \leq 0\\})$.
> * Example (1)-satisfy (M1): pick $g_{circ}(x,y) = 2(x^2+y^2) - 1 \leq 0$
>
> In this case, $\Sigma_p$ lies inside the $\text{int}(\Sigma)$ for $p \in (-1, 1)$, and satisfy the assumption (M1). Also, we note that the projected process $Y_t = \pi(X_t)$ fully lies on $\text{int}(\Sigma)$, not stuck at the boundary of $\Sigma$ (Here, $\pi$ is the neareast-point projection onto $\Sigma$).
>
> * Example (2)- violates (M1): pick $g_{hyp}(x,y) = 2(x^2-y^2) -1 \leq 0$
>
> In this case, while $\Sigma$ is identical to example (1), $\Sigma_p := \\{(x,0) \in \mathbb R^2 \mid |x| \leq \sqrt{\frac{1}{2} +p^2}\\} \not\subset \text{int}(\Sigma)$ and fails to satisfy (M1). In this example, the projected process $Y_t$ can stuck at the boundary when $\pi^h(X_t) \not\in \text{int}(\Sigma)$, slowing down the convergence of $Y_t$.
>
> * Example (3)- dramatric case: pick $g_{\text{loose}}(x,y) = 2(x^2 - 1000y^2) -1$.
>
> This inequality constraint can define the same $\Sigma$ defined on Example (1) or (2). However, under $g_{\text{loose}}$, the feasible region of this inequality stretches far along $\Sigma$, so boundary hits persits and the process $\pi^h(X_t)$ routinely overshoots $\Sigma$. This behavior degrades the sampling efficiency when $X_t$ hovers relatively high on $\Sigma$, even though $h(X_t)\rightarrow 0$ at the end.
>
> To incorporate this, we propose a milder assumption (M1'') below:
> * Assumption (M1''): Define $p_t$ = $P(\pi(X_t) \notin \text{int}(\Sigma))$. we assume that after sufficently large time $t_{bd}$, there exist $\alpha_{bd}, C_{bd}, \epsilon_{bd} >0$ such that $p_t \leq C_{bd}e^{-\alpha_{bd}t}$.
> * Corollary (informal): Under assumptions (C1)-(C3) and (M1'')-(M3), for $t \geq t_{bd}$ and $\alpha > 2\lambda_{LSI}, \beta \geq 1$, it holds that
> \begin{equation}
>     W_2(\rho_t, \rho_\Sigma) \leq \frac{M_h}{\kappa} e^{-\alpha t} + \sqrt{\frac{2}{\lambda_{LSI}} \textsf{KL}^{\Sigma} (\tilde{\rho}\_{t}^{\Sigma\_{in}} || \rho\_{\Sigma})  + C\_{bd}e^{-\alpha\_{bd}t}\cdot \text{diam}\_{\Sigma} (\Sigma)^2}
> \end{equation}
> where $\textsf{KL}^{\Sigma}(\tilde{\rho}_{t}^{\Sigma\_{in}} || \rho\_{\Sigma}) = \mathcal O(\exp(-2\lambda\_{LSI} t))$ as in Theorem 3.
>
> To comment on $p_t$, if the inequality constraints behave nicely so that (M1) is achieved or the feasible set of inequality constraints contracts to $\Sigma$ very fast, the $p_t$ term decays fast, leading to a better convergence rate. This aligns with our observation from previous examples.
>
> The remaining assumptions (M2) and (M3) are comparably milder than (M1). For the assumption (M2), we can pick sufficiently high $V, \beta$ accordingly to $\Sigma$, because we only require $V > 0, \beta>1$. Also, we left the comment of Assumption (M3) on Remark 5 (in Appendix A), and this is likely to be satisfied for large enough $t$ due to uniformly elliptic behavior of the associated Fokker-Planck equation.
>
> ---
> >Question (2): Since there is no theoretical analysis for OLLA-H, how does the Hutchinson trace estimator affect the convergence rate?
> >
> **Answer.** The stochastic trace is used only in the mean-curvature term $\mathcal H$. In our formulation, $\mathcal H$ is multiplied by $\nabla J^T G^{-1}$ (normal operator), hence the Hutchinson noise acts only in the normal direction and tangential mixing on $\Sigma$ is minimally affected. What change is the second-order discretization bias in the landing dynamics, which we will show below:
> * (Decomposition) Let $Y_t := \pi(X_t)$ and $Z_t$ be the target process ($Z_t \sim \rho_\Sigma$). Then, the distance between $X_t$ and $Z_t$ is decomposed into $\mathbb W_2(\rho_t, \rho_\Sigma) \leq E||X_t-Z_t||_2 \leq E||X_t-\pi(X_t)||_2 + E||Y_t-Z_t||_2$
>
> The first term primarily depends on $\alpha$ and the statistics of $\mathcal H$, while the second term heavily depends on the on-manifold geometry (e.g., LSI constant) and should be minimally affected by the Hutchinson noise (from our continuous dynamic analysis in Theorem 3). Therefore, we analyze how the noise affects the first term.
> * Lemma (Informal, equality-only, $m=1$). Suppose $||X_k||_2$ is bounded for $k\geq 1$ a.s. Then, we have
> \begin{equation}
>     \mathbb E [h(X_k)] \leq (1-\alpha \Delta t)^K \mathbb E[h(X_0)]  + \mathcal{O} (\frac{\Delta t}{\alpha} ( (1+ \frac{d^2}{N}))
> \end{equation}
> where $K$ is the total number of iterations.
>
> Therefore, the Hutchinson estimator may not affect the converence rate, but rather increase the discretization error, which again vanishes as $N \rightarrow \infty$.
>
> ---
> >Weakness (2): The algorithm is highly sensitive to its hyperparameters, yet the authors provide no guidance on how to select them.
> >Question (1): How should one select the landing rate and step size?
> >
> **Answer.**  Below we give a principled rule and a practical recipe.
>
> **Landing rate $\alpha$.** First, we observe that a monotonically decreasing upper bound of $W_2$ can be achieved whenever $\alpha > 2\lambda_{LSI}$ (from Theorem 3). However, $\lambda_{LSI}$ is unknown in most cases. So, we rely on the following informal theorem (Theorem 7.3, [2]):
> * Theorem (Informal) Let $\Sigma$ be a compact Riemannian manifold with diameter $D = \sup_{x,y \in \Sigma} d_\Sigma(x,y)$ and non-negative Ricci curvature. Then, $\lambda_{LSI} \geq \frac{\lambda_1}{1+2D\sqrt{\lambda_1}}$, or $\lambda_{LSI} \geq \frac{\pi^2}{(1+2\pi)D^2}$ holds, where $\lambda_1$ is the first eigenvalue of the Laplace-Beltrami operator.
>
> Therefore, we start searching a reasonable $\alpha$ at a scale of $\mathcal O(1/D^2)$, which not only decreases equality constraints fast but also maintains numerical stability of discretization.
>
> **Step size $\Delta t$.** For choosing a reasonable step size, we note the previous analysis on the decay of $h(X_k)$. Because $\mathbb E [h(X_k)]$ decays exponentially fast with base $(1-\alpha \Delta t)$, we choose a reasonable $\Delta t > 0$ such that $\alpha \Delta t < 1$ with some step size searching.
>
> **Boundary repulsion rate $\epsilon$.** These terms play a crucial role in satisfying the inequality constraints. Too small $\epsilon$ will make trajectories sticky near the boundary of $\Sigma$, and too large $\epsilon$ will make aggressive repulsion on the boundary so that the sample overshoots.
> In practice, we monitor the inequality violation during sampling and adjust $\epsilon$ so that the sample does not overshoot and show sticky behavior at the boundary. Fortunately, it turns out that $\epsilon$ is less sensitive than $\alpha$ (Table 6 in Appendix), and it is often not hard to choose a good $\epsilon$.
>
> ---
> Thank you again for your positive review and helpful feedback. We'd appreciate the opportunity to discuss further if you have any additional questions or suggestions.
>
> ### Reference
> [2] M. Ledoux (1999). Concentration of measure and logarithmic Sobolev inequalities. Séminaire de Probabilités 33.

---

> > ### Comment · Reviewer_cZ9Z · 2025-08-08
> >
> > Thank you for the detailed response. I will maintain my current positive score. I hope the authors will revise the paper based on the discussions with all reviewers.

---

> > > ### Author Response · Authors · 2025-08-08
> > >
> > > We appreciate your valuable feedback, questions, and also your time. We will incorporate the outcomes of these discussions into the final version.

---

### Official Review · Reviewer_tmLS · 2025-07-03

**Clarity:** 3
**Significance:** 2
**Originality:** 3
**Rating:** 4
**Confidence:** 4

**Summary:**

This paper makes use of the landing technique -- incorporting constraints directly into the guiding stochastic differential equation (SDE) -- from non-convex optimization to handle both equality and inequality constraints in sampling. The proposed algorithm, Overdamped Langevin with LAnding (OLLA), avoids projection to the feasible set, and hence runs efficiently and is proven to converge exponentially fast. With Hutchinson trace estimation, to approximate the Hessian in the mean curvature term, OLLA could scale to high dimensions.

**Questions:**

Here are a few more questions and suggestions:

* In the submanifold $\Sigma$, the inequality constraint is $g(x)\leq 0$. However, in the set of active inequality constraints, why we have $g_i(x)\geq 0$?
* line 97: $g_{I_x}$ $\to$ $g_{|I_x|}$.
* In Section 3, the constants $\kappa$, $\delta$ and tabular neighborhood were given very abruptly and there lacked some background.
* line 123, $h(X_t)$ $\to$ $h_i(X_t)$.

**Ethical Concerns:**

["NO or VERY MINOR ethics concerns only"]

**Final Justification:**

Thanks for the authors' rebuttal that resolved my concerns. I will keep my score.

**Limitations:**

Yes.

**Quality:**

3

**Strengths And Weaknesses:**

# Strengths:

* The paper integrates the both equality and inequality constraints into Langevin dynamics and avoids projection.
* The proposed OLLA algorithm comes with theoretic convergence guarantee.
* OLLA-H algorithm scales to high dimensions and has SOTA performance compared with similar samplers with constraints.

# Weaknesses:

* It is a little disappointing that OLLA-H has comparable performance with CGHMC in the included tests. Hence it remains doubt whether practitioners choose one over the other.
* In all the simulation tests, the highest dimensionality is 700. The paper could be strengthened by including an example of dimension bigger than 1000.
* In all the simulation tests, the target distributions are relatively simple -- either uniform or Gaussian, though the constraints are complicated and versatile. This may not be very realistic --  in practice both distribution and constraints could be complicated. It would be more convincing to test OLLA(-H) on some more practical distributions with certain difficulty to sample.

---

> ### Author Rebuttal · Authors · 2025-07-31
>
> We thank the reviewer for the thoughtful and constructive feedback. Below, we (1) give a brief overview of what we changed, and then (2) leave itemized responses to address each of the comments, and we hope they can clarify your questions and concerns.
>
>
>
> ---
> **Overview of main changes/clarifications.**
> * We added two new experiments (Table A, B) to demonstrate the advantages of OLLA-H under practical scenarios: (A) a physics-motivated molecular system with a nontrivial potential and complex constraints, and (B) a high-dimensional Bayesian logistic regression with fairness and monotonicity constraints.
> * We clarify computational trade-offs between OLLA-H and CGHMC (and other projection-based baselines), including per-iteration complexity.
> * We clarify the notations of active inequality set $|I_x|$, and backgrounds on $\kappa, \delta$.
> ---
> >Weakness (1): It is a little disappointing that OLLA-H has comparable performance with CGHMC in the included tests. Hence, there remains doubt whether practitioners choose one over the other.
> >
> >Weakness (2): In all the simulation tests, the highest dimensionality is 700. The paper could be strengthened by including an example of a dimension bigger than 1000.
> >
> >Weakness (3): In all the simulation tests, the target distributions are relatively simple -- either uniform or Gaussian, though the constraints are complicated and versatile. This may not be very realistic -- in practice, both distribution and constraints could be complicated. It would be more convincing to test OLLA(-H) on some more practical distributions with certain difficulty in sampling.
>
> **Answer.** OLLA-H is preferable to CGHMC (also other baselines) when (1) large portion of inequality constraints stays inactive (loose boundaries $\Leftrightarrow$ small $|I_x|$), (2) constrained domain $\Sigma$ and potential function $\nabla f(x)$ are so complicated that they force small step size $\Delta t$ for CGHMC to have reasonable acceptance rate on Metropolis-Hastings correction, or (3) Newton steps for projection become larger under geometrically complicated $\Sigma$.
>
> **Computational picture (per step).**
>
> Following notations on paper, let $m$ be the number of equality constraints, $l$ the number of inequalities, and $I_x \subset \\{1,...,l\\}$ the currently active inequality set. OLLA-H forms and inverts a Gram matrix $G$ using only equalities and active inequalities:
> * $\text{cost(OLLA-H)} = \mathcal O(N \cdot (m+ |I_x|)^3)$
>
> where $N$ is the number of Hutchinson probes. For the slack-variable + Newton's methods (CLangevin / CHMC), they inflate the cubic of constraint block to $(m+l)$:
> * $\text{cost(CLangevin, CHMC)} = \mathcal O(N_{newton} \cdot (m+l)^3)$
>
> where $N_{newton}$ is the number of required iterations for Newton's method to converge.
> The CGHMC (Metropolis-Hastings + Newton's method) avoids slack variables, but samples are accepted or rejected based on Metropolis-Hastings correction:
> * $\text{cost(CGHMC)} = \mathcal O(N_{newton} \cdot m^3), \quad \text{efficiency} \propto \text{acceptance rate}.$
>
> When the boundary of the feasible set $\Sigma$ is tight and complex (many inequalities intermittently active), $|I_x|$ can grow as OLLA-H's cubic term reflects this. However, in the same regime, CGHMC's acceptance rate typically deteriorates due to frequent boundary interactions. Also, if the step size is too big to drastically change the energy of Hamiltonian of the system, the CGHMC's acceptance rate again drops, yielding low ESS/CPU-time. Simultaneously, projection based methods (CLangevin, CHMC, CGHMC) often require larger $N_{newton}$ and can be extremely slowed down if the Newton method stalls on irregular or high-dimensional $\Sigma$.
>
> **Additional Empirical Results (Appendix Table A,B).**
>
> For the experiment setup (potential, constraints), please refer to our reply to the reviewer 74bE.
>
> ****Table A - Molecular system with realistic potential (complex $\Sigma$, small $|I_x|$, large $N_{newton}$)****
>
> The below figure shows the estimation value of the test function $\mathbb E [|x|]^2$ with total CPU time to run (wrapped by brackets). The constraint violations for both equality constraints and inequality constraints are below 0.005 / 0.02, respectively, across all algorithms. Also, the $(n, m, l)$ configurations are given by (15, 7, 6), (30, 17, 36), (45, 27, 91), (60, 37, 171), (90, 57, 406)
> | Method \ dim | 15 | 30 | 45 | 60 | 90 |
> | --- | --- | --- | --- | --- | --- |
> | OLLA-H | 23.02 (22.51s) | 236.6 (42.48s) | 852.9 (67.38s) | 2152 (107s) | 7561 (257.8s) |
> | CLangevin | 36.44 (294.4s) | 228.5 (1611s) | 925.5 (3724s) | 2162 (6631s) | 7572 (15230s) |
> | CHMC | 31.1 (79.34s) | 266.8 (280.1s) | 881.3 (614s) | 2235 (1090s) | 7640 (2514s) |
> | CGHMC | 29.8 (49.96s) | 256.8 (107.9s) | 895.7 (174.8s) | 2174 (234.3s) | 7607 (415.1s) |
>
>
> In this setting, only a small subset of inequalities is active at each step (so $|I_x| \ll l$), which makes OLLA-H's Gram matrix compact and economic to compute. Also, due to the complexity of the potential and constrained domain $\Sigma$, the $N_{newton}$ remains relatively high $(\approx 20)$. Across the dimensions, OLLA-H attains comparable estimates of test functions while achieving best CPU-time improvements of at least 2x to 60x relative to other baselines, further widening the gap in effectiveness. This highlights OLLA-H's advantage when $\Sigma$ is intricate but sparsely active in inequalities.
>
> ****Table B - Bayesian logistic regression with fairness + monotonicity constraints (High-dimensional $\Sigma$)****
>
> The below figure shows the test NLL loss on UCI german credit data with total CPU time to run (wrapped by brackets). All baselines kept equality constraint biolations at $\leq 10^{-3}$ level. For inequality constraints, each algorithms achieve around $0.99$ (OLLA-H), $0.05$ (CLangevin), $0.02$ (CHMC), across all dimensions.
>
> | Method \ dim | 706 | 1986 | 4994 | 9986 | 49986 |
> | --- | --- | --- | --- | --- | --- |
> | OLLA-H | 0.603 (8.25s) | 0.541 (8.65s) | 0.551 (8.30s) | 0.564 (9.85s) | 0.577 (15.85s) |
> | CLangevin | 0.604 (16.45s) | 0.546 (16.55s) | 0.550 (19.75s) | 0.544 (20.55s) | 0.557 (28.50s) |
> | CHMC | 0.605 (18.10s) | 0.602 (19.40s) | 0.599 (21.35s) | 0.602 (27.05s) | 0.601 (52.20s) |
>
>
> Here, the target density is a non-Gaussian posterior, and constraints are also nonlinear functionals, both implemented via a two-layer neural network. In this experiment, the CGHMC suffers from a low-acceptance rate problem ($\le 5 \\% $) due to the complicated potential (or high step size) and frequent inequality activations. So, we dropped the result of CGHMC. While the results show the projection-based algorithms (CLangevin, CHMC) attain smaller inequality constraint violations over OLLA-H, the lower computational cost of OLLA-H is still highlighted with moderate constraint violations, which corresponds to our purpose of the algorithm.
>
> ---
>
> >Question (1): In the submanifold $\Sigma$, the inequality constraint is $g(x) \leq 0$, However, in the set of active inequality constraints, why do we have $g_i(x) \geq 0$?
>
> **Answer.** Thank you for pointing out. Our notation follows:
> * Feasible set: $\Sigma:= \\{x \in \mathbb R^d  \mid h(x) =0,\  g(x) \leq 0 \\}$
> * Set of active inequality constraints: $I_x := \\{i \in [l] \mid g_i(x) \geq 0\\}$
>
> Under this notation, $g_i(x) \geq 0$ will be flagged as "activated", and we enforce them to decay exponentially fast via our OLLA dynamics. We understand our sign conventions for inequalities may not align with other literature. For example, [1] use feasibility $g_i(x) \geq 0$ and define the active set as $I_x := \\{i \in [l] \mid g_i(x) \leq 0\\}$. However, this is equivalent to ours under a sign change of inequalities.
>
> ---
>
> >Suggestion (1): In section 3, the constants $\kappa$, $\delta$, and tubular neighborhood were given very abruptly, and there lacked some background.
>
> **Answer.** Thank you. We agree that these should be introduced earlier with some intuition. We will move the definitions to Preliminaries and add a short intuitions with cross references on the revision. Here, we leave concise definitions and roles for $\kappa, \delta$ and their backgrounds.
> * (Lemma C.1/C.3) $\kappa :=  \frac{1}{2} \min_{y \in \Sigma} \sigma_{min} ([\nabla h(y)^T, \nabla g_{I_y}(y)^T]^T ) >0$
>
> This constant is the proportionality constant used when showing that reducing $h$ and $g$ decreases the normal displacement $||x-\pi(x)||_2$. The existence of this constant is guaranteed by LICQ condition and determined by the manifold structure of $\Sigma$.
>
> * (Theorem C.1/C.2) $\delta$ is the reach of "recoverable tubular neighborhood" $\hat{U}\_\delta (\Sigma)$, which is defined by $\hat{U}_\delta (\Sigma) :=\\{x \in \mathbb R^d \mid ||h(x)||_2 <\delta, \ g(x) < \delta \\}$.
>
> For $x \in \hat{U}\_\delta (\Sigma)$, the nearest-point projection $\pi(x) \in \Sigma$ is well-defined and there exists a recovery map $\zeta$ such that, writing $y =\pi(x), p = h(x), q = g_{I_y}(x)$, we can recover $x$ by $x=\zeta(y,p,q)$. This structure underlies our analysis of the projected process $Y_t = \pi(X_t)$.
>
> ---
> >Suggestion (2): Typos in Line 97 ($g_{I_x} \rightarrow g_{|I_x|}$), Line 123 ($h(X_t) \rightarrow h_i(X_t)$.
> >
> **Answer.** Thank you for pointing out. We'll fix both errata.
>
> ---
> Thank you again for your positive review and constructive feedback regarding the paper. We'd appreciate the opportunity to discuss further if you have any additional questions or suggestions.
>
> ### Reference
>
> [1] M. Muehlebach and M. I. Jordan (2022). On constraints in first-order optimization: A view from non-smooth dynamical systems. JMLR 23(256).

---

> > ### Comment · Reviewer_tmLS · 2025-08-01
> >
> > Thanks to the authors for their efforts to answer all my questions and address my concerns.

---

> > > ### Author Response · Authors · 2025-08-08
> > >
> > > Thank you for the helpful feedback, questions, and your time. We will incorporate these discussions into the final version.

---

### Decision · Program_Chairs · 2025-09-17

**Decision:**

Accept (poster)

**Comment:**

The paper proposes a new and unified projection free sampling method based on Langevin dynamics that can handle both equality and inequality constraints. The paper also shows non-asymptotic convergence analysis. Given the strong results and potential applications, I recommend it to be accepted.